# Inverse Entropic Optimal Transport
# Solves Semi-supervised Learning via Data Likelihood Maximization

**Mikhail Persiianov** [1 2]  **Arip Asadulaev** [* 3]  **Nikita Andreev** [*]  **Nikita Starodubcev** [4]  **Dmitry Baranchuk** [4]
**Anastasis Kratsios** [5 6]  **Evgeny Burnaev** [1 7]  **Alexander Korotin** [1 7]

## Abstract

Learning conditional distributions $\pi^*(\cdot|x)$ is a central problem in machine learning, which is typically approached via supervised methods with paired data $(x, y) \sim \pi^*$. However, acquiring paired data samples is often challenging, especially in problems such as domain translation. This necessitates the development of *semi-supervised* models that utilize both limited paired data and additional unpaired i.i.d. samples $x \sim \pi_x^*$ and $y \sim \pi_y^*$ from the marginal distributions. The usage of such combined data is complex and often relies on heuristic approaches. To tackle this issue, we propose a new learning paradigm called **EBiEOT** that integrates both paired and unpaired data seamlessly using data likelihood maximization techniques. We demonstrate that our approach also connects intriguingly with inverse entropic optimal transport (OT). This finding allows us to apply recent advances in computational OT to establish an *end-to-end* learning algorithm to get $\pi^*(\cdot|x)$. In addition, we derive the universal approximation property, demonstrating that our approach can theoretically recover true conditional distributions with arbitrarily small error. Finally, we demonstrate through empirical tests that our method effectively learns conditional distributions using paired and unpaired data simultaneously. Source code: https://github.com/MuXauJl11110/EBiEOT.

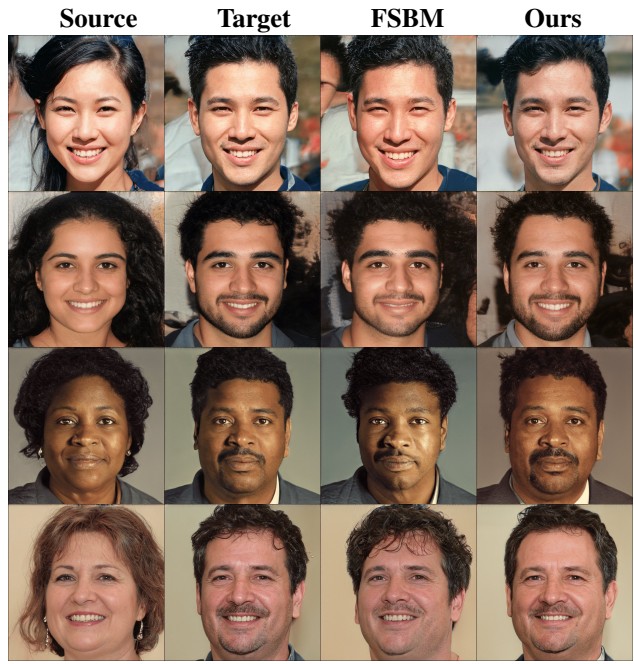

*Figure 1.* **Semi-supervised Woman-to-Man Translation:** A comparison between FSBM (Theodoropoulos et al., 2024) and our method. Both methods are applied within the ALAE latent space (Pidhorskyi et al., 2020) using $1024 \times 1024$ FFHQ images (Karras et al., 2019). See §5.3 for further experimental details.

## 1. Introduction

Recovering conditional distributions $\pi^*(y|x)$ from data is one of the fundamental problems in machine learning, which appears both in predictive and generative modeling. In predictive modeling, the standard examples of such tasks are the classification, where $x \in \mathbb{R}^{D_x}$ is a feature vector and $y \in \{y_1, \ldots, y_K\}$ is a class label, and regression, in which case $x$ is also a feature vector and $y \in \mathbb{R}$ is a real number. In generative modeling, both $x$ and $y$ are feature vectors in $\mathbb{R}^{D_x}, \mathbb{R}^{D_y}$, respectively, representing complex objects, and the goal is to find a transformation between them.

In this paper, we primarily focus on the continuous setting, where both $x$ and $y$ are multi-dimensional real-valued vectors, and the true joint distribution $\pi^*(x, y)$ is supported on $\mathbb{R}^{D_x} \times \mathbb{R}^{D_y}$. For completeness, we also discuss how the proposed framework can be adapted to classification

---
[1]Applied AI Institute, Moscow, Russia. [2]Yandex School of Data Analysis, Moscow, Russia. [3]MBZUAI, Abu Dhabi, UAE. [4]Yandex Research, Moscow, Russia. [5]Vector Institute, Toronto, Canada. [6]McMaster University, Hamilton, Canada [7]AXXX, Moscow, Russia.. Correspondence to: Mikhail Persiianov <persiianov.mi@gmail.com>.

*Proceedings of the 43rd International Conference on Machine Learning*, Seoul, South Korea. PMLR 306, 2026. Copyright 2026 by the author(s).

problems in Appendix B.2, and provide semi-supervised classification experiments on the MNIST dataset (Bottou et al., 1994) in Appendix C.2. The main focus of the paper is multi-dimensional probabilistic regression, which can also be viewed as a *domain translation* problem, since $x$ and $y$ typically represent feature vectors from different domains. The objective is to perform probabilistic prediction: given a new sample $x_{\text{new}}$ from the source domain, we aim to model the corresponding conditional distribution $\pi^*(y|x_{\text{new}})$ over outputs in the target domain.

It is natural to assume that learning the conditional distribution $\pi^*(y|x)$ requires access to input–target data pairs $(x, y) \sim \pi^*$, where $\pi^*$ denotes the true joint distribution of the data. In such cases, $\pi^*(y|x)$ can be modeled using standard supervised learning approaches, ranging from simple regression to conditional generative models (Mirza & Osindero, 2014; Winkler et al., 2019; Ardizzone et al., 2019; Hagemann et al., 2024). However, acquiring paired data may be costly, while getting unpaired samples $x \sim \pi_x^*$ or $y \sim \pi_y^*$ from two domains may be much easier and cheaper. This fact inspired the development of unsupervised (or unpaired) learning methods, e.g., (Zhu et al., 2017) among many others, which aim to somehow reconstruct the dependencies $\pi^*(y|x)$ with access to unpaired data only.

While both paired (supervised) and unpaired (unsupervised) domain translation approaches are being extremely well developed nowadays, surprisingly, the semi-supervised setup when both paired and unpaired data is available is much less explored. This is due to the challenge of designing learning objective (loss) which can simultaneously take into account both paired and unpaired data. A common approach involves heuristically combining standard paired and unpaired losses (cf. (Tripathy et al., 2019, §3.5), (Jin et al., 2019, §3.3), (Yang & Chen, 2020, §C), (Vasluianu et al., 2021, §3), (Panda et al., 2023, Eq. 8), (Tang et al., 2024, Eq. 8), (Theodoropoulos et al., 2024, §3.2), (Gu et al., 2023, §3)). However, as demonstrated in §5.1, these composite objectives fail to recover the true conditional distribution even in simple cases $D_x = D_y = 2$. This raises the question: *Can we design a simple loss to learn $\pi^*(y|x)$ that naturally integrates both paired and unpaired data?*

In our paper, we positively answer the above-raised question. Our **main contributions** are:

1. We introduce a novel loss function designed to facilitate the learning of conditional distributions $\pi^*(\cdot|x)$ using both paired and unpaired training samples drawn from $\pi^*$ (§3.1). This loss function is grounded in the well-established principle of likelihood maximization. A key advantage of our approach is its ability to support end-to-end learning, thereby *seamlessly* integrating both paired and unpaired data into the training process.

2. We demonstrate the theoretical equivalence between our proposed loss function and the *inverse entropic optimal transport* problem (§3.2). This finding enables us to leverage established computational OT methods to address challenges encountered in semi-supervised learning.

3. Building upon recent advances in computational optimal transport, we develop an *end-to-end* algorithm, called EBiEOT-GMM, based on a Gaussian mixture parameterization specifically designed to optimize the proposed likelihood-based objective (§3.3). For completeness, in Appendix A we additionally demonstrate that the same objective can be optimized using a fully neural-network-based parameterization, which we refer to as EBiEOT-NN.

4. We prove that our proposed parameterization satisfies the universal approximation property, namely, the proposed model class can approximate the target conditional distributions arbitrarily well under mild assumptions (§3.4).

Our empirical evaluation in §5 and Appendix C.2 demonstrates the impact of both unpaired and paired data on overall performance for domain translation and classification tasks. In particular, our findings show that the conditional distributions $\pi^*(\cdot|x)$ can be effectively learned even with a modest amount of paired data $(x, y) \sim \pi^*$, provided that sufficient auxiliary unpaired data $x \sim \pi_x^*$ and $y \sim \pi_y^*$ is available.

**Notations.** Throughout the paper, $\mathcal{X}$ and $\mathcal{Y}$ represent Euclidean spaces, equipped with the standard norm $\|\cdot\|$, induced by the inner product $\langle \cdot, \cdot \rangle$, i.e., $\mathcal{X} \stackrel{\text{def}}{=} \mathbb{R}^{D_x}$ and $\mathcal{Y} \stackrel{\text{def}}{=} \mathbb{R}^{D_y}$. The set of absolutely continuous probability distributions on $\mathcal{X}$ is denoted by $\mathcal{P}_{\text{ac}}(\mathcal{X})$. For simplicity, we use the same notation for both the distributions and their corresponding probability density functions. The joint probability distribution over $\mathcal{X} \times \mathcal{Y}$ is denoted by $\pi$ with corresponding marginals $\pi_x$ and $\pi_y$. The set of joint distributions with given marginals $\alpha$ and $\beta$ is represented by $\Pi(\alpha, \beta)$. We use $\pi(\cdot|x)$ for the conditional distribution, while $\pi(y|x)$ represents the conditional density at a specific point $y$. The differential entropy is given by $\text{H}(\beta) = -\int_{\mathcal{Y}} \beta(y) \log \beta(y)\, dy$.

## 2. Background

First, we recall the formulation of the domain translation problem (§2.1). We remind the difference between its paired, unpaired, and semi-supervised setups. Next, we recall the basic concepts of the inverse entropic optimal transport, which are relevant to our paper (§2.2).

### 2.1. Domain Translation Problems

The goal of *domain translation* task is to transform data samples from the source domain to the target domain while maintaining the essential content or structure. This approach

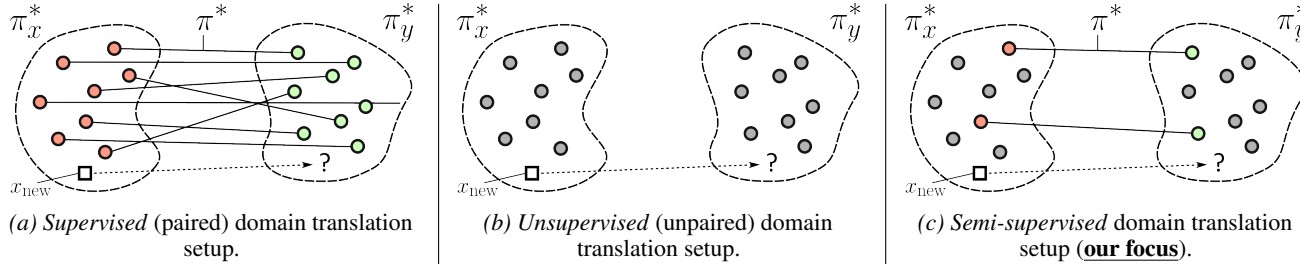

*(a) Supervised* (paired) domain translation setup.

*(b) Unsupervised* (unpaired) domain translation setup.

*(c) Semi-supervised* domain translation setup (**our focus**).

*Figure 2.* Visualization of domain translation setups. Red and green colors indicate paired training data $XY_{\text{paired}}$, while grey color indicates the unpaired training data $X_{\text{unpaired}}$, $Y_{\text{unpaired}}$.

is widely used in applications like computer vision (Zhu et al., 2017; Lin et al., 2018; Peng et al., 2023), natural language processing (Jiang et al., 2021; Morishita et al., 2022), audio processing (Du et al., 2022), etc. Domain translation task setups can be classified into supervised (paired), unsupervised (unpaired), and semi-supervised approaches based on the data used for training (Figure 2).

**Supervised domain translation** relies on matched examples from both the source and target domains, where each input corresponds to a specific output, enabling direct supervision during the learning process. Formally, this setup assumes access to a set of $P$ empirical pairs $XY_{\text{paired}} \overset{\text{def}}{=} \{(x_1, y_1), \ldots, (x_P, y_P)\} \sim \pi^*$ from some unknown joint distribution. The goal here is to recover the conditional distributions $\pi^*(\cdot|x)$ to generate samples $y|x_{\text{new}}$ for new inputs $x_{\text{new}}$ that are not present in the training data. While this task is relatively straightforward to solve, obtaining such paired training datasets can be challenging, as it often involves significant time, cost, and effort.

**Unsupervised domain translation**, in contrast, does not require direct correspondences between the source and target domains (Zhu et al., 2017, Figure 2). Instead, it involves learning to translate between domains using unpaired data, which offers greater flexibility but demands more advanced techniques to achieve accurate translation. Formally, we are given $Q$ unpaired samples $X_{\text{unpaired}} \overset{\text{def}}{=} \{x_1, \ldots, x_Q\} \sim \pi_x^*$ from the source distribution, and $R$ unpaired samples $Y_{\text{unpaired}} \overset{\text{def}}{=} \{y_1, \ldots, y_R\} \sim \pi_y^*$ from the target distribution. Our objective is to learn the conditional distributions $\pi^*(\cdot|x)$ of the unknown joint distribution $\pi^*$, whose marginals are $\pi_x^*$ and $\pi_y^*$, respectively. The unsupervised setup is inherently ill-posed, often yielding ambiguous solutions (Moriakov et al., 2020). Accurate translation requires constraints and regularization (Yuan et al., 2018). Still, it is highly relevant due to the prevalence of unpaired data in practice.

**Semi-supervised domain translation** integrates both paired and unpaired data to enhance the translation process (Tripathy et al., 2019; Jiang et al., 2023a). This approach leverages the precision of paired data to guide the model while exploiting the abundance of unpaired data to improve performance and generalization. Formally, the setup as-

sumes access to paired data $XY_{\text{paired}} \sim \pi^*$ as well as additional unpaired samples $X_{\text{unpaired}} \sim \pi_x^*$ and $Y_{\text{unpaired}} \sim \pi_y^*$. Note that paired samples can also be used in an unpaired manner. By convention, we assume $P \leq Q, R$, where the first $P$ unpaired samples are identical to the paired ones. The goal remains to learn the true conditional mapping $\pi^*(\cdot|x)$ using the available data. For extended discussion of real-world applications in which the semi-supervised setting arises naturally, see Appendix B.4.

### 2.2. Optimal Transport (OT)

We emphasize that this section is included primarily to clarify the connection between our proposed loss function (§3.1) and inverse entropic optimal transport (Dupuy et al., 2019), established in §3.2. Understanding this connection is not necessary for following the derivation of the loss itself, which is presented constructively in order to remain accessible to a broader audience.

For a more detailed discussion of entropic, weak, and inverse optimal transport, we refer the reader to Appendix B.1. Comprehensive introductions to the theoretical foundations of optimal transport can be found in (Villani et al., 2009; Santambrogio, 2015; Peyré & Cuturi, 2019).

**Entropic OT** (Genevay, 2019). Given source and target distributions $\alpha \in \mathcal{P}_{\text{ac}}(\mathcal{X})$ and $\beta \in \mathcal{P}_{\text{ac}}(\mathcal{Y})$, and a cost function $c^* : \mathcal{X} \times \mathcal{Y} \to \mathbb{R}$, the *entropic* optimal transport (EOT) problem is defined as:

$$\text{EOT}_{c^*, \varepsilon}(\alpha, \beta) \overset{\text{def}}{=} \min_{\pi \in \Pi(\alpha, \beta)} \mathbb{E}_{x, y \sim \pi}[c^*(x, y)] \\ -\varepsilon \mathbb{E}_{x \sim \alpha} \text{H}(\pi(\cdot|x)), \tag{1}$$

where $\varepsilon > 0$ is the regularization parameter; setting $\varepsilon = 0$ recovers the classic OT formulation (Villani et al., 2009) originally proposed by (Kantorovich, 1942). Under mild assumptions, a unique minimizer $\pi^* \in \Pi(\alpha, \beta)$ exists and is known as the *entropic optimal transport plan*. We note that in the literature, the entropy regularization term in (1) is typically written as either $-\varepsilon \text{H}(\pi)$ or $+\varepsilon \text{KL}(\pi \| \alpha \otimes \beta)$. These formulations are equivalent up to additive constants; see the discussion in (Mokrov et al., 2024, §2) or (Gushchin et al., 2023b, §1). In this paper, we adopt the *weak* formulation of

entropic OT (1) see (Gozlan et al., 2017; Backhoff-Veraguas et al., 2019; Backhoff-Veraguas & Pammer, 2022).

**Semi-dual EOT**. Under mild assumptions on $c^*, \alpha, \beta$, the further EOT formulation of (1) in semi-dual form holds:

$$\text{EOT}_{c^*,\varepsilon}(\alpha, \beta) = \max_f \left\{ \mathbb{E}_{x \sim \alpha} f^{c^*}(x) + \mathbb{E}_{y \sim \beta} f(y) \right\}, \tag{2}$$

where $f$ ranges over a subset of continuous functions (dual potentials) subject to mild boundedness conditions; see (Backhoff-Veraguas & Pammer, 2022, Eq. 3.3) for details. The term $f^c$ denotes the so-called *weak entropic c-transform of $f$*, defined as:

$$f^{c^*}(x) \stackrel{\text{def}}{=} \min_{\mu \in \mathcal{P}(\mathcal{Y})} \left\{ \mathbb{E}_{y \sim \mu}[c^*(x,y)] - \varepsilon \text{H}(\mu) - \mathbb{E}_{y \sim \mu} f(y) \right\}. \tag{3}$$

It has closed-form (Mokrov et al., 2024, Eq. 14) given by:

$$f^c(x) = -\varepsilon \log \int_{\mathcal{Y}} \exp\left( \frac{f(y) - c(x,y)}{\varepsilon} \right) dy. \tag{4}$$

**Inverse EOT.** The classical forward EOT problem (1) seeks an optimal transport plan $\pi^*$ between two given marginal distributions $\alpha$ and $\beta$ under a fixed cost function $c^*$. In contrast, the *inverse* EOT problem considers the reverse setting (Chan et al., 2025, §5.1): given a joint distribution $\pi^*$ with marginals $\pi_x^*$ and $\pi_y^*$, the goal is to recover a cost function $c^*$ such that $\pi^*$ is the EOT plan for $c^*$.

This inverse formulation is not uniquely defined in the literature – each version is typically tailored to specific applications (Stuart & Wolfram, 2020; Ma et al., 2020; Galichon & Salanié, 2022; Andrade et al., 2023). In this work, we adopt a version that aligns with our learning objective described in §3.1. This choice enables us, in §3.2, to formally relate our proposed loss to the inverse EOT framework. We further conjecture that this connection could potentially enable the application of advanced EOT solvers (e.g., diffusion Schrödinger bridges (Vargas et al., 2021; De Bortoli et al., 2021; Gushchin et al., 2023a; Shi et al., 2024; Gushchin et al., 2024b)) to enhance performance in semi-supervised learning scenarios, which we leave for future work.

With this motivation, we consider the *inverse EOT problem* as the following minimization problem:

$$c^* \in \arg\min_c \left[ \underbrace{\mathbb{E}_{x,y \sim \pi^*}[c(x,y)] - \overbrace{\varepsilon \mathbb{E}_{x \sim \pi_x^*} \text{H}(\pi^*(\cdot|x))}^{\text{not depend on } c}}_{\geq \text{EOT}_{c,\varepsilon}(\pi_x^*, \pi_y^*)} \right. \tag{5}$$
$$\left. - \text{EOT}_{c,\varepsilon}(\pi_x^*, \pi_y^*) \right],$$

where $c : \mathcal{X} \times \mathcal{Y} \to \mathbb{R}$ ranges over measurable cost functions. Consider the term $\text{EOT}_{c,\varepsilon}(\pi_x^*, \pi_y^*)$: due to entropic regularization and under mild assumptions: this expression

admits a unique optimal transport plan $\pi_c^*$ for every quadruple $(c, \varepsilon, \pi_x^*, \pi_y^*)$. While $\pi_c^*$ matches the marginals of $\pi^*$, its internal structure – i.e., the conditional distributions – may differ. The term $\mathbb{E}_{x,y \sim \pi^*}[c(x,y)] - \varepsilon \mathbb{E}_{x \sim \pi_x^*} \text{H}(\pi^*(\cdot|x))$ represents the *transportation cost* of using $c$ to transport mass according to $\pi^*$ (cf. the minimization objective in (1)). If the "inner" part of $\pi_c^*$ differs from that of $\pi^*$, this cost exceeds $\text{EOT}_{c,\varepsilon}(\pi_x^*, \pi_y^*)$. Therefore, the minimum of the full objective is achieved only when $\pi^*$ coincides with the optimal transport plan for some cost $c^*$, in which case the objective value is zero. Notably, the term $-\varepsilon \mathbb{E}_{x \sim \pi_x^*} \text{H}(\pi^*(\cdot|x))$ is independent of $c$ and can be omitted from the optimization. Additionally:

- **Invariance to $\varepsilon$.** Unlike the forward EOT problem (1), the inverse formulation (5) is invariant to the choice of the regularization parameter $\varepsilon > 0$, up to a rescaling of the cost function. Indeed, let $\varepsilon' > 0$ and define $c'(x,y) = \frac{\varepsilon'}{\varepsilon} c(x,y)$. Then, $\frac{1}{\varepsilon'} c'(x,y) = \frac{1}{\varepsilon} c(x,y)$, so the factor $\exp\left(-\frac{c(x,y)}{\varepsilon}\right)$ remains unchanged. Consequently, the corresponding entropic OT plan is identical: $\pi_{c,\varepsilon}^* = \pi_{c',\varepsilon'}^*$. Moreover, substituting $c'$ into (5) multiplies the entire objective by the constant factor $\varepsilon'/\varepsilon$, which does not affect the minimizers. Therefore, the inverse problem depends only on the ratio $c/\varepsilon$, and different choices of $\varepsilon$ lead to equivalent solutions after rescaling the cost.

- **Multiple solutions.** The inverse problem (5) generally admits *many* valid cost functions. For instance, $c^*(x,y) = -\varepsilon \log \pi^*(x,y)$ achieves the minimum by construction. More generally, any function of the form $c'(x,y) = -\varepsilon \log \pi^*(x,y) + u(x) + v(y)$ is also valid, since additive terms depending only on $x$ or $y$ do not affect the resulting EOT plan. In particular, setting $u(x) = \varepsilon \log \pi_x^*(x)$ and $v(y) = 0$ yields $c^*(x,y) = -\varepsilon \log \pi^*(y|x)$.

In practice, $\pi^*$ is known only through samples and not via its density. Therefore, closed-form expressions like $-\varepsilon \log \pi^*(x,y)$ or $-\varepsilon \log \pi^*(y|x)$ cannot be computed directly. This necessitates learning a parametric estimator $\pi^\theta$ to approximate the unknown conditional distributions.

## 3. Semi-supervised Domain Translation via Inverse EOT

In §3.1, we introduce a novel likelihood-based objective, called EBiEOT (Energy-Based inverse Entropic Optimal Transport), derived from the minimization of the KL divergence. In §3.2, we demonstrate the proposed loss is equivalent to solving the inverse EOT problem (5), thereby connecting optimal transport theory with our practical framework. To implement this approach, §3.3 introduces a tailored parameterization. We subsequently prove in §3.4 that this parameterization, when combined with our loss minimization, guarantees arbitrarily accurate reconstruction of

the true conditional plan under mild assumptions. For an extension to fully neural parameterizations, see Appendix A. Complete proofs for all results are located in Appendix E.

### 3.1. Loss Derivation

**Part I. Data likelihood maximization and its limitation.**
Our goal is to approximate the true distribution $\pi^*$ by some parametric model $\pi^\theta$, where $\theta$ represents the parameters of the model. To achieve this, we would like to employ the standard KL-divergence minimization framework, also known as data likelihood maximization. Namely, we aim to minimize:

$$\text{KL}\left(\pi^*\|\pi^\theta\right) = \mathbb{E}_{x,y\sim\pi^*} \log \frac{\pi_x^*(x)\pi^*(y|x)}{\pi_x^\theta(x)\pi^\theta(y|x)} = \quad (6)$$

$$\mathbb{E}_{x\sim\pi_x^*} \log \frac{\pi_x^*(x)}{\pi_x^\theta(x)} + \mathbb{E}_{x,y\sim\pi^*} \log \frac{\pi^*(y|x)}{\pi^\theta(y|x)} = \quad (7)$$

$$\text{KL}\left(\pi_x^*\|\pi_x^\theta\right) + \mathbb{E}_{x\sim\pi_x^*}\mathbb{E}_{y\sim\pi^*(\cdot|x)} \log \frac{\pi^*(y|x)}{\pi^\theta(y|x)} = \quad (8)$$

$$\underbrace{\text{KL}\left(\pi_x^*\|\pi_x^\theta\right)}_{\text{Marginal}} + \underbrace{\mathbb{E}_{x\sim\pi_x^*}\text{KL}\left(\pi^*(\cdot|x)\|\pi^\theta(\cdot|x)\right)}_{\text{Conditional}}. \quad (9)$$

It is clear that objective (9) splits into two independent components: the *marginal* and the *conditional* matching terms. Our focus will be on the conditional component $\pi^\theta(\cdot|x)$, as it is the necessary part for the domain translation. Note that the marginal part $\pi_x^\theta$ is not actually needed. The conditional part of (9) can further be divided into the following terms:

$$\mathbb{E}_{x\sim\pi_x^*}\mathbb{E}_{y\sim\pi^*(\cdot|x)}\left[\log\pi^*(y|x) - \log\pi^\theta(y|x)\right] = \\ -\mathbb{E}_{x\sim\pi_x^*}\text{H}\left(\pi^*(\cdot|x)\right) - \mathbb{E}_{x,y\sim\pi^*}\log\pi^\theta(y|x). \quad (10)$$

The first term is independent of $\theta$, so we obtain the following minimization objective:

$$\mathcal{L}(\theta) \overset{\text{def}}{=} -\mathbb{E}_{x,y\sim\pi^*}\log\pi^\theta(y|x). \quad (11)$$

It is important to note that minimizing (11) is equivalent to maximizing the conditional likelihood, a strategy utilized in conditional normalizing flows (Papamakarios et al., 2021, CondNF). However, a major limitation of this approach is its reliance solely on paired data from $\pi^*$, which can be difficult to obtain in real-world scenarios. In the following section, we modify this strategy to incorporate available unpaired data within a semi-supervised learning setup (§2.1).

**Part II. Solving the limitations via a tailored parameterization.** To address the above-mentioned issue and utilize unpaired data, we first use Gibbs-Boltzmann distribution (LeCun et al., 2006) density parameterization:

$$\pi^\theta(y|x) \overset{\text{def}}{=} \frac{\exp\left(-E^\theta(y|x)\right)}{Z^\theta(x)}, \quad (12)$$

where $E^\theta(\cdot|x) : \mathcal{Y} \to \mathbb{R}$ is *the Energy function*, and $Z^\theta(x) \overset{\text{def}}{=} \int_{\mathcal{Y}} \exp\left(-E^\theta(y|x)\right)\mathrm{d}y$ is the normalization constant. Substituting (12) into (11), we obtain:

$$\mathcal{L}(\theta) = \mathbb{E}_{x,y\sim\pi^*} E^\theta(y|x) + \mathbb{E}_{x\sim\pi_x^*}\log Z^\theta(x). \quad (13)$$

This objective already provides an opportunity to exploit the unpaired samples from the marginal distribution $\pi_x^*$ to learn the conditional distributions $\pi^\theta(\cdot|x) \approx \pi^*(\cdot|x)$. Namely, it helps to estimate the part of the objective related to the normalization constant $Z^\theta$. To incorporate independent samples from the second marginal distribution $\pi_y^*$, it is crucial to adopt a parameterization that separates the term in the energy function $E^\theta(y|x)$ that depends only on y. Thus, we propose:

$$E^\theta(y|x) \overset{\text{def}}{=} \frac{c^\theta(x,y) - f^\theta(y)}{\varepsilon}. \quad (14)$$

In fact, this parameterization allows us to decouple the cost function $c^\theta(x,y)$ and the potential function $f^\theta(y)$. Specifically, changes in $f^\theta(y)$ can be offset by corresponding changes in $c^\theta(x,y)$, resulting in the same energy function $E^\theta(y|x)$. For example, by setting $f^\theta(y) \equiv 0$ and $\varepsilon = 1$, the parameterization of the energy function $E^\theta(y|x)$ remains consistent, as it can be exclusively derived from $c^\theta(x,y)$. Substituting (14) into the energy term of (13), and using the identity $\mathbb{E}_{x,y\sim\pi^*} f^\theta(y) = \mathbb{E}_{y\sim\pi_y^*} f^\theta(y)$, yields *our final objective*, which integrates both paired and unpaired data:

$$\mathcal{L}(\theta) = \underbrace{\varepsilon^{-1}\mathbb{E}_{x,y\sim\pi^*}\left[c^\theta(x,y)\right]}_{\text{Joint, requires pairs } (x,y)\sim\pi^*} \\ - \underbrace{\varepsilon^{-1}\mathbb{E}_{y\sim\pi_y^*}f^\theta(y)}_{\text{Marginal, requires } y\sim\pi_y^*} + \underbrace{\mathbb{E}_{x\sim\pi_x^*}\log Z^\theta(x)}_{\text{Marginal, requires } x\sim\pi_x^*} \quad (15)$$

In Appendix E.1, we present a rigorous, step-by-step derivation starting from (6) and arriving at (15), using only *formal mathematical* transitions. Throughout this derivation, we assume that paired samples are drawn from the full joint distribution $\pi^*$. However, in practice the paired data may be restricted to a subset of $\pi^*$, discussed in Appendix B.3.

At this point, a reader may come up with 2 reasonable questions regarding (15):

1. How to perform the optimization of the proposed objective? This question is not straightforward due to the existence of the (typically intractable) normalizing constant $Z_\theta$ in the objective.

2. To which extent do the separate terms in (15) (paired, unpaired data) contribute to the objective, and which type of data is the most important to get the correct solution?

We answer these questions in §3.3 and §5. Before doing that, we show a surprising finding that our proposed objective actually solves the inverse entropic OT problem (5).

## 3.2. Relation to Inverse EOT

To show that (5) is equivalent to (15), we begin by substituting the semi-dual EOT formulation (2) into (5). After dropping the constant entropy term and utilizing the identity $\min(-g) = -\max g$, the expression simplifies to:

$$\min_{c,f} \left\{ \mathbb{E}_{x,y\sim\pi^*}[c(x,y)] - \mathbb{E}_{x\sim\pi_x^*} f^c(x) - \mathbb{E}_{y\sim\pi_y^*} f(y) \right\}. \quad (16)$$

Let $c^\theta$ and $f^\theta$ be parameterized by $\theta$. Using (4) and the energy function in (14), we obtain $(f^\theta)^{c^\theta}(x) = -\varepsilon \log Z^\theta(x)$. This shows that the (5) formulation is equivalent to our proposed loss (15) up to scaling factor $\varepsilon$.

This result shows that *inverse entropic OT can be viewed as a likelihood maximization problem*, enabling the use of established techniques like ELBO and EM (Barber, 2012; Alemi et al., 2018; Bishop & Bishop, 2023). It also reframes inverse EOT as a semi-supervised domain translation task. Notably, prior work on inverse OT has largely focused on discrete, fully paired settings (see §4).

## 3.3. Practical Parameterization

The most computationally intensive aspect of optimizing the loss function in (15) lies in calculating the integral for the normalization constant $Z^\theta$. To tackle this challenge, we propose a tailored parameterization, called EBiEOT-GMM, that yields closed-form expressions for each term in the loss function. Our proposed cost function parameterization $c^\theta$ is based on the log-sum-exp function (Murphy, 2012, §3.5.3):

$$c^\theta(x,y) = -\varepsilon \log \sum_{m=1}^{M} v_m^\theta(x) \exp\left( \frac{\langle a_m^\theta(x), y \rangle}{\varepsilon} \right), \quad (17)$$

where $\{v_m^\theta(x) : \mathbb{R}^{D_x} \to \mathbb{R}_+, a_m^\theta(x) : \mathbb{R}^{D_x} \to \mathbb{R}^{D_y}\}_{m=1}^{M}$ are arbitrary parametric functions, e.g., *neural networks*, with learnable parameters denoted by $\theta_c$. The parametric form of the cost is motivated by (Korotin et al., 2024), from which we derived a more general functional form appropriate for our setting. Therefore, we adopt a Gaussian mixture parameterization for the dual potential $f^\theta$:

$$f^\theta(y) = \varepsilon \log \sum_{n=1}^{N} w_n^\theta \mathcal{N}(y \mid b_n^\theta, \varepsilon B_n^\theta), \quad (18)$$

where $\theta_f \stackrel{\text{def}}{=} \{w_n^\theta, b_n^\theta, B_n^\theta\}_{n=1}^{N}$ are learnable parameters of the potential, with $w_n^\theta \geq 0$, $b_n^\theta \in \mathbb{R}^{D_y}$, and $B_n^\theta \in \mathbb{R}^{D_y \times D_y}$ being a symmetric positive definite matrix. Thereby, our framework comprises a total of $\theta \stackrel{\text{def}}{=} \theta_f \cup \theta_c$ learnable parameters. For clarity and to avoid notation overload, we will omit the superscript $^\theta$ associated with learnable parameters and functions in the subsequent formulas.

**Proposition 3.1** (Tractable normalization constant). *Our parameterization of the cost function* (17) *and dual potential*

(18) *delivers* $Z^\theta(x) \stackrel{\text{def}}{=} \sum_{m=1}^{M} \sum_{n=1}^{N} z_{mn}(x)$, *where*

$$z_{mn}(x) \stackrel{\text{def}}{=} w_n v_m(x) \exp\left( \frac{a_m^\top(x) B_n a_m(x) + 2b_n^\top a_m(x)}{2\varepsilon} \right).$$

The proposition offers a closed-form expression for $Z^\theta(x)$, which is essential for optimizing (15). Furthermore, our following proposition supplements the previous one and provides a method for sampling $y$ given a new sample $x_{\text{new}}$.

**Proposition 3.2** (Tractable conditional distributions). *From our parameterization of the cost function* (17) *and dual potential* (18) *it follows that the* $\pi^\theta(\cdot|x)$ *are Gaussian mixtures:*

$$\pi^\theta(y|x) = \frac{1}{Z^\theta(x)} \sum_{m=1}^{M} \sum_{n=1}^{N} z_{mn}(x) \mathcal{N}(y \mid d_{mn}(x), \varepsilon B_n), \quad (19)$$

*where* $d_{mn}(x) \stackrel{\text{def}}{=} b_n + B_n a_m(x)$ *and* $z_{mn}(x)$ *defined in Proposition 3.1.*

TRAINING. As stated in §2.1, since we only have access to samples from the distributions, we minimize the empirical counterpart of (15) via the gradient descent w.r.t. $\theta$:

$$\mathcal{L}(\theta) \approx \widehat{\mathcal{L}}(\theta) \stackrel{\text{def}}{=} \varepsilon^{-1} \frac{1}{P} \sum_{p=1}^{P} c^\theta(x_p, y_p)$$

$$-\varepsilon^{-1} \frac{1}{R} \sum_{r=1}^{R} f^\theta(y_r) + \frac{1}{Q} \sum_{q=1}^{Q} \log Z^\theta(x_q). \quad (20)$$

INFERENCE. According to our Proposition 3.2, the conditional distributions $\pi^\theta(\cdot|x)$ are Gaussian mixtures (19). As a result, sampling $y$ given $x$ is fast and straightforward.

## 3.4. Universal Approximation of the Proposed Parameterization

One may naturally wonder how expressive is our proposed parameterization of $\pi_\theta$ in §3.3. Below we show that this parameterization allows approximating any distribution $\pi^*$ that satisfies mild assumptions on boundness and regularity assumptions, see the details in Appendix E.4.

**Theorem 3.3** (Proposed parameterization guarantees universal conditional distributions). *Under mild assumptions on the joint distribution* $\pi^*$, *for all* $\delta > 0$ *there exists (a) an integer* $N > 0$ *and a Gaussian mixture* $f^\theta$ (18) *with* $N$ *components, (b) an integer* $M > 0$ *and cost* $c^\theta$(17) *defined by fully-connected neural networks* $a_m : \mathbb{R}^{D_x} \to \mathbb{R}^{D_y}, v_m : \mathbb{R}^{D_x} \to \mathbb{R}_+$ *with ReLU activations such that* $\pi^\theta$ *defined by* (12) *and* (14) *satisfies* $\text{KL}\left(\pi^* \| \pi^\theta\right) < \delta$.

# 4. Related Works

In this section, we briefly summarize the most relevant prior work; a more detailed discussion appears in Appendix B.5.

Existing semi-supervised domain translation approaches typically combine ad hoc objectives based on GAN losses and paired-data regularization (Chen et al., 2023; Panda et al., 2023), or use *keypoint-guided OT* (Gu et al., 2022), later extended to diffusion-based models (Gu et al., 2023; Theodoropoulos et al., 2024). Importantly, the paradigms outlined above do not offer any theoretical guarantees for reconstructing the conditional distribution $\pi^*(y|x)$, as they depend on heuristic loss constructions. We show that such approaches actually fail to recover the true plan even in toy 2-dimensional cases, refer to experiments in §5 for an illustrative example. **Inverse OT solvers:** works (Dupuy et al., 2019; Stuart & Wolfram, 2020) focuses on reconstructing cost functions (often in discrete settings), whereas our aim is to learn conditional distribution $\pi^\theta(\cdot|x)$. **Forward OT solvers:** Building on (Mokrov et al., 2024) and Gaussian-mixture parameterizations (Korotin et al., 2024; Gushchin et al., 2024a), our solver extends forward OT methods to general cost functions (Eq. (17)) and incorporates paired data through likelihood-based cost learning. Full details and additional discussion of **metric-learning** (Cuturi & Avis, 2014) provided in the Appendix B.5.

# 5. Experimental Illustrations

We evaluate the proposed solver on several tasks, including synthetic datasets (§5.1 and Appendix C.1), real-world data (§5.2), image translation (§5.3), and semi-supervised classification (Appendix C.2). Our PyTorch implementation is publicly available at the following repository[1]. Additional experimental details are provided in Appendix D.

## 5.1. Gaussian to Swiss Roll Mapping

**Setup.** For illustration, we adapt the experimental setup from (Korotin et al., 2024) to our purposes. We consider the task of learning conditional distributions from a Gaussian distribution $\pi_x^*$ to a Swiss Roll distribution $\pi_y^*$ (Figure 3a), guided by paired samples (Figure 3b) drawn from the ground-truth plan $\pi^*$. The ground-truth plan $\pi^*$ is obtained from a mini-batch OT plan after solving the *forward* OT problem with a specially designed cost that induces bimodal conditionals $\pi^*(\cdot|x)$. Specifically, the cost matrix is defined as $C = \min(C^{+\varphi}, C^{-\varphi})$, where $C^{\pm\varphi}$ contains pairwise $\ell_2$ distances between $x$ and $-y^{\pm\varphi}$, with $-y^{\pm\varphi}$ denoting the vector $-y$ rotated by an angle of $\varphi = \pm 90°$. In other words, each $x \sim \pi_x^*$ is mapped to a point $y$ on the opposite side of the Swiss Roll, rotated either by $+\varphi$ or $-\varphi$ (Figure 3c). Additional details on the generation of paired data are provided in Appendix D.4. We also consider an alternative analytically defined ground-truth plan of the form $y = T(x) + \xi$, where $\xi$ is an noise term, described in Appendix C.1.4. We evaluate each method by its ability to

recover these multi-modal conditional distributions. Unless stated otherwise, training uses $P = 128$ paired samples together with $Q = R = 1024$ unpaired samples. In Appendix C.1.2, we study the effect of varying the amounts of paired and unpaired data, while Appendix C.1.3 analyzes the sensitivity of the method to the choice of the parameters $N$ and $M$ defined in §3.3.

**Baselines.** We evaluate our method against several baselines (see Appendix D.1 for details):

1. **Semi-supervised log-likelihood** methods (Atanov et al., 2019; Izmailov et al., 2020): CNF (SS) and CGMM (SS).

2. *Semi-supervised methods*: Neural OT with pair-guided cost (Asadulaev et al., 2024, GNOT, Appendix E), Differentiable Cost-Parameterized Entropic Mapping Estimator (Howard et al., 2024, DCPEME), (Panda et al., 2023, parOT), (Gu et al., 2023, OTCS), Feedback Schrödinger Bridge Matching (Theodoropoulos et al., 2024, FSBM).

3. *Standard generative & predictive models*: MLP regression with $\ell^2$ loss, Unconditional GAN with $\ell^2$ loss supplement (Goodfellow et al., 2014, UGAN+$\ell^2$), Conditional GAN (Mirza & Osindero, 2014, CGAN), Conditional Normalizing Flow (Winkler et al., 2019, CondNF).

Note that some baselines can fully utilize both paired and unpaired data during training, while others rely solely on paired data. Refer to Table 7 for specifics on data usage.

**Metrics.** We evaluate the generated target distributions $\pi_y^*$ and conditional distributions $\pi^*(\cdot|x)$ using the Maximum Mean Discrepancy (MMD) (Gretton et al., 2012) and Sinkhorn divergence (Feydy et al., 2019) metrics approximating $\mathcal{W}_2$. Definitions of these metrics together with implementation details are provided in Appendix D.2. The corresponding results are reported in Table 5 for the setting considered in this section ($P = 128$, $Q = R = 1024$), as well as for the almost infinite amount of samples setting with $P = Q = R = 16$k discussed in Appendix C.1.1.

**Discussion.** The results of the aforementioned methods are depicted in Figure 3. Clearly, the Regression model simply predicts the conditional mean $\mathbb{E}_{y \sim \pi^*(\cdot|x)} y$, failing to capture the full distribution. The CGAN is unable to accurately learn the target distribution $\pi_y^*$, while the UGAN+$\ell^2$ fails to capture the underlying conditional distribution, resulting in suboptimal performance. The CondNF model suffers from overfitting, likely due to the limited availability of paired data $XY_{\text{paired}}$. Methods GNOT, DCPEME, parOT learn deterministic mapping and therefore are unable to capture the conditional distribution. Similar to parOT, both OTCS and FSBM build on the idea of key-points but are designed for stochastic setup. However, these methods fail to capture bi-modal conditional mappings, presumably due to a biased objective introduced by the artificial cost function that enforces alignment with key-points. The CondNF (SS)

---

[1] ⬤ https://github.com/MuXauJl11110/EBiEOT

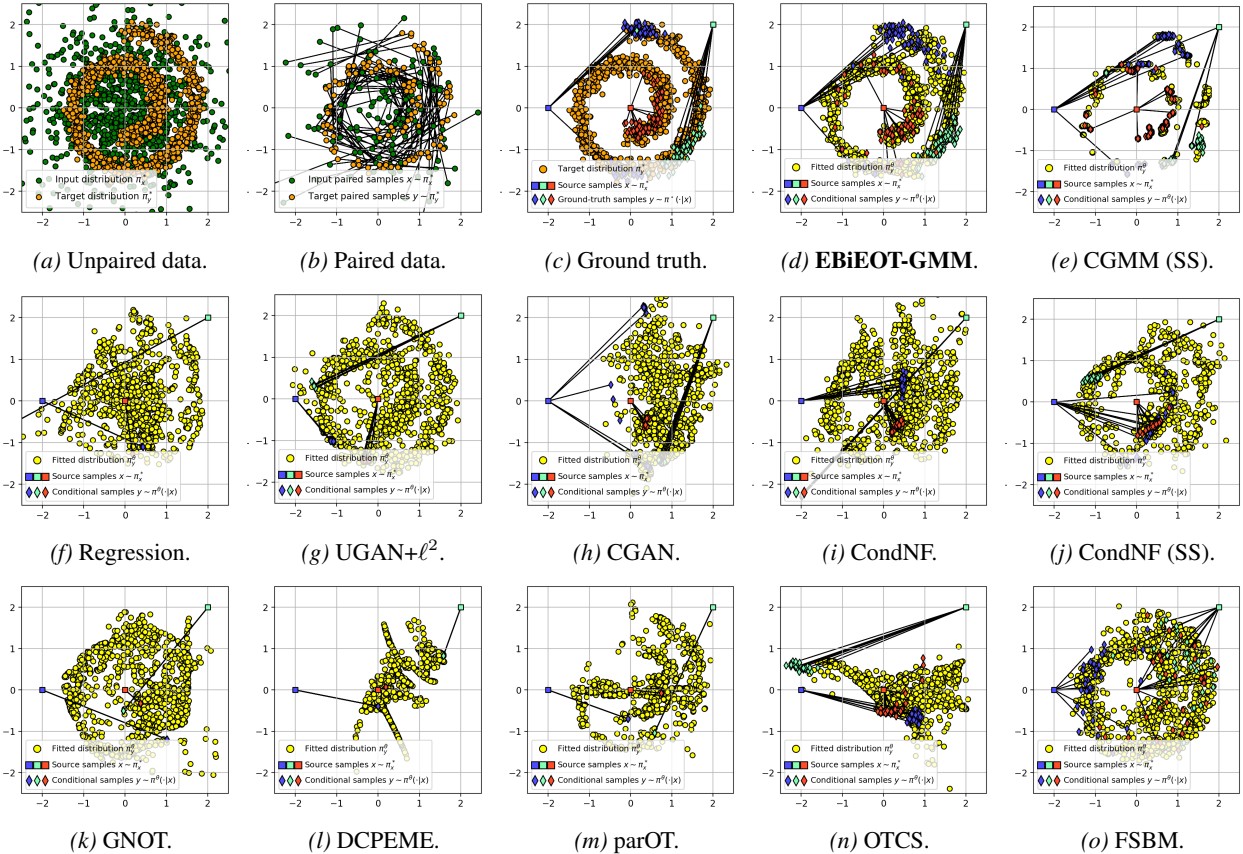

*Figure 3.* Learned mapping on the *Gaussian → Swiss Roll* task for $P = 128$ and $Q = R = 1024$.

does not provide improvement compared to CondNF, and CGMM (SS) model learns a degenerate solution, which is presumably due to the overfitting. As a sanity check, we evaluate all baselines using a large amount of paired data. Details are given in Appendix C.1.1. In fact, even in this case, almost all the methods fail to learn true $\pi^*(\cdot|x)$.

## 5.2. Weather Prediction

Here we aim to evaluate our proposed approach on real-world data. We consider the *weather prediction* dataset (Malinin et al., 2021; Rubachev et al., 2025). The data is collected from weather stations across the world and weather forecast physical models. It consists of 94 meteorological features, e.g., pressure, wind, humidity, etc., which are measured over a period of one year at different spatial locations.

**Setup.** Initially, the problem was formulated as the prediction and uncertainty estimation of the air temperature at a specific time and location. We expand this task to the probabilistic prediction of all meteorological features, thereby reducing reliance on measurement equipment in remote and difficult-to-access locations, e.g. the Polar regions.

**Metrics and baselines**. We evaluate the performance of our approach by calculating the *log-likelihood* (LL) on the

test target features. A natural baseline for this task is a probabilistic model that maximizes the likelihood of the target data. Thus, we implement an MLP that learns to predict the parameters of a mixture of Gaussians and is trained on the paired data only via the LL optimization (11). We also compare with semi-supervised LL methods CGMM (SS) and CondNF (SS). For completeness, we also add standard generative models. These models are trained using the available paired and unpaired data. Note that GAN models do not provide the density estimation and log-likelihood can not be computed for them. Therefore, we report Conditional Fréchet Distance (CFD): for each test $x$, we compute the FID (Heusel et al., 2017, Eq. 6) between predicted and true features $y$, then average over $x$.

**Discussion.** Tables 1 and 2 summarize our findings. From Table 1, the main observation is that even a small amount of unpaired data leads to substantial performance gains, underscoring the effectiveness of our semi-supervised formulation. Furthermore, Table 2 shows that our method obtains the best LL and is statistically competitive on CFD; CGMM (SS) attains a slightly lower CFD within overlapping uncertainty. For more detailed discussion regarding low-data regimes, see Appendix D.6.

| # Paired \ # Unpaired | Baseline | Ours | | | | | |
|---|---|---|---|---|---|---|---|
| | 0 | 5 | 10 | 50 | 100 | 250 | 500 |
| 5 | diverged | 9.4 ±.1 | 14.2 ±1.7 | 15.47 ±.02 | 16.6 ±.0 | 17.91 ±.07 | 9.40 ±.03 |
| 10 | 0.4 ±.2 | 9.48 ±.02 | 17.9 ±.3 | 18.5 ±.4 | 18.4 ±.2 | 18.8 ±.2 | 19.2 ±.3 |
| 25 | 3.5 ±.09 | 9.40 ±.03 | 18.3 ±.06 | 18.7 ±.2 | 18.8 ±.07 | 19.5 ±.1 | 19.8 ±.1 |
| 50 | 6.4 ±.05 | 9.47 ±.01 | 18.7 ±.2 | 18.9 ±.04 | 19.2 ±.2 | 19.8 ±.03 | 20.3 ±.4 |
| 90 | 6.5 ±.1 | 9.30 ±.05 | 19 ±.01 | 19.4 ±.05 | 19.4 ±.2 | 20.3 ±.05 | **20.5** ±.09 |

*Table 1.* The values of the test *log-likelihood* $\uparrow$ on the *weather prediction* dataset obtained for a different number of paired and unpaired training samples.

| | Ours | CGAN | UGAN+$\ell^2$ | CondNF | Regression | CGMM (SS) | CondNF (SS) |
|---|---|---|---|---|---|---|---|
| LL$\uparrow$ | **20.5** ±.09 | N/A | N/A | 1.29 ±.03 | N/A | 0.32 ±.03 | 0.52 ±.02 |
| CFD$\downarrow$ | **7.21** ±.04 | 15.79 ±1.11 | 15.44 ±1.89 | 18.72 ±.09 | 8.29 ±.04 | **7.17** ±.07 | 28.5 ±.5 |

*Table 2.* The values of the test *Log-Likelihood* (LL) and *Conditional Fréchet distance* (CFD) on the *weather prediction* dataset of our approach and baselines (500 unpaired and 90 paired samples).

### 5.3. Image Translation via ALAE

**Setup.** In this section, following the setup from (Theodoropoulos et al., 2024), we demonstrate our method capabilities for image translation in the 512-dimensional latent space of the ALAE encoder (Pidhorskyi et al., 2020), pretrained on the 1024×1024 FFHQ dataset (Karras et al., 2019). We consider two translation tasks: **(i)** Woman-to-Man translation and **(ii)** Old-to-Young translation. Similarly, we generate 2K paired samples using (Korotin et al., 2024).

**Baselines.** We used the publicly available FSBM (Theodoropoulos et al., 2024) implementation from GitHub[2]. However, due to reproducibility issues in the repository, we generated 2K paired samples ourselves via the procedure described in Appendix C.3 of the original paper.

**Metrics.** Metrics were computed using `TorchMetrics` (Falcon et al., 2020) with a batch size of 128 and averaged over three trainings with different seeds (LPIPS (Zhang et al., 2018), FID (Heusel et al., 2017), SSIM (Wang et al., 2004)). All metrics measure similarity between the generated and target distributions and are averaged across three independent runs with different seeds. Results are reported rounded to the first significant digit.

**Discussion.** Qualitative results are presented in Figure 1. Quantitative results, averaged across three independent runs, are reported in Table 3 and Table 4. Additional visual examples are provided in Figure 10 and Figure 11. Notably, our method achieves comparable performance to FSBM while being significantly more efficient: it requires only 3 minutes of training on an A100 GPU, whereas FSBM requires 5 hours on the same hardware. Additional implementation and training details are provided in Appendix D.5.

[2] https://github.com/panostheo98/FSBM

| Method | FID $\downarrow$ | SSIM $\uparrow$ | LPIPS $\downarrow$ |
|---|---|---|---|
| FSBM | $10.2 \pm 0.6$ | $0.5237 \pm 0.0005$ | $0.5625 \pm 0.0003$ |
| Ours | $\mathbf{9.3 \pm 0.1}$ | $\mathbf{0.5315 \pm 0.0002}$ | $\mathbf{0.5531 \pm 0.0006}$ |

*Table 3.* Metrics for Woman-to-Man translation.

| Method | FID $\downarrow$ | SSIM $\uparrow$ | LPIPS $\downarrow$ |
|---|---|---|---|
| FSBM | $11.5 \pm 0.6$ | $0.5285 \pm 0.0008$ | $0.5628 \pm 0.0004$ |
| Ours | $\mathbf{9.4 \pm 0.2}$ | $\mathbf{0.5361 \pm 0.0004}$ | $\mathbf{0.5560 \pm 0.0005}$ |

*Table 4.* Metrics for Old-to-Young translation.

## 6. Discussion

**Contributions & Potential impact.** Our framework offers a simple, non-minimax objective that naturally integrates both paired and unpaired data. We expect that these advantages, together with the connection to entropic optimal transport, will encourage adoption in more advanced semi-supervised methods, including approaches based on diffusion Schrödinger bridges (Vargas et al., 2021; De Bortoli et al., 2021; Shi et al., 2024) and flow matching (Chen et al., 2025; Balcerak et al., 2025).

The primary focus of this paper is semi-supervised learning for domain translation with continuous targets $y \in \mathbb{R}^{D_y}$. For completeness, Appendix B.2 discusses an extension of the proposed loss to discrete targets $y \in \mathcal{Y}_K^{D_y}$, where $\mathcal{Y}_K = \{y_1, \ldots, y_K\}$ denotes a finite set of categories. We view this discrete setting as an important direction for future research. In addition, Appendix C.2 demonstrates the applicability of our method to semi-supervised classification tasks.

**Limitations & Future Work.** A limitation of our method is its reliance on Gaussian Mixture parameterization (§3.3), which may affect scalability. To address this, we provide a proof of concept for fully neural parameterizations of the cost and potential functions below, with a detailed discussion in Appendix A. These parameterizations can be integrated into our loss via energy-based modeling (Song & Kingma, 2021) and could, in principle, scale to large image domains (Schröder et al., 2023; Yu et al., 2023; Zhu et al., 2024). A full investigation of such large-scale applications, however, lies beyond the scope of our methodological work.

## Acknowledgments

The work was supported by the grant for research centers in the field of AI provided by the Ministry of Economic Development of the Russian Federation in accordance with the agreement 000000C313925P4F0002 and the agreement №139-10-2025-033.

## Impact Statement

This paper presents work whose goal is to advance the field of Machine Learning. There are many potential societal consequences of our work, none which we feel must be

specifically highlighted here.

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

# Appendix

## A. Neural Parameterization

Throughout the main text, we parameterized the cost $c^\theta$ and potential $f^\theta$ using log-sum-exp functions and Gaussian mixtures (see §3.3), resulting in the method we refer to as EBiEOT-GMM. A natural question is whether more general parameterizations can be used within our framework, for example by directly parameterizing both $c^\theta$ and $f^\theta$ with neural networks. In this section, we answer this question affirmatively and introduce EBiEOT-NN, a neural-network-based version

of our method for optimizing the objective $\mathcal{L}(\theta)$ in (15).

## A.1. Algorithm Derivation

We note that a key advantage of our chosen parameterization (see §3.3) is that the normalizing constant $Z_\theta$ appearing in $\mathcal{L}(\theta)$ is available in the closed form. Unfortunately, this is not the case with general parameterizations of $c^\theta$ and $f^\theta$, necessitating the use of more advanced optimization techniques. While the objective $\mathcal{L}(\theta)$ itself may be intractable, we can derive its gradient, which is essential for optimization. The following proposition is derived in a manner similar to (Mokrov et al., 2024), who proposed methods for solving forward entropic OT problems with neural nets.

**Proposition A.1** (Gradient of our main loss (15))**.** *It holds that*

$$
\frac{\partial}{\partial\theta}\mathcal{L}(\theta) = \varepsilon^{-1}\Bigg\{ \mathbb{E}_{x,y\sim\pi^*}\left[\frac{\partial}{\partial\theta}c^\theta(x,y)\right] - \mathbb{E}_{y\sim\pi_y^*}\left[\frac{\partial}{\partial\theta}f^\theta(y)\right] \tag{21}
$$
$$
+ \mathbb{E}_{x\sim\pi_x^*}\mathbb{E}_{y\sim\pi^\theta(\cdot|x)}\left[\frac{\partial}{\partial\theta}\big(f^\theta(y)-c^\theta(x,y)\big)\right]\Bigg\}.
$$

The gradient formula eliminates the need for the intractable normalizing constant $Z_\theta$, but computing it still requires sampling from the current model $y\sim\pi^\theta(\cdot|x)$. Unlike the Gaussian mixture case in §3.3, we now only access the unnormalized density defined by $c^\theta$ and $f^\theta$, which is not necessarily a Gaussian mixture. To address this, we rely on standard methods for sampling from unnormalized densities, such as Markov Chain Monte Carlo (MCMC) (Andrieu et al., 2003). This enables practical gradient estimation and motivates the training procedure in Algorithm 1, where the conditional distribution is modeled as $\pi^\theta(y|x)\propto\exp\left(\frac{f^\theta(y)-c^\theta(x,y)}{\varepsilon}\right)$, with energy $\varepsilon^{-1}(c^\theta(x,y)-f^\theta(y))$.

---

**Algorithm 1:** EBiEOT-NN

---

**Input** : Paired samples $XY_{\text{paired}}\sim\pi^*$; unpaired samples $X_{\text{unpaired}}\sim\pi_x^*$, $Y_{\text{unpaired}}\sim\pi_y^*$;
potential network $f^\theta:\mathbb{R}^{D_y}\to\mathbb{R}$, cost network $c^\theta(x,y):\mathbb{R}^{D_x}\times\mathbb{R}^{D_y}\to\mathbb{R}$;
number of Langevin steps $K>0$, Langevin discretization step size $\eta>0$;
basic noise std $\sigma_0>0$; batch sizes $\hat{P},\hat{Q},\hat{R}>0$.
**Output**: trained potential network $f^{\theta^*}$ and cost network $c^{\theta^*}$ recovering $\pi^{\theta^*}(y|x)$ from (12).
**for** $i=1,2,\dots$ **do**

> Derive batches $\{\hat{x}_p,\hat{y}_p\}_{p=1}^{\hat{P}}=XY\sim\pi^*$, $\{x_n\}_{q=1}^{\hat{Q}}=X\sim\pi_x^*$, $\{y_r\}_{r=1}^{\hat{R}}=Y\sim\pi_y^*$;
> Sample basic noise $Y^{(0)}\sim\mathcal{N}(0,\sigma_0)$ of size $\hat{Q}$;
> **for** $k=1,2,\dots,K$ **do**
>
>> Sample $Z^{(k)}=\{z_q^{(k)}\}_{q=1}^{\hat{Q}}$, where $z_q^{(k)}\sim\mathcal{N}(0,1)$;
>> Obtain $Y^{(k)}=\{y_q^{(k)}\}_{q=1}^{\hat{Q}}$ with Langevin step:
>> $y_q^{(k)}\leftarrow y_q^{(k-1)}+\frac{\eta}{2\varepsilon}\cdot\texttt{stop\_grad}\Big(\frac{\partial}{\partial y}\left[f^\theta(y)-c^\theta(x_q,y)\right]\big|_{y=y_q^{(k-1)}}\Big)+\sqrt{\eta}z_q^{(k)}$
>
> $\widehat{\mathcal{L}}\leftarrow\frac{1}{\hat{P}}\left[\sum_{x_p,y_p\in XY}c^\theta(x_p,y_p)\right]+\frac{1}{\hat{Q}}\left[\sum_{x_q\in X,y_q^{(K)}\in Y^{(K)}}\big(f^\theta(y_q^{(K)})-c^\theta(x_q,y_q^{(K)})\big)\right]-\frac{1}{\hat{R}}\left[\sum_{y_r\in Y}f_\theta(y_r)\right]$;
> Perform a gradient step over $\theta$ by using $\frac{\partial\widehat{\mathcal{L}}}{\partial\theta}$;

---

In Algorithm 1, we use the Unadjusted Langevin Algorithm (ULA) (Roberts & Tweedie, 1996), a standard MCMC method. For an in-depth discussion on EBM training methods, see the recent surveys (Song & Kingma, 2021; Carbone, 2024).

Our proposed *inverse* OT algorithm is closely related to the *forward* OT framework in (Mokrov et al., 2024, Algorithm 1), with key distinctions: **(1)** it learns the cost function $c^\theta$ during training, and **(2)** it leverages both paired and unpaired data.

Below, we demonstrate a proof-of-concept performance of Algorithm 1 on two setups: a 2D example and colored images.

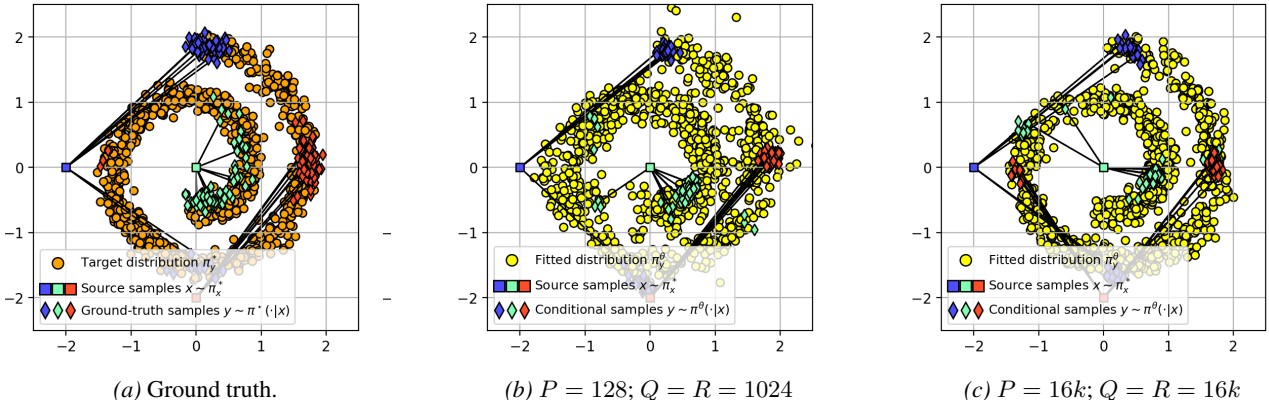

*(a)* Ground truth.   *(b)* $P = 128; Q = R = 1024$   *(c)* $P = 16k; Q = R = 16k$

*Figure 4.* Performance of the proposed neural parameterization in the *Gaussian → Swiss Roll* mapping task (§5.1). We use MLPs to parametrize both the potential function $f^\theta$ and the cost function $c^\theta$.

## A.2. Illustrative Example

**Setup.** We begin with a 2D example to showcase the capability of EBiEOT-NN to learn conditional plans using a fully neural network-based parameterization. Specifically, we conduct experiments on the *Gaussian → Swiss Roll* mapping problem (see §5.1) using two datasets: one containing 128 paired samples (described in §5.1) and another with 16k paired samples (see Appendix C.1.1).

**Discussion.** It is worth noting that the model's ability to fit the target distribution is influenced by the amount of labeled data used during training. When working with partially labeled samples (as shown in Figure 4b), the model's fit to the target distribution is less accurate compared to using a larger dataset. However, even with limited labeled data, the model still maintains good accuracy in terms of the paired samples. On the other hand, when provided with fully labeled data (see Figure 4c), the model generates more consistent results and achieves a better approximation of the target distribution. A comparison of the results obtained using EBiEOT-NN with neural network parameterization and those achieved using EBiEOT-GMM with Gaussian parameterization (Figure 3d) reveals that EBiEOT-NN exhibits greater instability. This observation aligns with the findings of (Mokrov et al., 2024, Section 2.2), which emphasize the instability and mode collapse issues commonly encountered when working with EBMs.

**Implementation Details.** We employ MLPs with hidden layer configurations of $[128, 128]$ and $[256, 256, 256]$, using $LeakyReLU(0.2)$ for the parameterization of the potential $f^\theta$ and the cost $c^\theta$, respectively. The learning rates are set to $lr_{\text{paired}} = 5 \times 10^{-4}$ and $lr_{\text{unpaired}} = 2 \times 10^{-4}$. The sampling parameters follow those specified in (Mokrov et al., 2024).

## A.3. Colored Images Example

**Setup.** We adapted an experiment from (Mokrov et al., 2024) using the colored MNIST dataset (Arjovsky et al., 2019). While the original task involved translating digit 2 into digit 3 using unpaired images, we modified the setup to demonstrate our method's ability to perform translations according to paired data. Namely, we created pairs by shifting the hue (Joblove & Greenberg, 1978) of the source images by $120°$. Specifically, for a source image with a hue $h$ in the range $0° \le h < 360°$, the target image's hue was set to $(h + 120°)$ mod $360°$.

**Discussion.** The results of this experiment are shown in Figure 5. Notably, the model successfully learned the color transformation using only 10 pairs (third row). Increasing the number of pairs to 200 further improved the quality of the translation (fourth row).

**Implementation Details.** We adopt the same parameters as in (Mokrov et al., 2024), except for the cost function:

$$c^\theta(x, y) = \frac{1}{D_y}\|U^\theta_{\text{net}} - y\|^2_2.$$

Here, the dimensions of source and target spaces are $D_x = D_y = 3 \times 32 \times 32$ and $U^\theta_{\text{net}} : \mathbb{R}^{D_x} \to \mathbb{R}^{D_y}$ is a neural net function with U-Net architecture (Ronneberger et al., 2015) with 16 layers. The first layer has 64 filters, and the number of filters doubles in each subsequent layer. The experiment was run for 10,000 iterations on a 2080 Ti GPU, completing in approximately 40 minutes.

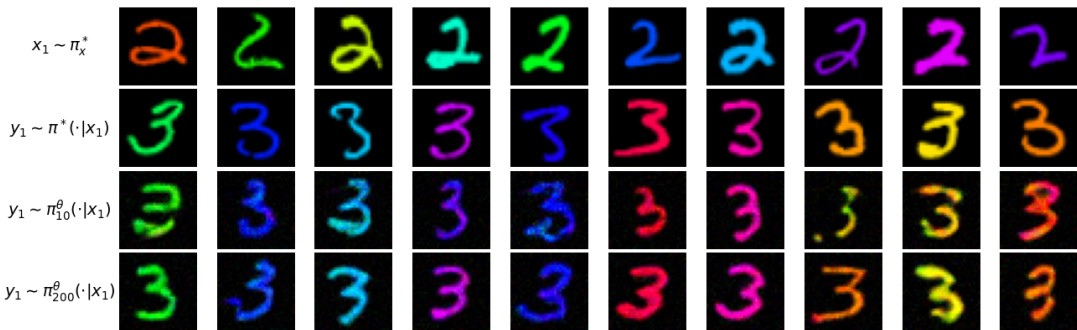

*Figure 5.* Performance of the proposed EBiEOT-NN method on the colored MNIST mapping task. Each pair consists of digits 2 and 3 with a hue shift of $120°$. The first row shows the source images, the second row displays target images with ground-truth colors, the third row presents the mapping results for 10 pairs in the train data, and the fourth row shows results for 200 pairs.

### A.4. Conclusion

It is important to recognize that the field of Energy-Based Models has undergone significant advancements in recent years, with the development of numerous scalable approaches. For examples of such progress, we refer readers to recent works by (Geng et al., 2024; Carbone et al., 2023; Du et al., 2021; Gao et al., 2021) and other the references therein. Additionally, we recommend the comprehensive tutorial by (Song & Kingma, 2021; Carbone, 2024) for an overview of training methods for EBMs. Given these advancements, it is reasonable to expect that by incorporating more sophisticated techniques into our basic Algorithm 1, it may be possible to scale the method to handle high-dimensional setups, such as image data. However, exploring these scaling techniques is beyond the scope of the current paper, which primarily focuses on the general methodology for semi-supervised domain translation. We leave the development of methods for further scaling our approach as a promising direction for future research.

## B. Additional Discussions

### B.1. Entropic/Weak/Inverse Optimal Transport

In this section, we explain our motivation for adopting the *entropic* OT formulation rather than the standard OT formulation (OT). Specifically, we focus on the *weak semi-dual* formulation of the entropic OT problem (Mokrov et al., 2024, §3.1), as opposed to its standard *semi-dual* form (Genevay, 2019, §4.3), and highlight its connections to the existing inverse entropic optimal transport frameworks in the literature.

**Classic OT.** Given source and target distributions $\alpha \in \mathcal{P}_{ac}(\mathcal{X})$ and $\beta \in \mathcal{P}_{ac}(\mathcal{Y})$, and a cost function $c^* : \mathcal{X} \times \mathcal{Y} \to \mathbb{R}$, the *primal* optimal transport problem (Villani et al., 2009) is defined as:

$$\text{OT}_{c^*}(\alpha, \beta) \stackrel{\text{def}}{=} \min_{\pi \in \Pi(\alpha,\beta)} \mathbb{E}_{x,y\sim\pi}[c^*(x, y)]. \tag{OT}$$

This formulation was originally introduced by Kantorovich (Kantorovich, 1942) as a relaxation of Monge's original problem (Monge, 1781), which is more restrictive because it does not allow mass to be split, resulting in *deterministic* solutions called optimal transport *maps*. However, as is well-known in optimal transport theory (see (Villani et al., 2009, §9)), solutions to problem (OT), called optimal transport *plans*, can still be deterministic. For example, when the cost is quadratic and the measures are absolutely continuous on the same space $\mathbb{R}^{D_x} = \mathbb{R}^{D_y}$, Brenier's theorem (Peyré & Cuturi, 2019, Remark 2.24) guarantees that the optimal transport plan is deterministic. Specifically, each $x$ is mapped deterministically to $y = T^*(x)$ for some optimal map $T^*$, meaning that the conditional distribution $\pi^*(y|x)$ collapses to a single point mass $\delta_{T^*(x)}$.

Such deterministic plans, however, are unsuitable for our semi-supervised domain translation setup, where a multimodal transport behavior of $\pi^*(y|x)$ may be necessary. Our synthetic experiments in §5.1 (Figure 3) illustrate these cases. To enforce mapping uniqueness while allowing *stochastic* (i.e., non-deterministic) mappings, a common approach is to regularize (OT) with an entropy term, which makes the objective strictly convex with respect to $\pi$, as discussed below.

**Entropic OT.** The work of (Cuturi, 2013) proposed regularizing (OT) with an entropy term, known as entropic OT (EOT), to improve computational tractability of OT (Genevay, 2019). Moreover, besides the computational advantages, the EOT

problem has a connection to the *Static Schrödinger Bridge* (SB) problem (Léonard, 2014):

$$\pi^* = \underset{\pi \in \Pi(\alpha,\beta)}{\arg\min} \, \text{KL}\left(\pi \| \pi^{\text{ref}}\right), \tag{SB}$$

where the aim of the problem is to find the transport plan $\pi \in \Pi(\alpha,\beta)$ closest to $\pi^{\text{ref}}$ in terms of the Kullback-Leibler (KL) divergence. Observe that EOT and the static SB problem are equivalent:

$$\min_{\pi \in \Pi(\alpha,\beta)} \text{KL}\left(\pi \| \pi^{\text{ref}}\right) = \min_{\pi \in \Pi(\alpha,\beta)} \mathbb{E}_{x,y \sim \pi} \log \frac{\pi(x,y)}{\pi^{\text{ref}}(x,y)} = \tag{22}$$

$$\min_{\pi \in \Pi(\alpha,\beta)} \left\{ \mathbb{E}_{x,y \sim \pi} \underbrace{\left[ -\log \pi^{\text{ref}}(x,y) \right]}_{\overset{\text{def}}{=} c^*(x,y)} - \text{H}\left(\pi\right) \right\} = \min_{\pi \in \Pi(\alpha,\beta)} \left\{ \mathbb{E}_{x,y \sim \pi}[c^*(x,y)] - \text{H}\left(\pi\right) \right\}. \tag{23}$$

Using the equivalence of the following formulations (see (Mokrov et al., 2024, Eq. 2–4) and (Gushchin et al., 2023b, Eq. 3–5) for details):

$$\begin{cases} \text{EOT}^{(1)}_{c^*,\varepsilon}(\alpha,\beta) \\ \text{EOT}^{(2)}_{c^*,\varepsilon}(\alpha,\beta) \\ \text{EOT}_{c^*,\varepsilon}(\alpha,\beta) \end{cases} = \min_{\pi \in \Pi(\alpha,\beta)} \mathbb{E}_{x,y \sim \pi}[c^*(x,y)] + \begin{cases} +\varepsilon \text{KL}\left(\pi \| \alpha \otimes \beta\right), \\ -\varepsilon H(\pi), \\ -\varepsilon \mathbb{E}_{x \sim \alpha} H(\pi(\cdot|x)), \end{cases}$$

we conclude that equation (23) is equivalent to (1) for $\varepsilon = 1$.

From the equations (22)–(23), we see that the cost function $c^*(x,y)$ defines a reference measure that determines the mapping we aim to reconstruct in the forward problem (1). Furthermore, since KL minimization is equivalent to maximum likelihood estimation, EOT is theoretically consistent with standard probabilistic modeling principles.

**Weak OT.** Following (Mokrov et al., 2024), we provide more details regarding *weak* OT. For a more rigorous treatment, see (Gozlan et al., 2017; Backhoff-Veraguas et al., 2019). Given a *weak* transport cost $C^* : \mathcal{X} \times \mathcal{P}(\mathcal{Y}) \to \mathbb{R}$, which penalizes the displacement of a point $x \in \mathcal{X}$ into a distribution $\pi(\cdot|x) \in \mathcal{P}(\mathcal{Y})$, the weak OT problem is defined as:

$$\text{WOT}_{C^*}(\alpha,\beta) \overset{\text{def}}{=} \min_{\pi \in \Pi(\alpha,\beta)} \mathbb{E}_{x \sim \alpha} C^*(x,\pi(\cdot|x)). \tag{p-WOT}$$

Just as in the classical OT problem (OT), the weak OT formulation (p-WOT) also enjoys *strong duality* under mild assumptions (see (Gozlan et al., 2017, Theorem 9.5); (Backhoff-Veraguas & Pammer, 2022, Theorem 3.3)). This means that the weak formulation (p-WOT) admits an equivalent *weak semi-dual* representation:

$$\text{WOT}_{C^*}(\alpha,\beta) = \max_{f \in \mathcal{C}(\mathcal{Y})} \left\{ \mathbb{E}_{x \sim \alpha} f^{C^*}(x) + \mathbb{E}_{y \sim \beta} f(y) \right\}, \tag{sd-WOT}$$

where $\mathcal{C}(\mathcal{Y})$ denotes the set of continuous functions over $\mathcal{Y}$ and $f^C$ so-called *weak C-transform*:

$$f^C(x) \overset{\text{def.}}{=} \min_{\mu \in \mathcal{P}(\mathcal{Y})} \left\{ C(x,\mu) - \mathbb{E}_{y \sim \mu} f(y) \right\}. \tag{24}$$

Furthermore, note that the EOT formulation in (1) can be seen as a special case of the weak OT problem (p-WOT), corresponding to the following weak transport cost $C^*_{\text{EOT}}$:

$$C^*_{\text{EOT}}(x,\pi(\cdot|x)) \overset{\text{def}}{=} \mathbb{E}_{y \sim \pi(\cdot|x)}[c^*(x,y)] - \varepsilon \text{H}\left(\pi(\cdot|x)\right). \tag{25}$$

Substituting expression above into (24), we obtain equation (3) for the weak *entropic c*-transform:

$$f^{c^*}(x) = \min_{\mu \in \mathcal{P}(\mathcal{Y})} \left\{ \mathbb{E}_{y \sim \mu}[c^*(x,y)] - \varepsilon \text{H}\left(\mu\right) - \mathbb{E}_{y \sim \mu} f(y) \right\},$$

which admits a closed-form expression given in (Mokrov et al., 2024, Eq. 14), and which we use in our work (4):

$$f^c(x) = -\varepsilon \log \int_{\mathcal{Y}} \exp \left( \frac{f(y) - c(x,y)}{\varepsilon} \right) \mathrm{d}y.$$

Furthermore, Appendix A.1 of (Mokrov et al., 2024) provides a detailed discussion of the relationship between the weak entropic $c$-transform and the so-called $(c, \varepsilon)$-*transform* (Genevay et al., 2019, 4.15), (Marino & Gerolin, 2020, Theorem 1.2):

$$v^{c,\varepsilon}(x) = -\varepsilon \log \mathbb{E}_{y\sim\beta} \left[ \exp\left( \frac{v(y) - c(x,y)}{\varepsilon} \right) \right], \tag{26}$$

which is used in the *semi-dual* formulation of the EOT problem (Genevay, 2019, §4.3):

$$\mathrm{EOT}^{\text{semi-dual}}_{c^*,\varepsilon}(\alpha, \beta) = \max_{v\in\mathcal{C}(\mathcal{Y})} \left\{ \mathbb{E}_{x\sim\alpha} v^{c^*,\varepsilon}(x) + \mathbb{E}_{y\sim\beta} v(y) \right\}. \tag{sd-EOT}$$

As noted in (Mokrov et al., 2024), the main difference between (4) and (26) lies in the integration measure: (26) integrates with respect to $\beta$, while (4) uses the standard Lebesgue measure.

For completeness, we present below the *dual* formulation of EOT with a slightly different regularization term, $+\varepsilon\mathrm{KL}\left(\pi\|\alpha\otimes\beta\right)$. As noted above, this is equivalent to our choice of regularization, but it is the version commonly used in inverse problems and will be discussed later:

$$\mathrm{EOT}^{\text{dual}}_{c^*,\varepsilon}(\alpha, \beta) = \max_{\substack{u\in\mathcal{C}(\mathcal{X}) \\ v\in\mathcal{C}(\mathcal{Y})}} \left\{ \mathbb{E}_{x\sim\alpha} u(x) + \mathbb{E}_{y\sim\beta} v(y) \right.$$
$$\left. - \mathbb{E}_{x,y\sim\alpha\otimes\beta} \left[ \exp\left( \frac{u(x) + v(y) - c(x,y)}{\varepsilon} \right) \right] \right\}, \tag{d-EOT}$$

where the optimization is performed over two Kantorovich potentials $u$ and $v$, in contrast to the single potential used in our formulation (15). With that said, we are ready to discuss the existing formulations of inverse entropic optimal transport.

**Inverse OT.** The use of the entropic formulation for inverse optimal transport was first proposed in (Du & Mordatch, 2019, Eq. 8). Their setup, identical to our formulation (5), restricted attention to bilinear cost functions of the form $c_A(x,y) = x^\top A y$, (Eq. 5), with the goal of recovering the matrix $A$ in a discrete setting. A subsequent work (Ma et al., 2020, Eq. 21) extended this idea to the continuous setting by introducing a loss function for learning cost functions, based on the dual formulation (d-EOT) of the EOT problem. As shown in (Andrade et al., 2023, Appendix A.1), their formulation and ours are equivalent, and both admit a maximum likelihood interpretation, consistent with our derivation in §3.1.

The most directly related approach is that of (Andrade et al., 2025, Lemma 1), which addresses the unbalanced OT framework (Chizat et al., 2018) while still relying on the dual formulation (d-EOT). They employ a linearly parameterized cost function (Eq. 4), but their focus is on establishing bounds for cost recovery, in contrast to our emphasis on semi-supervised domain translation. For further formulations of inverse OT, we refer readers to the works cited in the introduction of (Andrade et al., 2023).

## B.2. Discrete Spaces Extension

Our theoretical framework is not limited to continuous spaces $\mathcal{X}, \mathcal{Y}$. For instance, if the target space $\mathcal{Y}$ is discrete and takes values in a finite set $\mathbb{Y}_K = \{y_1, \ldots, y_K\}$, such as a set of categories, our method remains directly applicable. In this case, the dual potential $f^\theta$ (18) can be represented as a vector of length $K$, and the cost function $c^\theta(x,y)$ (17) can be implemented with a standard neural network. The partition function $Z^\theta(x)$ can then be computed as a finite sum over the $K$ terms:

$$Z^\theta(x) \overset{\text{def}}{=} \int_{\mathcal{Y}} \exp\left(-E^\theta(y|x)\right) \mathrm{d}y = \sum_{k=1}^{K} \exp\left( \frac{f^\theta(y_k) - c^\theta(x, y_k)}{\varepsilon} \right), \tag{27}$$

making the implementation straightforward. Note that the input $x$ can be either continuous or discrete - it does not affect the formulation. Challenges arise when $y$ is a more complex discrete object, such as a structured output like a sequence of $T$ tokens drawn from a dictionary of size $K$, i.e., $\mathbb{Y}_K^T$. In such cases, parameterizing $f_\theta$, computing $Z_\theta$, and sampling from the associated energy-based model become significantly more difficult, requiring advanced inference and training techniques, see (Holderrieth et al., 2025) for details. Discrete domains (Austin et al., 2021; Campbell et al., 2022; Gat et al., 2024; Ksenofontov & Korotin, 2025) have received considerable attention recently, and extending our methodology to such spaces represents a promising direction for future research.

### B.3. Partially Paired Data

Equation (15) assumes that paired training samples accurately reflect the true marginal distributions $\pi_x^*$ and $\pi_y^*$. One might argue that if these samples are biased or artificially selected, the objective would no longer align with the KL functional derivation in §3.1. However, we show below that the theoretical framework can be corrected under density-ratio assumptions.

Assume that the observed pairs $(x, y)$ come from a joint distribution $\pi_{\text{subset}}^*$ supported on a limited subset of the support of $\pi^*$, with $x$-marginal $\mu_x$ and conditional density $\pi^*(y|x)$. In this setting, the induced $y$-marginal is $\nu_y(y) = \mathbb{E}_{x \sim \mu_x} \pi^*(y|x)$ and the *ground-truth joint density* becomes

$$\pi_{\text{subset}}^*(x, y) = \mu_x(x)\pi^*(y|x).$$

Applying the same derivation as in §3.1, we obtain:

$$\text{KL}\left(\pi_{\text{subset}}^* \| \pi^\theta\right) = \underbrace{\text{KL}\left(\mu_x \| \pi_x^\theta\right)}_{\text{Marginal}} + \underbrace{\mathbb{E}_{x \sim \mu_x} \text{KL}\left(\pi^*(\cdot|x) \| \pi^\theta(\cdot|x)\right)}_{\text{Conditional}}. \tag{28}$$

Analogously to §3.1, we consider only the conditional term:

$$\mathbb{E}_{x \sim \mu_x} \mathbb{E}_{y \sim \pi^*(\cdot|x)} \left[\log \pi^*(y|x) - \log \pi^\theta(y|x)\right] = -\mathbb{E}_{x \sim \mu_x} \text{H}\left(\pi^*(\cdot|x)\right) - \mathbb{E}_{x, y \sim \pi_{\text{subset}}^*} \log \pi^\theta(y|x). \tag{29}$$

Thus, we recover the same conditional log-likelihood structure as in (11). Substituting the EBM parameterizations (12) and (14), we obtain

$$\mathbb{E}_{x, y \sim \pi_{\text{subset}}^*}[c(x, y)] - \mathbb{E}_{y \sim \nu_y} f(y) + \mathbb{E}_{x \sim \mu_x} \log Z^\theta(x) = \tag{30}$$

$$\mathbb{E}_{x, y \sim \pi_{\text{subset}}^*}[c(x, y)] - \mathbb{E}_{y \sim \pi_y^*}\left[\frac{\nu_y(y)}{\pi_y^*(y)} f(y)\right] + \mathbb{E}_{x \sim \pi_x^*}\left[\frac{\mu_x(x)}{\pi_x^*(x)} \log Z^\theta(x)\right]. \tag{31}$$

Introducing the *weights*:

$$w_x(x) = \frac{\mu_x(x)}{\pi_x^*(x)}, \qquad w_y(y) = \frac{\nu_y(y)}{\pi_y^*(y)}, \tag{32}$$

we obtain the corrected objective:

$$\mathcal{L}_q(\theta) = \underbrace{\varepsilon^{-1}\mathbb{E}_{x, y \sim \pi_{\text{subset}}^*}[c^\theta(x, y)]}_{\text{Joint, requires pairs } (x, y) \sim \pi_{\text{subset}}^*} - \underbrace{\varepsilon^{-1}\mathbb{E}_{y \sim \pi_y^*}[w_y(y) f^\theta(y)]}_{\text{Marginal, requires } y \sim \pi_y^*} + \underbrace{\mathbb{E}_{x \sim \pi_x^*}[w_x(x) \log Z^\theta(x)]}_{\text{Marginal, requires } x \sim \pi_x^*} \to \min_\theta. \tag{33}$$

A practical way to estimate the required ratios is classifier-based density ratio estimation, widely used in covariate-shift adaptation (Gretton et al., 2009; Sugiyama et al., 2012). To estimate a marginal ratio such as $w_x(x) = \mu_x(x)/\pi_x^*(x)$, we draw samples from the true marginal $\pi_x^*$ and the biased marginal $\mu_x$, label them as target (1) and observed (0), and train a probabilistic classifier $s_\varphi(x) = \text{Prob}(\text{target}|x)$. With balanced class priors, $\widehat{w}_x(x) = \frac{s_\varphi(x)}{1 - s_\varphi(x)}$. The same holds for $w_y(y)$. This method requires no density estimation. For recent advancement in density ratio estimation, please refer to (Nagumo & Fujisawa, 2024; Wang et al., 2025). Thus, even if the paired data are *artificially biased*, the loss remains correct as long as the true marginals are known and appropriate weights are applied.

### B.4. Examples of Semi-supervised Domain Translation Setups

In this section we outline some real-world scenarios, where *semi-supervised* setups are very natural.

- **Image Harmonization in Photo Editing** (Wang et al., 2023). Photo compositing often involves placing a foreground object into a new background, but realistic blending (e.g., matching lighting and color tone) is challenging. While only a small set of artist-labeled (paired) composites may be available, large collections of unlabeled (unpaired) composites can be gathered from the web.

- **Scene Stylization (e.g., Anime Rendering)** (Jiang et al., 2023b). Transforming real-world photos into anime-style renderings is popular in gaming and animation but is limited by the scarcity of labeled real–anime image pairs.

- **Image Enhancement for Outdoor Vision** (Li et al., 2019a; Liu et al., 2024; Cui et al., 2024; Li & Chang, 2025; Hou et al., 2025). Adverse weather and low-light conditions can compromise the visual systems of autonomous vehicles, such as self-driving cars and UAVs, leading to challenges in both decision-making and navigation. For a comprehensive overview of these scenarios and existing semi-supervised approaches, see (Mo et al., 2025).

- **Biomedical Image Registration (Microscopy)** (Skibbe et al., 2021). In neuroscience research, aligning images from different modalities (e.g., tracer vs. Nissl stain) is crucial but difficult due to modality shifts. Only a limited number of images can be manually registered (paired data), while many are unregistered (unpaired).

The examples above underscore the importance of developing methods for semi-supervised domain translation, which have applications in rendering, image editing, design, computer graphics, and autonomous driving, while also streamlining existing digital content creation pipelines. At the same time, it is important to recognize that the rapid advancement of generative models may have unintended consequences, potentially impacting certain jobs within these industries.

### B.5. Related Works

This section provides the detailed discussion of related work and methods that were only briefly summarized in §4, and includes additional coverage of metric learning.

**Semi-supervised models.** As discussed in §1, many existing semi-supervised domain translation methods combine paired and unpaired data by introducing multiple loss terms into *ad hoc optimization objectives*. Several works, such as (Jin et al., 2019, §3.3), (Tripathy et al., 2019, §3.5), (Oza et al., 2019, §C), (Paavilainen et al., 2021, §2), (Chen et al., 2023, §3.3), (Ren et al., 2023, §3) and (Panda et al., 2023, Eq. 8), employ GAN-based objectives, which incorporate the GAN losses (Goodfellow et al., 2014) augmented with specific regularization terms to utilize paired data. Although most of these methods were initially designed for the image-to-image translation, their dependence on GAN objectives enables their application to broader domain translation tasks. In contrast, the approaches introduced by (Mustafa & Mantiuk, 2020, §3.2) and (Tang et al., 2024, Eq. 8) employ loss functions specifically tailored for the image-to-image translation, making them unsuitable for the general domain translation problem described in §2.1.

Another line of research explores methods based on *key-point guided OT* (Gu et al., 2022), which integrates paired data information into the discrete transport plan. Building on this concept, (Gu et al., 2023) uses such transport plans as heuristics to train a conditional score-based model on unpaired or semi-paired data. Furthermore, recent work (Theodoropoulos et al., 2024) heuristically incorporates paired data into the cost function $c(x, y)$ in (1) with corresponding dynamical formulation.

Importantly, the paradigms outlined above do not offer any theoretical guarantees for reconstructing the conditional distribution $\pi^*(y|x)$, as they depend on heuristic loss constructions. We show that such approaches actually fail to recover the true plan even in toy 2-dimensional cases, refer to experiments in §5 for an illustrative example. We also note that there exist works addressing the question of incorporating unpaired data to the log-likelihood training (11) by adding an extra likelihood terms, see (Atanov et al., 2019; Izmailov et al., 2020). However, they rely on $x$ being a discrete object (e.g., a class label) and do not easily generalize to the continuous case, see Appendix D.1 for details.

**Inverse OT solvers**. As highlighted in §2.2, the task of inverse optimal transport (IOT) implies learning the cost function from samples drawn from an optimal coupling $\pi^*$. Existing IOT solvers (Dupuy et al., 2019; Li et al., 2019b; Stuart & Wolfram, 2020; Galichon & Salanié, 2022; Andrade et al., 2025) focus on reconstructing cost functions primarily from discrete marginal distributions, in particular, using the log-likelihood maximization techniques (Dupuy et al., 2019), see the introduction of (Andrade et al., 2023) for a review. Additionally, the recent work by (Shi et al., 2023) explores the IOT framework in the context of contrastive learning. In contrast, we develop a log-likelihood based approach aimed at learning conditional distributions $\pi^\theta(\cdot|x) \approx \pi^*(\cdot|x)$ using both paired and unpaired data but not the cost function itself.

**Forward OT solvers**. Our solver is based on the framework of (Mokrov et al., 2024), which proposed a *forward* solver for *unsupervised* domain translation. In contrast, our approach integrates the optimization of the cost function directly into the objective (equation (20)), allowing for effective utilization of paired data. Additionally, we extend the Gaussian Mixture parameterization proposed by (Korotin et al., 2024; Gushchin et al., 2024a), which was originally developed as a forward solver for entropic OT with a quadratic cost function $c^*(x, y) = \frac{1}{2}\|x - y\|_2^2$. Our work generalizes this solver to accommodate a wider variety of cost functions, as specified in equation (17). As a result, our approach also functions as a novel forward solver for these generalized cost functions.

Recent work by (Howard et al., 2024) proposes a framework for learning cost functions to improve the mapping between the

domains. However, it is limited by the use of deterministic mappings, i.e., does not have the ability to model non-degenerate conditional distributions.

Another work by (Asadulaev et al., 2024) introduces a neural network-based OT framework for semi-supervised scenarios, utilizing general cost functionals for OT. However, their method requires *manually* constructing cost functions which can incorporate class labels or predefined pairs. In contrast, our method dynamically adjusts the cost function during training, offering a more flexibility.

**Metric-learning and OT.** In addition to purely inverse OT approaches, there is a line of work that aims to learn the ground metric used by optimal transport. A seminal work (Cuturi & Avis, 2014) introduced *ground metric learning* in a supervised setting, where they optimize over metric matrices so that OT distances between labeled histograms better reflect the class structure. Building on this, (Huizing et al., 2022) propose *unsupervised ground metric learning* via what they call Wasserstein singular vectors. They jointly learn a ground metric on features and a distance between samples by finding positive singular vectors of the mapping from metric matrices to OT distance matrices. Their method uses stochastic approximation with entropic regularization and is scalable to high-dimensional data. More recently, the work (Auffenberg et al., 2025) analyze this fixed-point problem more deeply: they prove convergence for a stochastic fixed-point iteration (even in scenarios where classical contraction assumptions do not hold) and show that their framework naturally recovers Mahalanobis-type metrics and graph-Laplacian parameterizations as special cases.

In another direction, (Scarvelis & Solomon, 2023) introduce a *Riemannian metric-learning* framework: they parametrize a spatially-varying metric tensor as a neural network over a manifold, and optimize it so that OT distances under this learned geometry better explain meaningful interpolations, such as trajectories in scRNA data or bird migration. In graph-structured domains, (Heitz et al., 2021) learn ground metrics constrained to be geodesic distances on a graph, allowing a structured and efficient metric learning aligned with the graph topology.

Moreover, (Jawanpuria et al., 2025) propose to learn a symmetric positive definite (SPD) ground metric matrix by optimizing over the Riemannian manifold of SPD matrices, enabling the cost metric to adapt flexibly to data while jointly optimizing the OT distance. Finally, in the context of *domain adaptation*, (Kerdoncuff et al., 2020) present `MLOT`, which learns a global Mahalanobis metric that improves the alignment of source and target distributions under OT.

While these metric-learning works learn a distance function (or cost metric) via OT, they typically assume particular parametric forms (Mahalanobis, SPD matrices, or constructed on manifolds) and focus on matching distributions or aligning domains. In contrast, our approach learns conditional couplings $\pi^\theta(\cdot|x)$ (not just a ground cost), and integrates cost learning dynamically into a likelihood-based solver over paired and unpaired data. Moreover, our cost parameterization extends beyond classical metric forms, enabling more flexible and expressive cost functions (see Eq. (17)).

# C. Additional Experiments

## C.1. Gaussian to Swiss Roll Mapping

### C.1.1. BASELINES WITH THE LARGE AMOUNT OF DATA (16K)

In this section, we show the results of training of the baselines on the large amount of both paired (16k) and unpaired (16k) data (Figure 6). Recall that the ground truth $\pi^*$ is depicted in Figure 3c.

**Metrics.** As in §5.1, we evaluate the agreement between the target and learned conditional distributions using Maximum Mean Discrepancy (MMD) (Gretton et al., 2012) and $\mathcal{W}_2$, approximated with the Sinkhorn divergence (Feydy et al., 2019). Definitions of both metrics and implementation details are provided in Appendix D.2. The quantitative results are reported in Table 5. The trends observed in Table 5 are consistent with the qualitative results shown in Figure 3 for $P = 128$ and Figure 6 for 16k samples. In the large-sample regime, most methods successfully reconstruct the marginal distributions, but still have difficulty accurately learning the conditional distribution.

**Discussion.** Regression fails to learn anything meaningful due to the averaging effect (Figure 6f). In contrast, the unconditional GAN+$\ell^2$ (Figure 6g) nearly succeeds in generating the target data $\pi_y^*$, but the learned plan is still incorrect, also due to the averaging effect. Given a sufficient amount of training data, Conditional GAN (Figure 6h) nearly succeeds in learning the true conditional distributions $\pi^*(\cdot|x)$. The same applies to the conditional normalizing flow (Figure 6i), but its results are slightly worse, presumably due to the limited expressiveness of invertible flow architecture.

Experiments using the natural semi-supervised loss function in (36) demonstrate that this loss function can reasonably

recover the conditional mapping with both CondNF (Figure 6j) and CGMM (Figure 6e) parameterizations. However, it requires significantly more training data compared to our proposed loss function (15). This conclusion is supported by the observation that the CGMM model trained with (36) tends to overfit, as shown in Figure 3e. In contrast, our method, which uses the objective (15), achieves strong results, as illustrated in Figure 3d.

Other methods, unfortunately, also struggle to handle this illustrative 2D task effectively, despite their success in large-scale problems. This discrepancy raises questions about the theoretical justification and general applicability of these methods, particularly in scenarios where simpler tasks reveal limitations not evident in more complex settings.

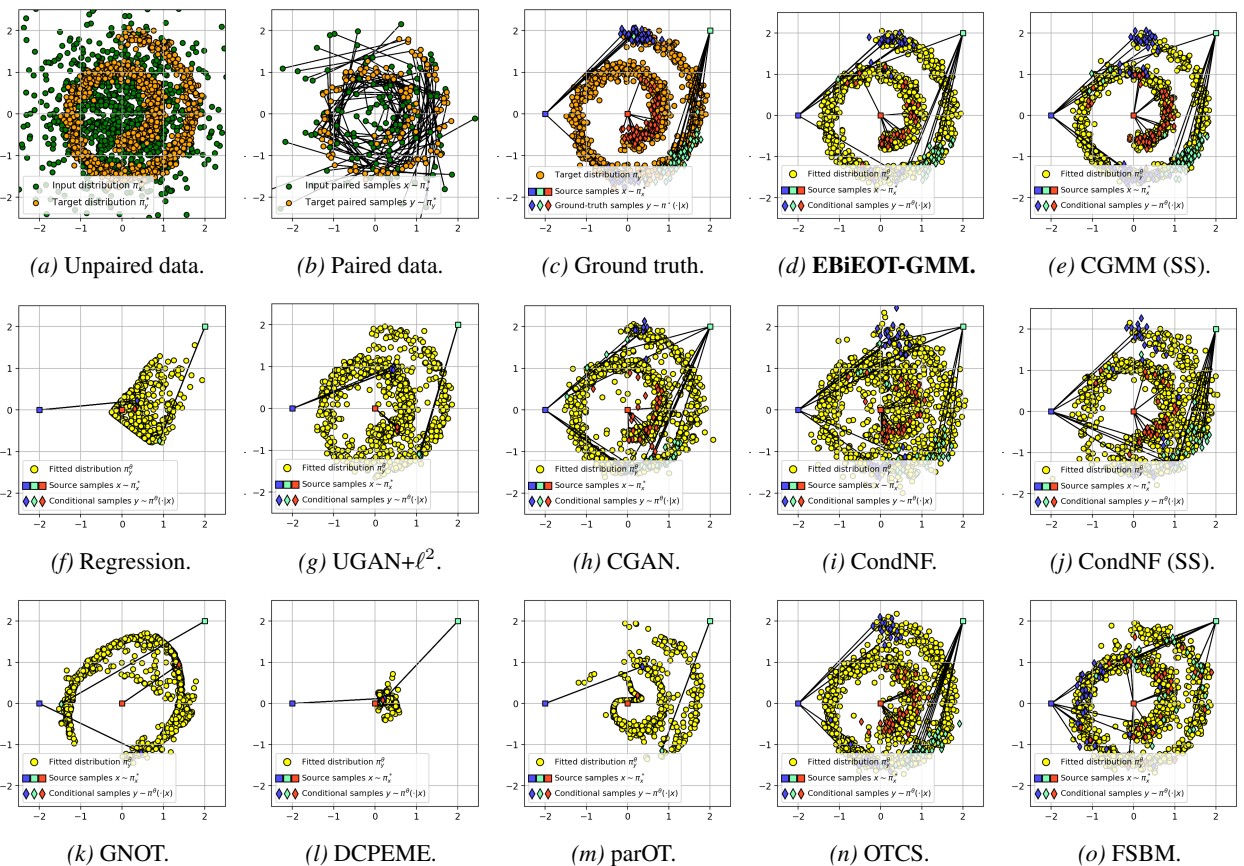

*Figure 6.* Comparison of the mapping learned by baselines on *Gaussian → Swiss Roll* task (§5.1). We use $P = 16k$ paired data, $Q = R = 16k$ unpaired data for training.

| Method Order | $P = 128, Q = R = 1024$ | | | | $P = 16k, Q = R = 16k$ | | | |
|---|---|---|---|---|---|---|---|---|
| | Unconditional | | Conditional | | Unconditional | | Conditional | |
| | MMD ↓ $[10^{-5}]$ | $\mathcal{W}_2$ ↓ $[10^{-2}]$ | MMD ↓ $[10^{-3}]$ | $\mathcal{W}_2$ ↓ $[10^{-2}]$ | MMD ↓ $[10^{-5}]$ | $\mathcal{W}_2$ ↓ $[10^{-2}]$ | MMD ↓ $[10^{-3}]$ | $\mathcal{W}_2$ ↓ $[10^{-2}]$ |
| Regression | $8.25_{\pm 5.39}$ | $3.77_{\pm 0.58}$ | $37.61_{\pm 0.00}$ | $89.76_{\pm 0.00}$ | $59.90_{\pm 4.85}$ | $33.40_{\pm 0.92}$ | $18.35_{\pm 0.00}$ | $48.42_{\pm 0.00}$ |
| CGAN | $666.68_{\pm 55.69}$ | $21.78_{\pm 1.31}$ | $41.72_{\pm 0.07}$ | $101.09_{\pm 0.09}$ | $15.65_{\pm 17.27}$ | $0.98_{\pm 0.46}$ | $22.48_{\pm 0.39}$ | $96.24_{\pm 1.03}$ |
| UGAN+$\ell^2$ | $46.60_{\pm 18.81}$ | $1.93_{\pm 0.44}$ | $40.32_{\pm 0.00}$ | $94.35_{\pm 0.04}$ | $23.89_{\pm 13.99}$ | $2.70_{\pm 0.67}$ | $9.55_{\pm 0.01}$ | $\mathbf{30.20}_{\pm 0.01}$ |
| CondNF | $39.76_{\pm 17.11}$ | $\mathbf{0.02}_{\pm 3.00}$ | $\mathbf{26.56}_{\pm 0.10}$ | $142.93_{\pm 0.04}$ | $30.83_{\pm 18.58}$ | $1.88_{\pm 0.48}$ | $20.59_{\pm 0.29}$ | $84.79_{\pm 0.61}$ |
| CondNF (SS) | $70.74_{\pm 8.70}$ | $3.84_{\pm 0.10}$ | $26.94_{\pm 0.22}$ | $\mathbf{71.09}_{\pm 0.36}$ | $57.42_{\pm 2.80}$ | $0.94_{\pm 1.19}$ | $14.77_{\pm 0.16}$ | $69.69_{\pm 0.28}$ |
| OTCS | $102.52_{\pm 14.18}$ | $19.61_{\pm 0.86}$ | $39.38_{\pm 0.03}$ | $91.73_{\pm 0.06}$ | $\mathbf{6.05}_{\pm 3.32}$ | $\mathbf{0.69}_{\pm 0.09}$ | $\mathbf{9.13}_{\pm 0.06}$ | $54.11_{\pm 0.25}$ |
| FSBM | $15.88_{\pm 7.89}$ | $\mathbf{0.93}_{\pm 0.19}$ | $33.75_{\pm 0.23}$ | $126.06_{\pm 0.44}$ | $\mathbf{4.65}_{\pm 5.49}$ | $\mathbf{0.52}_{\pm 0.09}$ | $30.88_{\pm 0.16}$ | $126.17_{\pm 0.47}$ |
| EBiEOT-GMM **(Ours)** | $\mathbf{4.19}_{\pm 0.84}$ | $\mathbf{0.43}_{\pm 0.34}$ | $28.35_{\pm 0.40}$ | $100.59_{\pm 0.63}$ | $6.18_{\pm 3.55}$ | $1.00_{\pm 0.14}$ | $35.72_{\pm 0.27}$ | $109.16_{\pm 0.58}$ |

*Table 5.* Comparison of methods on the 128 and 16k datasets using MMD and $\mathcal{W}_2$ metrics in both unconditional and conditional settings. Results are reported as mean ± standard deviation over multiple runs. The best results among methods with overlapping confidence intervals are highlighted in bold.

### C.1.2. ABLATION STUDY

In this section, we conduct an ablation study to address the question posed in §3.1 regarding how the number of source and target samples influences the quality of the learned mapping. The results, shown in Figure 7, indicate that the quantity of target points $R$ has a greater impact than the number of source points $Q$ (compare Figure 7c with Figure 7b). Additionally, it is evident that the inclusion of unpaired data helps mitigate over-fitting, as demonstrated in Figure 7a.

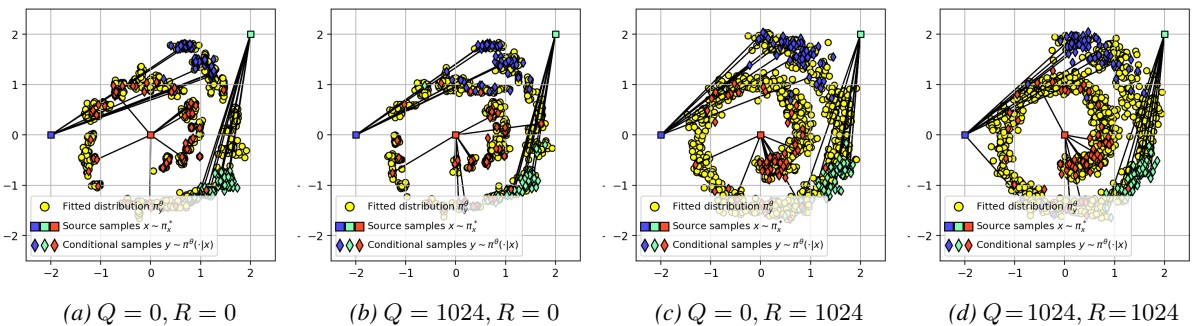

| *(a)* $Q = 0, R = 0$ | *(b)* $Q = 1024, R = 0$ | *(c)* $Q = 0, R = 1024$ | *(d)* $Q = 1024, R = 1024$ |

*Figure 7.* Ablation study analyzing the impact of varying source and target data point quantities on the learned mapping for the *Gaussian → Swiss Roll* task (using $P = 128$ paired samples).

### C.1.3. SENSITIVITY ANALYSIS

**Setup.** We performed an ablation study to investigate the sensitivity of EBiEOT-GMM to the number of Gaussian components $M$ and $N$. Specifically, we evaluated all combinations on the grid $M, N \in \{8, 16, 32, 64\}$, including the symmetric setting $M = N$. The ground-truth conditional mapping was constructed using the same procedure as in §5.1, while varying the numbers of source and target Gaussian components according to the selected values of $M$ and $N$.

**Metrics.** For evaluation, we computed MMD (Gretton et al., 2012) and Sinkhorn divergence (Feydy et al., 2019) with regularization parameter $\varepsilon = 0.001$, which serves as an approximation of $\mathcal{W}_2$; see Appendix D.2.

**Discussion.** The results are presented in Figure 8. Increasing the number of Gaussian components improves the conditional metrics while degrading the unconditional ones. This behavior is expected: a larger number of components increases the flexibility of the model and may lead to overfitting of the conditional distribution. The results show that the model quite sensitive to $N$ and $M$. At the same time, the EBiEOT-GMM parameterization remains computationally lightweight, making hyperparameter search relatively inexpensive. In practice, the number of components can be efficiently selected using automated optimization frameworks such as Optuna (Akiba et al., 2019).

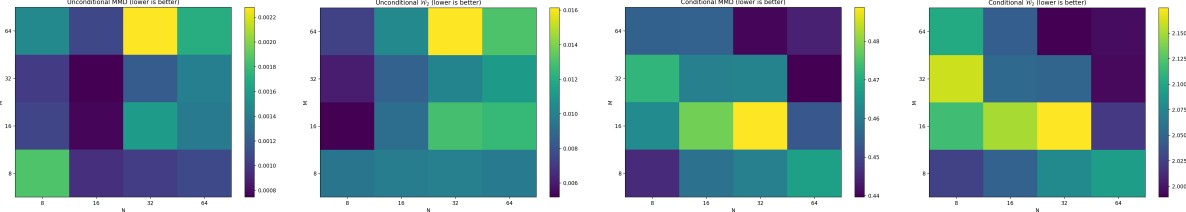

*Figure 8.* Results for sensitivity analysis described in Appendix C.1.3 for experiment §5.1.

### C.1.4. ADDITIONAL ANALYTICAL CONDITIONAL DISTRIBUTION

**Setup.** To demonstrate the behavior of our method in scenarios where the ground-truth conditional plan is not induced by optimal transport (Flamary et al., 2021), we consider an experiment in which the conditional distribution $\pi^*(\cdot|x)$ is analytically known. We construct the target distribution by applying a deterministic mapping from the source distribution onto a Swiss-roll manifold, followed by point dependent noise. For each source point $x$, we first compute $r(x) = \tanh(\|x\|)$. This value is then mapped to the Swiss-roll parameter range: $t(x) = t_{\min} + (t_{\max} - t_{\min}) \, r(x)$, where we set $t_{\min} = \frac{3}{2}\pi$ and $t_{\max} = \frac{9}{2}\pi$. The final target samples are generated using the Swiss-roll transformation $y = (t\cos t, \ t\sin t) + z_{SR}$, where $z_{SR} \sim \mathcal{N}(0, \sigma_{SR}^2 I)$ is isotropic Gaussian noise added in the target space. We use $\sigma_{SR} = 0.8$ in all experiments. This construction preserves the characteristic geometry of the Swiss-roll distribution while providing an analytically known

| Method Order | Unconditional | | Conditional | |
|---|---|---|---|---|
| | MMD $\downarrow$ $[10^{-3}]$ | $\mathcal{W}_2 \downarrow$ $[10^{-2}]$ | MMD $\downarrow$ $[10^{-3}]$ | $\mathcal{W}_2 \downarrow$ $[10^{-1}]$ |
| Regression | $18.35_{\pm 0.00}$ | $48.42_{\pm 0.00}$ | $88.91_{\pm 5.80}$ | $24.94_{\pm 0.86}$ |
| CGAN | $22.48_{\pm 0.39}$ | $96.24_{\pm 1.03}$ | $91.31_{\pm 8.30}$ | $25.55_{\pm 1.56}$ |
| UGAN+$\ell^2$ | $9.55_{\pm 0.01}$ | $30.20_{\pm 0.01}$ | $91.96_{\pm 8.25}$ | $25.76_{\pm 1.58}$ |
| CondNF | $20.59_{\pm 0.29}$ | $84.79_{\pm 0.61}$ | $85.34_{\pm 5.65}$ | $23.99_{\pm 1.05}$ |
| CondNF (SS) | $14.77_{\pm 0.16}$ | $69.69_{\pm 0.28}$ | $84.94_{\pm 5.22}$ | $23.87_{\pm 0.88}$ |
| OTCS | $9.13_{\pm 0.06}$ | $54.11_{\pm 0.25}$ | $\mathbf{41.70}_{\pm 1.52}$ | $19.57_{\pm 0.82}$ |
| FSBM | $30.88_{\pm 0.16}$ | $126.17_{\pm 0.47}$ | $43.63_{\pm 2.91}$ | $\mathbf{17.68}_{\pm 0.71}$ |
| EBiEOT-GMM (Ours) | $\mathbf{0.03}_{\pm 0.04}$ | $\mathbf{0.36}_{\pm 0.16}$ | $68.84_{\pm 1.59}$ | $22.08_{\pm 1.01}$ |

*Table 6.* Comparison of methods on the new 128 dataset using MMD and $\mathcal{W}_2$ metrics for unconditional and conditional settings for setup described in Appendix C.1.4.

conditional ground-truth transport plan $\pi^\star(\cdot|x)$. We use the same data setup as in the main text (§5.1), with $P = 128$ paired samples and $Q = R = 1024$ unpaired samples.

**Metrics.** For evaluation, we computed MMD (Gretton et al., 2012) and Sinkhorn divergence (Feydy et al., 2019) with regularization parameter $\varepsilon = 0.001$, which serves as an approximation of $\mathcal{W}_2$; see Appendix D.2.

**Discussion.** The results for this conditional transport plan are reported in Table 6. All methods struggle to accurately reconstruct the true conditional mapping, although they achieve partial success in recovering the target marginal distribution.

We believe that the main difficulty arises from the structure of the mapping itself: all points with the same radius $\|x\|$ are collapsed onto a single point on the Swiss-roll segment. This many-to-one transformation makes reconstruction of the conditional relationship from paired samples particularly challenging.

## C.2. Semi-supervised Classification

To demonstrate that our framework extends beyond domain translation, we apply the proposed objective (15) to a semi-supervised classification problem, following the discrete formulation sketch outlined in Appendix B.2. The goal is to learn the conditional distribution $\pi^*(y|x)$ from a small set of labeled examples together with additional unlabeled inputs.

In the classification setting, the target variable $y$ takes values in a finite set of classes $\{y_1, \ldots, y_K\}$. Using the Gibbs-Boltzmann parameterization (12) together with the energy decomposition (14), we obtain the discrete conditional model

$$\pi^\theta(y_k|x) = \frac{1}{Z^\theta(x)} \exp\left(\frac{f_k^\theta - c^\theta(x, y_k)}{\varepsilon}\right), \qquad \log Z^\theta(x) = \log \sum_{k=1}^{K} \exp\left(\frac{f_k^\theta - c^\theta(x, y_k)}{\varepsilon}\right), \qquad (34)$$

Here, $f^\theta \in \mathbb{R}^K$ is a learnable vector of class potentials, and $c^\theta(x, y_k)$ is a neural-network-based cost function. Substituting this parameterization into the general objective (15), the continuous integral over $\mathcal{Y}$ becomes a finite sum over classes:

$$\mathcal{L}(\theta) = \frac{1}{\varepsilon} \mathbb{E}_{(x,y) \sim \pi^*} \left[c^\theta(x, y)\right] - \frac{1}{\varepsilon} \mathbb{E}_{y \sim \pi_y^*} \left[f^\theta(y)\right] + \mathbb{E}_{x \sim \pi_x^*} \left[\log Z^\theta(x)\right]. \qquad (35)$$

When the class prior is uniform, i.e. $\pi_y^*(y_k) = 1/K$, the second term simplifies to $-\frac{1}{K\varepsilon} \sum_{k=1}^{K} f_k^\theta$.

**Setup.** We conducted experiments on the MNIST dataset (Bottou et al., 1994), which contains ten classes corresponding to digits $0, \ldots, 9$. We performed an ablation study by varying the number of labeled samples $P$ and unlabeled samples $Q$. To obtain a balanced target marginal distribution $\pi_y^*$, we used the same number of labeled samples from each class.

**Parameterization.** We parameterize the joint cost $c(x, y)$ using embedding-based MLP and CNN architectures that evaluate all classes in a single forward pass. Each label $y$ is represented by a learned embedding, which is combined with features of $x$ to predict $c(x, y)$. The MLP uses flattened images and two 256-unit hidden layers, while the CNN uses two convolutional blocks followed by a 128-dimensional projection. In both cases, energies for all classes are computed

in parallel. Training uses the joint energies, a learnable class potential $f(y)$, and a log-sum-exp marginal over labels; predictions follow $p(y \mid x) \propto \exp((f(y) - c(x, y))/\varepsilon)$.

**Metrics.** We report classification accuracy. For each image $x$, we compute the Gibbs posterior over all classes, $p(y \mid x) \propto \exp((f(y) - c(x, y))/\varepsilon)$, and predict the label with the highest probability: $\hat{y} = \arg\max_y p(y \mid x)$. A prediction is counted as correct if $\hat{y}$ matches the ground-truth label. Accuracy is then computed as the fraction of correctly classified test images.

**Discussion.** The results are presented in Figure 9. The heatmap on test set show that adding unlabeled data generally improves accuracy, especially in the low-label regime ($P < 100$). This trend is consistent with the findings already reported in Table 1. At the same time, very large $Q$ can hurt final performance, suggesting overfitting or reduced stability in that regime. We also observe that $P = 200$ paired samples is already sufficient to learn a strong classifier, while the test heatmaps indicate that some reduction in paired data can be compensated by increasing the amount of unlabeled data.

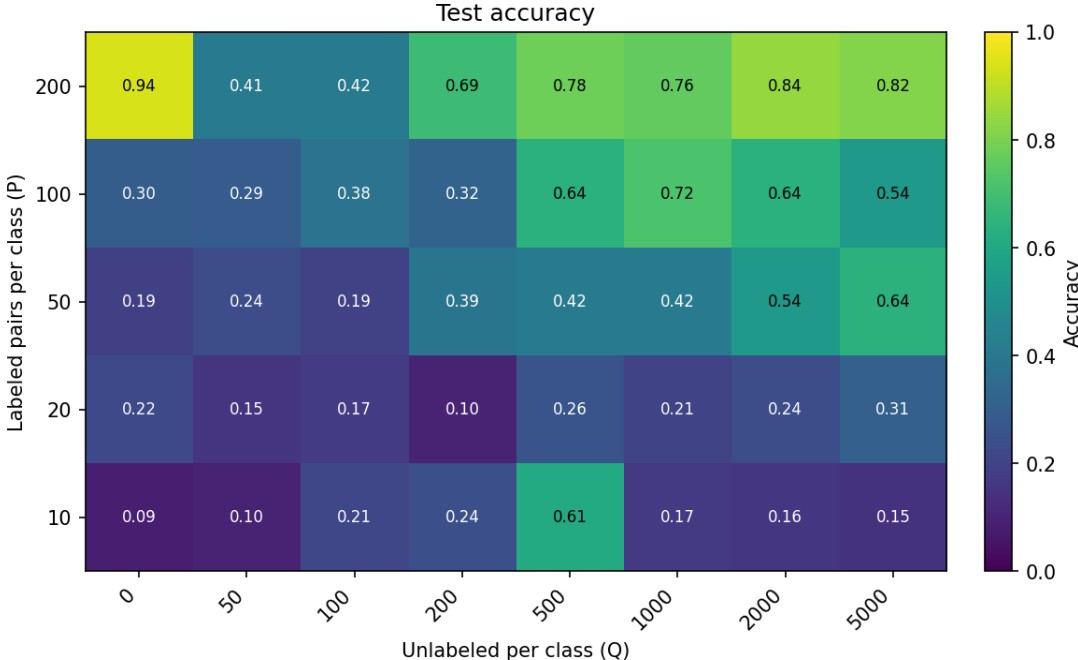

*Figure 9.* Heatmap for ablation of EBiEOT method for classification task described in AppendixC.2 for MNIST dataset.

# D. Experimental Details

### D.1. Baseline Details

This section details the loss functions employed by the baseline models, providing context and explanation for the data usage summarized in Table 7. Furthermore, it explains a straightforward adaptation of the log-likelihood loss function presented in (11) to accommodate unpaired data, offering a natural comparative approach to the method proposed in our work. Finally, it includes details about our reproduction of other methods and their discussion.

1. **Standard generative & predictive models:**

   - **Regression Model** (MLP) uses the following simple $\ell^2$ loss

     $$\min_\theta \mathbb{E}_{(x,y)\sim\pi^*} \|y - G_\theta(x)\|^2,$$

     where $G_\theta : \mathcal{X} \to \mathcal{Y}$ is a generator MLP with trainable parameters $\theta$. Clearly, such a model can use only paired data. Furthermore, it is known that the optimal regressor $G^*$ coincides with $\mathbb{E}_{y\sim\pi^*(\cdot|x)}y$, i.e., predicts the conditional expectation. Therefore, such a model will never learn the true data distribution unless all $\pi^*(\cdot|x)$ are degenerate.

- **Conditional GAN** uses the following $\min \max$ loss:

$$\min_\theta \max_\phi \left[ \underbrace{\mathbb{E}_{x,y \sim \pi^*} \log\left(D_\phi(y|x)\right)}_{\text{Joint, requires pairs } (x,y) \sim \pi^*} + \underbrace{\mathbb{E}_{x \sim \pi_x^*} \mathbb{E}_{z \sim p_z(z)} \log\left(1 - D_\phi(G_\theta(z|x)|x)\right)}_{\text{Marginal, requires } x \sim \pi_x^*} \right],$$

where $G_\theta : \mathcal{Z} \times \mathcal{X} \to \mathcal{Y}$ is the conditional generator with parameters $\theta$, $p_z$ is a distribution on latent space $\mathcal{Z}$, and $D_\phi : \mathcal{Y} \times \mathcal{X} \to (0,1)$ is the conditional discriminator with parameters $\phi$. It is clear that the model can use not only paired data during the training, but also samples from $\pi_x^*$. The minimum of this loss is achieved when $G_\theta(\cdot|x)$ generates $\pi^*(\cdot|x)$ from $p_z$.

- **Unconditional GAN + $\ell^2$ loss** optimizes the following $\min \max$ objective:

$$\min_\theta \max_\phi \left[ \lambda \underbrace{\mathbb{E}_{(x,y) \sim \pi^*} \mathbb{E}_{z \sim p_z} ||y - G_\theta(x,z)||^2}_{\text{Joint, requires pairs } (x,y) \sim \pi^*} + \underbrace{\mathbb{E}_{y \sim \pi_y^*} \log\left(D_\phi(y)\right)}_{\text{Marginal, requires } y \sim \pi_y^*} + \underbrace{\mathbb{E}_{x \sim \pi_x^*} \mathbb{E}_{z \sim p_z} \log\left(1 - D_\phi(G_\theta(x,z))\right)}_{\text{Marginal, requires } x \sim \pi_x^*} \right],$$

where $\lambda > 0$ is a hyperparameter and $G_\theta : \mathcal{X} \times \mathcal{Z} \to \mathcal{Y}$ is the stochastic generator. Compared to the unconditional case, the main idea here is to use the unconditional discriminator $D_\phi : \mathcal{Y} \to (0,1)$. This allows using unpaired samples from $\pi_y^*$. However, using only GAN loss would ignore the paired information in any form, this is why the supervised $\ell^2$ loss is added ($\lambda = 1$).

We note that this model has a trade-off between the target matching loss (GAN loss) and regression loss (which suffers from averaging). Hence, the model is unlikely to learn the true paired data distribution and can be considered as a heuristic loss for using both paired and unpaired data. Overall, we believe this baseline is representative of many existing GAN-based solutions (Tripathy et al., 2019, §3.5), (Jin et al., 2019, §3.3), (Yang & Chen, 2020, §C), (Vasluianu et al., 2021, §3), which use objectives that are *ideologically* similar to this one for paired and unpaired data.

- **Conditional Normalizing Flow** (Winkler et al., 2019) learns an explicit density model

$$\pi^\theta(y|x) = p_z(G_\theta^{-1}(y|x)) \left| \frac{\partial G_\theta^{-1}(y|x)}{\partial y} \right|$$

via optimizing log-likelihood (11) of the paired data. Here $G_\theta : \mathcal{Z} \times \mathcal{X} \to \mathcal{Y}$ is the conditional generator function. It is assumed that $\mathcal{Z} = \mathcal{Y}$ and $G_\theta(\cdot|x)$ is invertible and differentiable. In the implementation, we use the well-celebrated RealNVP neural architecture (Dinh et al., 2017). The optimal values are attained when the generator $G_\theta(\cdot|x)$ indeed generates $\pi^\theta(\cdot|x) = \pi^*(\cdot|x)$.

The conditional flow is expected to accurately capture the true conditional distributions, provided that the neural architecture is sufficiently expressive and there is an adequate amount of paired data available. However, as mentioned in §3.1, a significant challenge arises in integrating unpaired data into the learning process. For instance, approaches such as those proposed by (Atanov et al., 2019; Izmailov et al., 2020) aim to extend normalizing flows to a semi-supervised context. However, these methods primarily assume that the input conditions $x$ are discrete, making it difficult to directly apply their frameworks to our continuous case. For completeness, below we discuss a variant of the log-likelihood loss (Atanov et al., 2019, Eq. 1) when both $x, y$ are continuous.

2. **Semi-supervised log-likelihood** methods (Atanov et al., 2019; Izmailov et al., 2020):

- **Semi-supervised Conditional Normalizing Flows**. As noted by the authors, a natural strategy for log-likelihood semi-supervised training that leverages both paired and unpaired data is to optimize the following loss:

$$\max_\theta \left[ \underbrace{\mathbb{E}_{(x,y) \sim \pi^*} \log \pi^\theta(y|x)}_{\text{Joint, requires pairs } (x,y) \sim \pi^*} + \underbrace{\mathbb{E}_{y \sim \pi_y^*} \log \pi^\theta(y)}_{\text{Marginal, requires } y \sim \pi_y^*} \right]. \tag{36}$$

This straightforward approach involves adding the unpaired data component, $\mathbb{E}_{y \sim \pi_y^*} \log \pi^\theta(y)$ to the loss function alongside the standard paired data component (11). While loss (36) looks natural, its optimization is *highly non-trivial* since the marginal log-likelihood $\log \pi^\theta(y)$ is not directly available. In fact, (Atanov et al., 2019; Izmailov et al., 2020) use this loss exclusively in the case when $x$ is a discrete object, e.g., the class label $x \in \{1, 2, ..., K\}$. In this case $\log \pi^\theta(y)$ can be analytically computed:

$$\log \pi^\theta(y) = \log \mathbb{E}_{x \sim \pi_x^*} \pi^\theta(y|x) = \log \sum_{k=1}^{K} \pi^\theta(y|x=k) \pi_x^*(x=k),$$

and $\pi^*(x = k)$ are known class probabilities. Unfortunately, in the continuous case $\pi_x^*(x)$ is typically not available explicitly, and one has to exploit *approximations* such as

$$\log \pi^\theta(y) = \log \mathbb{E}_{x \sim \pi_x^*} \pi^\theta(y|x) \approx \log \frac{1}{Q} \sum_{q=1}^{Q} \pi^\theta(y|x_q),$$

where $x_q$ are train (unpaired) samples. However, such Monte-Carlo estimates are generally **biased** (because of the logarithm) and do not lead to good results, especially in high dimensions. Nevertheless, for completeness, we also test how this approach performs. In our 2D example (Figure 3j), we found there is no significant difference between this loss and the fully supervised loss (11): both models incorrectly map to the target and fail to learn conditional distributions.

- **Semi-supervised Conditional Gaussian Mixture Model**. Using the natural loss (36) for semi-supervised learning, one could also consider a (conditional) Gaussian mixture parameterization for $\pi^\theta(y|x)$ instead of the normalizing flow. For completeness, we include this baseline for comparison. Using the same Gaussian mixture parameterization (19) as in our method, we observed that this loss quickly overfits and leads to degenerate solutions, see Figure 3e.

3. **Semi-supervised Methods.** These methods are designed to learn deterministic OT maps with general cost functions and, as a result, cannot capture stochastic conditional distributions.

  - **Neural optimal transport with pair-guided cost functional** (Asadulaev et al., 2024, GNOT). This method employs a general cost function for the neural optimal transport approach, utilizing a neural network parameterization for the mapping function and potentials. In our experiments, we focus on the paired cost function setup, enabling the use of both paired and unpaired data. We use the publicly available implementation[3], which has been verified through toy experiments provided in the repository.
  - **Differentiable cost-parameterized entropic mapping estimator** (Howard et al., 2024, DCPEME). We obtained the implementation from the authors but were unable to achieve satisfactory performance. This is likely due to the deterministic map produced by their method based on the entropic map estimator from (Cuturi et al., 2023). In particular, scenarios where nearby or identical points are mapped to distant locations may introduce difficulties, potentially leading to optimization stagnation during training.
  - **Parametric Pushforward Estimation With Map Constraints** (Panda et al., 2023, parOT)[4]. We evaluated this method using the $\ell_2$ cost function, where it performed as expected. However, on our setup, the method proved unsuitable because it learns a fully deterministic transport map, which lacks the flexibility needed to model stochastic multimodal mapping. This limitation is visually evident in Figure 6m.
  - **Optimal Transport-guided Conditional Score-based diffusion model** (Gu et al., 2023, OTCS). We evaluated this method on a two-dimensional example from their GitHub repository[5], where it performed as expected. However, when applied to our setup (described in §5.1), the method failed to yield satisfactory results, even when provided with a large amount of training data (refer to Figure 6n and detailed in Appendix C.1.1).
  - **Feedback Schrödinger Bridge Matching** (Theodoropoulos et al., 2024, FSBM). We first tested the method on a two-dimensional example from their GitHub repository[6], where it performed as reported in the original paper. However, as shown in Figure 3o, the learned target distribution is very noisy with a small amount of data. With more samples (Figure 6o), the method approximates the target distribution better but still fails to capture the ground-truth conditional distribution, presumably due to misleading guidance from the key-points.

### D.2. Metric Computation Details

In this section, we describe the evaluation metrics used in our experiments.

**Sinkhorn divergence.** The *Wasserstein-2 distance* is defined as

$$\mathcal{W}_2(\alpha, \beta) \stackrel{\text{def}}{=} \left( \min_{\pi \in \Pi(\alpha, \beta)} \mathbb{E}_{x,y \sim \pi} \|x - y\|_2^2 \right)^{\frac{1}{2}}, \tag{$\mathcal{W}_2$}$$

---

[3] https://github.com/machinestein/GNOT
[4] https://github.com/natalieklein229/uq4ml/tree/parot
[5] https://github.com/XJTU-XGU/OTCS/
[6] https://github.com/panostheo98/FSBM

i.e., the square root of the OT problem (OT) with quadratic cost $c^*(x, y) = \|x - y\|_2^2$. Computing OT exactly is computationally expensive: the complexity of solving the OT problem is $O(s^3 \log s)$ for sample size $s$ (Cuturi, 2013). Moreover, OT suffers from the curse of dimensionality. In particular, for empirical measures

$$\hat{\alpha}_s = \frac{1}{s} \sum_{i=1}^{s} \delta_{X_i}, \qquad \hat{\beta}_s = \frac{1}{s} \sum_{i=1}^{s} \delta_{Y_i},$$

the estimation error scales as (Weed & Bach, 2019)

$$\mathbb{E} \left| \mathrm{OT}_{c^*}(\alpha, \beta) - \mathrm{OT}_{c^*}(\hat{\alpha}_s, \hat{\beta}_s) \right| = O(s^{-1/D}), \tag{37}$$

where $D$ is the data dimension. To address these issues, in practice it's common to use entropic regularized OT (1), discussed in detail in Appendix B.1. This formulation reduces the computational complexity to $O(s^2)$ (up to polylogarithmic terms) (Altschuler et al., 2017; Dvurechensky et al., 2018) and achieves improved sample complexity (Niles-Weed & Rigollet, 2022; Groppe & Hundrieser, 2024):

$$\mathbb{E} \left| \mathrm{EOT}_{c^*, \varepsilon}(\alpha, \beta) - \mathrm{EOT}_{c^*, \varepsilon}(\hat{\alpha}_s, \hat{\beta}_s) \right| \lesssim \frac{1}{\sqrt{s}}, \tag{38}$$

where $\lesssim$ denotes inequality up to a constant independent of $s$. However, entropic OT is biased, since $\mathrm{EOT}_{c^*, \varepsilon}(\alpha, \alpha) \neq 0$. To remove this bias, Feydy et al. (2019) introduced the *Sinkhorn divergence*:

$$SD_{c^*, \varepsilon}(\alpha, \beta) \stackrel{\mathrm{def}}{=} \mathrm{EOT}_{c^*, \varepsilon}(\alpha, \beta) - \frac{1}{2} \mathrm{EOT}_{c^*, \varepsilon}(\alpha, \alpha) - \frac{1}{2} \mathrm{EOT}_{c^*, \varepsilon}(\beta, \beta). \tag{39}$$

Moreover, as $\varepsilon \to 0$, the Sinkhorn divergence converges to the OT distance (Feydy et al., 2019). In our experiments, we approximate $\mathcal{W}_2$ using Sinkhorn divergence with $\varepsilon = 0.001$, computed with the `GeomLoss` library (Feydy et al., 2019).

**MMD.** We compute the *Maximum Mean Discrepancy* (MMD) (Gretton et al., 2012) using its kernel formulation. Let $K : \mathcal{X} \times \mathcal{X} \to \mathbb{R}$ be a positive-definite kernel with associated reproducing kernel Hilbert space (RKHS) $\mathcal{H}$ and feature map $\phi$. The squared MMD between distributions $\rho_k$ and $\hat{\rho}_k$ is defined as

$$\mathrm{MMD}_K^2(\rho_k, \hat{\rho}_k) = \mathbb{E}_{x, x' \sim \rho_k} K(x, x') - 2\mathbb{E}_{x \sim \rho_k, \, y \sim \hat{\rho}_k} K(x, y) + \mathbb{E}_{y, y' \sim \hat{\rho}_k} K(y, y'). \tag{40}$$

Importantly, MMD can be computed directly through kernel evaluations without explicitly constructing feature maps. Given samples $\{x_i\}_{i=1}^{N} \sim \rho_k$ and $\{y_j\}_{j=1}^{M} \sim \hat{\rho}_k$, we use the unbiased estimator

$$\widehat{\mathrm{MMD}}_K^2 = \frac{1}{N(N-1)} \sum_{i \neq i'} K(x_i, x_{i'}) - \frac{2}{NM} \sum_{i=1}^{N} \sum_{j=1}^{M} K(x_i, y_j) + \frac{1}{M(M-1)} \sum_{j \neq j'} K(y_j, y_{j'}). \tag{41}$$

The estimator has quadratic complexity in the number of samples. In all experiments, we use a uniform mixture of five Gaussian (RBF) kernels,

$$K(x, y) = \frac{1}{5} \sum_{r=1}^{5} \exp\left( -\frac{\|x - y\|^2}{2\sigma_r^2} \right). \tag{42}$$

The bandwidths $\{\sigma_r\}_{r=1}^{5}$ are chosen on a geometric grid with ratio 2. First, we compute the median pairwise Euclidean distance over the pooled test samples $\mathrm{X}_{\text{test}} \cup \mathrm{Y}_{\text{test}}$ (the median heuristic). This value defines the central bandwidth. The remaining bandwidths are obtained by scaling it by factors $\{2^{-2}, 2^{-1}, 1, 2, 2^2\}$.

**Estimation Protocol.** As discussed above, empirical Wasserstein distances have poor sample complexity and therefore typically require a large number of samples for accurate estimation. However, in the two-dimensional setting this issue is less pronounced. We report both unconditional and conditional metrics. The unconditional metric evaluates how accurately a method reconstructs the target marginal distribution. For this evaluation, we use 1024 generated samples. The conditional metric evaluates how accurately the conditional transport plan is recovered. In this setting, we use 30 initial points and 1024 corresponding target samples.

## D.3. General Implementation Details

**Parameterization.** The depth and number of hidden layers vary depending on the experiment.

For $f^\theta$ (18) we represent:

- $w_n$ as $\log w_n$,
- $b_n$ directly as a vector,
- the matrix $B_n$ in diagonal form, with $\log(B_n)_{i,i}$ on its diagonal. This choice not only reduces the number of learnable parameters in $\theta_f$ but also enables efficient computation of $B_n^{-1}$ with a time complexity of $\mathcal{O}(D_y)$.

For $c^\theta$ (17), we represent:

- $v_m(x)$ as a multilayer perceptron (MLP) with ReLU activations (Agarap, 2018) and a LogSoftMax output layer,
- $a_m(x)$ as an MLP with ReLU activations.

**Optimizers.** We employ two separate Adam optimizers (Kingma, 2014) with different step sizes for paired and unpaired data to enhance convergence.

**Initialization.**

- $\log w_n$ as $\log \frac{1}{n}$,
- $b_n$ using random samples from $\pi_y^*$,
- $\log(B_n)_{j,j}$ with $\log(0.1)$,
- for the neural networks, we use the default PyTorch initialization (Ansel et al., 2024),
- $\varepsilon = 1$ for all experiments, since the solver is independent of $\varepsilon$, as discussed in §2.2.

## D.4. Gaussian to Swiss Roll Mapping Details

**Generation process.** To create the ground truth plan $\pi^*$, we utilize the following procedure: sample a mini-batch of size 64 and then determine the optimal mapping using the entropic Sinkhorn algorithm, as outlined in (Cuturi, 2013) and implemented in (Flamary et al., 2021). This process is repeated $P$ times to generate the required number of pairs.

**Cost Matrix.** Let $x \in \mathbb{R}^2$ and $y \in \mathbb{R}^2$ be points from the source and target distributions, respectively. Define the rotated vectors as

$$y^{\pm\varphi} = R_{\pm\varphi}(y) = \begin{bmatrix} \cos(\pm\varphi) & -\sin(\pm\varphi) \\ \sin(\pm\varphi) & \cos(\pm\varphi) \end{bmatrix} \begin{bmatrix} y_1 \\ y_2 \end{bmatrix},$$

where $\varphi$ is a given rotation angle, in our case, it's $\pm 90°$. The corresponding elements of mini-batch OT cost matrices are then

$$C_{ij}^{+\varphi} = \|x_i - y_j^{+\varphi}\|_2, \quad C_{ij}^{-\varphi} = \|x_i - y_j^{-\varphi}\|_2,$$

and the final cost matrix is

$$C_{ij} = \min(C_{ij}^{+\varphi}, C_{ij}^{-\varphi}), \quad \forall i, j.$$

| Method | Paired $(x,y) \sim \pi^*$ | Unpaired $x \sim \pi_x^*$ | Unpaired $y \sim \pi_y^*$ |
|---|:---:|:---:|:---:|
| Regression | ✓ | ✗ | ✗ |
| UGAN + $\ell^2$ | ✓ | ✓ | ✓ |
| CGAN | ✓ | ✓ | ✗ |
| CondNF | ✓ | ✗ | ✗ |
| CondNF (SS) | ✓ | ✓ | ✓ |
| GNOT | ✓ | ✓ | ✓ |
| DCPEME | ✓ | ✓ | ✓ |
| parOT | ✓ | ✓ | ✓ |
| OTCS | ✓ | ✓ | ✓ |
| FSBM | ✓ | ✓ | ✓ |
| CGMM (SS) | ✓ | ✓ | ✓ |
| **Our method** | ✓ | ✓ | ✓ |

*Table 7.* The ability to use paired/unpaired data by various models.

In other words, each $x_i \sim \pi_x^*$ is mapped to a point $y_j$ on the opposite side of the Swiss Roll, rotated either by $+90°$ or $-90°$, depending on which distance is smaller.

**Implementation Details.** We choose the parameters as follows: $N = 50$, $M = 25$, with learning rates $lr_{\text{paired}} = 3 \times 10^{-4}$ and $lr_{\text{unpaired}} = 0.001$. We utilize a two-layer MLP network for the function $a_m(x)$ and a single-layer MLP for $v_m(x)$. The experiments are executed in parallel on a 2080 Ti GPU for a total of 25,000 iterations, taking approximately 20 minutes to complete.

### D.5. Image Translation via ALAE Details

**Implementation Details.** We largely follow the setup in Appendix D.4, setting $N = 10$, $M = 1$, and using 10K optimization steps. Our method employs a single-layer MLP to predict the parameters of a mixture of 10 Gaussians.

### D.6. Weather Prediction Details

We select two distinct months from the dataset (Malinin et al., 2021; Rubachev et al., 2025) and translate the meteorological features from the source month (January) to the target month (June). To operate at the monthly scale, we represent a source data point $x \in \mathbb{R}^{188}$ as the mean and standard deviation of the features collected at a specific location over the source month. The targets $y \in \mathbb{R}^{94}$ correspond to individual measurements in the target month.

Pairs are constructed by aligning a source data point with the target measurements at the same location. Consequently, multiple target data points $y$ may correspond to a single source point $x$ and represent samples from conditional distributions $\pi^*(y|x)$. The measurements from non-aligned locations are treated as unpaired. Such unpaired data naturally arise because stations may not provide reliable measurements in both months, for example, due to maintenance, sensor failures, extreme weather, or connectivity issues.

We obtain 500 unpaired and 192 paired data samples. For testing, 100 pairs are randomly selected.

**Implementation Details.** In general, we consider the same setting as in D.4. Specifically, we set $N = 10$, $M = 1$ and the number of optimization steps to $30,000$. The baseline uses an MLP network with the same number of parameters, predicting the parameters of a mixture of 10 Gaussians.

**Extremely Low-Data Regimes Discussion.** As is clear from Table 1, our method diverges when trained on very few samples (e.g., 5 paired and no unpaired). This is not surprising given the high dimensionality of the data ($D = 94$) and the number of learnable parameters ($|\theta| = 2668$). In such low-data regimes, the model likely overfits the cost function $c^\theta$ to the small paired dataset, which can cause instability. This issue could potentially be alleviated by simplifying the model, for instance by using a shallow or even linear parameterization of $c^\theta$ (Andrade et al., 2025). However, for consistency and fairness, we kept the architecture fixed across all experiments in the table.

## E. Proofs

### E.1. Loss Derivation

Below, we present a step-by-step derivation of the mathematical transitions, allowing the reader to follow and verify the validity of our approach. We denote as $C_1, C_2$ all terms that are not involved in learning the conditional plan $\pi^\theta(y|x)$, i.e., not dependent on $\theta$ or marginal distributions such as $\pi_x^*$. Starting from (6), we deduce

$$\text{KL}\left(\pi^*\|\pi^\theta\right) = \mathbb{E}_{x,y\sim\pi^*} \log \frac{\pi_x^*(x)\pi^*(y|x)}{\pi_x^\theta(x)\pi^\theta(y|x)} = \mathbb{E}_{x\sim\pi_x^*} \log \frac{\pi_x^*(x)}{\pi_x^\theta(x)} + \mathbb{E}_{x,y\sim\pi^*} \log \frac{\pi^*(y|x)}{\pi^\theta(y|x)} =$$

$$\text{KL}\left(\pi_x^*\|\pi_x^\theta\right) + \mathbb{E}_{x\sim\pi_x^*}\mathbb{E}_{y\sim\pi^*(\cdot|x)} \log \frac{\pi^*(y|x)}{\pi^\theta(y|x)} = \underbrace{\text{KL}\left(\pi_x^*\|\pi_x^\theta\right)}_{\text{Marginal}} + \underbrace{\mathbb{E}_{x\sim\pi_x^*}\text{KL}\left(\pi^*(\cdot|x)\|\pi^\theta(\cdot|x)\right)}_{\text{Conditional}} =$$

$$C_1 + \mathbb{E}_{x\sim\pi_x^*}\mathbb{E}_{y\sim\pi^*(\cdot|x)} \log \frac{\pi^*(y|x)}{\pi^\theta(y|x)} = C + \mathbb{E}_{x\sim\pi_x^*}\mathbb{E}_{y\sim\pi^*(\cdot|x)} \left[\log \pi^*(y|x) - \log \pi^\theta(y|x)\right] =$$

$$C_1 - \mathbb{E}_{x\sim\pi_x^*}\text{H}\left(\pi^*(\cdot|x)\right) - \mathbb{E}_{x,y\sim\pi^*} \log \pi^\theta(y|x) = C_2 - \mathbb{E}_{x,y\sim\pi^*} \log \pi^\theta(y|x) \overset{(12)}{=}$$

$$C_2 - \mathbb{E}_{x,y\sim\pi^*} \log \frac{\exp\left(-E^\theta(y|x)\right)}{Z^\theta(x)} = C_2 + \mathbb{E}_{x,y\sim\pi^*} E^\theta(y|x) + \mathbb{E}_{x,y\sim\pi^*} \log Z^\theta(x) \overset{(14)}{=}$$

$$C_2 + \mathbb{E}_{x,y\sim\pi^*} \frac{c^\theta(x,y) - f^\theta(y)}{\varepsilon} + \mathbb{E}_{x,y\sim\pi^*} \log Z^\theta(x) =$$

$$C_2 + \varepsilon^{-1}\mathbb{E}_{x,y\sim\pi^*}[c^\theta(x,y)] - \varepsilon^{-1}\mathbb{E}_{x,y\sim\pi^*} f^\theta(y) + \mathbb{E}_{x,y\sim\pi^*} \log Z^\theta(x) =$$

$$C_2 + \varepsilon^{-1}\mathbb{E}_{x,y\sim\pi^*}[c^\theta(x,y)] - \varepsilon^{-1}\mathbb{E}_{y\sim\pi_y^*}\mathbb{E}_{x\sim\pi^*(\cdot|y)} f^\theta(y) + \mathbb{E}_{x\sim\pi_x^*}\mathbb{E}_{y\sim\pi^*(\cdot|x)} \log Z^\theta(x) =$$

$$C_2 + \varepsilon^{-1}\mathbb{E}_{x,y\sim\pi^*}[c^\theta(x,y)] - \varepsilon^{-1}\mathbb{E}_{y\sim\pi_y^*} f^\theta(y) \underbrace{\mathbb{E}_{x\sim\pi^*(\cdot|y)}1}_{=1} + \mathbb{E}_{x\sim\pi_x^*} \log Z^\theta(x) \underbrace{\mathbb{E}_{y\sim\pi^*(\cdot|x)}1}_{=1} =$$

$$C_2 + \varepsilon^{-1}\mathbb{E}_{x,y\sim\pi^*}[c^\theta(x,y)] - \varepsilon^{-1}\mathbb{E}_{y\sim\pi_y^*}f^\theta(y) + \mathbb{E}_{x\sim\pi_x^*}\log Z^\theta(x).$$

The mathematical derivation presented above demonstrates that our defined loss function (15) is essentially a framework for minimizing KL-divergence. In other words, when the loss (15) equals to $-C_2$, it implies that we have successfully recovered the true conditional plan $\pi^*$ in the KL sense.

### E.2. Expressions for the Gaussian Parameterization

*Proof of Proposition 3.1.* Our parameterization of the cost $c^\theta$ (17) and the dual potential $f^\theta$ (18) gives:

$$\exp\left(\frac{f^\theta(y) - c^\theta(x,y)}{\varepsilon}\right) = \exp\left(\log\sum_{n=1}^N w_n\mathcal{N}(y\,|\,b_n, \varepsilon B_n) + \log\sum_{m=1}^M v_m(x)\exp\left(\frac{\langle a_m(x), y\rangle}{\varepsilon}\right)\right)$$

$$= \sum_{m=1}^M\sum_{n=1}^N \frac{v_m(x)w_n}{\sqrt{\det\left(2\pi\varepsilon^{-1}B_n^{-1}\right)}}\exp\left(-\frac{1}{2}(y-b_n)^\top\frac{B_n^{-1}}{\varepsilon}(y-b_n) + \frac{\langle a_m(x), y\rangle}{\varepsilon}\right)$$

We now rewrite the expression inside the exponent, scaled by $-2\varepsilon$, using the symmetry of $B_n$, to cast it into a Gaussian mixture form:

$$(y-b_n)^\top B_n^{-1}(y-b_n) - 2\langle a_m(x), y\rangle = y^\top B_n^{-1}y - 2b_n^\top B_n^{-1}y + b_n^\top B_n^{-1}b_n - 2\langle a_m(x), y\rangle =$$

$$y^\top B_n^{-1}y - 2\underbrace{(b_n + B_n a_m(x))^\top}_{\overset{\text{def}}{=}d_{mn}^\top(x)}B_n^{-1}y + b_n^\top B_n^{-1}b_n =$$

$$(y - d_{mn}(x))^\top B_n^{-1}(y - d_{mn}(x)) + b_n^\top B_n^{-1}b_n - d_{mn}^\top(x)B_n^{-1}d_{mn}(x).$$

Afterwards, we rewrite the last two terms:

$$b_n^\top B_n^{-1}b_n - d_{mn}^\top(x)B_n^{-1}d_{mn}(x) = b_n^\top B_n^{-1}b_n - (b_n + B_n a_m(x))^\top B_n^{-1}(b_n + B_n a_m(x)) =$$

$$\underbrace{b_n^\top B_n^{-1}b_n - b_n^\top B_n^{-1}b_n}_{=0} - b_n^\top\underbrace{B_n^{-1}B_n}_{=I}a_m(x) - a_m^\top(x)\underbrace{B_n B_n^{-1}}_{=I}b_n - a_m^\top(x)\underbrace{B_n B_n^{-1}}_{=I}B_n a_m(x) =$$

$$-a_m^\top(x)B_n a_m(x) - 2b_n^\top a_m(x).$$

Finally, we get

$$\exp\left(\frac{f^\theta(y) - c^\theta(x,y)}{\varepsilon}\right) = \sum_{m=1}^M\sum_{n=1}^N w_n v_m(x)\underbrace{\exp\left(\frac{a_m^\top(x)B_n a_m(x) + 2b_n^\top a_m(x)}{2\varepsilon}\right)}_{\overset{\text{def}}{=}z_{mn}(x)}$$

$$\cdot\underbrace{\frac{1}{\sqrt{\det\left(2\pi\varepsilon^{-1}B_n^{-1}\right)}}\exp\left(-\frac{1}{2}(y - d_{mn}(x))^\top\frac{B_n^{-1}}{\varepsilon}(y - d_{mn}(x))\right)}_{=\mathcal{N}(y\,|\,d_{mn}(x), \varepsilon B_n)},$$

and, since $\int_{\mathcal{Y}}\mathcal{N}(y\,|\,d_{mn}(x), \varepsilon B_n)\mathrm{d}y = 1$, the normalization constant simplifies to the sum of $z_{mn}(x)$:

$$Z^\theta(x) = \int_{\mathcal{Y}}\exp\left(\frac{f^\theta(y) - c^\theta(x,y)}{\varepsilon}\right)\mathrm{d}y$$

$$= \int_{\mathcal{Y}}\sum_{m=1}^M\sum_{n=1}^N z_{mn}(x)\mathcal{N}(y\,|\,d_{mn}(x), \varepsilon B_n)\mathrm{d}y = \sum_{m=1}^M\sum_{n=1}^N z_{mn}(x).$$

$\square$

*Proof of Proposition 3.2.* Combining equations (12), (14) and derivation above, we seamlessly obtain the expression (19) needed for Proposition 3.2. $\square$

### E.3. Gradient of our Loss for Energy-Based Modeling

*Proof of Proposition A.1.* Direct differentiation of (15) gives:

$$\frac{\partial}{\partial\theta}\mathcal{L}(\theta) = \varepsilon^{-1}\mathbb{E}_{x,y\sim\pi^*}\left[\frac{\partial}{\partial\theta}c^\theta(x,y)\right] - \varepsilon^{-1}\mathbb{E}_{y\sim\pi_y^*}\left[\frac{\partial}{\partial\theta}f^\theta(y)\right] + \mathbb{E}_{x\sim\pi_x^*}\left[\frac{\partial}{\partial\theta}\log Z^\theta(x)\right]. \tag{43}$$

Recalling expression for the normalization constant, the last term can be expressed as follows:

$$\mathbb{E}_{x\sim\pi_x^*}\left[\frac{1}{Z^\theta(x)}\frac{\partial}{\partial\theta}Z^\theta(x)\right] = \mathbb{E}_{x\sim\pi_x^*}\left[\frac{1}{Z^\theta(x)}\int_{\mathcal{Y}}\frac{\partial}{\partial\theta}\exp\left(\frac{f^\theta(y)-c^\theta(x,y)}{\varepsilon}\right)\mathrm{d}y\right] =$$

$$\mathbb{E}_{x\sim\pi_x^*}\left[\frac{1}{Z^\theta(x)}\int_{\mathcal{Y}}\frac{\frac{\partial}{\partial\theta}\left(f^\theta(y)-c^\theta(x,y)\right)}{\varepsilon}\exp\left(\frac{f^\theta(y)-c^\theta(x,y)}{\varepsilon}\right)\mathrm{d}y\right] =$$

$$\varepsilon^{-1}\mathbb{E}_{x\sim\pi_x^*}\left[\int_{\mathcal{Y}}\frac{\partial}{\partial\theta}\left(f^\theta(y)-c^\theta(x,y)\right)\underbrace{\left\{\frac{1}{Z^\theta(x)}\exp\left(\frac{f^\theta(y)-c^\theta(x,y)}{\varepsilon}\right)\right\}}_{\pi^\theta(y|x)}\mathrm{d}y\right].$$

From equation above we obtain:

$$\frac{\partial}{\partial\theta}\mathcal{L}(\theta) = \varepsilon^{-1}\Bigg\{\mathbb{E}_{x,y\sim\pi^*}\left[\frac{\partial}{\partial\theta}c^\theta(x,y)\right] - \mathbb{E}_{y\sim\pi_y^*}\left[\frac{\partial}{\partial\theta}f^\theta(y)\right]$$

$$+ \mathbb{E}_{x\sim\pi_x^*}\mathbb{E}_{y\sim\pi^\theta(y|x)}\left[\frac{\partial}{\partial\theta}\left(f^\theta(y)-c^\theta(x,y)\right)\right]\Bigg\},$$

which concludes the proof. $\square$

### E.4. Universal Approximation

Our objective is to set up and use the very general universal approximation result in (Acciaio et al., 2024, Theorem 3.8). Hereinafter, we use the following notation that slightly abuse notation from the main text.

**Intra-Section Notation.** For any $D \in \mathbb{N}$ we denote the Lebesgue measure on $\mathbb{R}^D$ by $\lambda_D$, suppressing the subscript $D$ whenever clear from its context, we use $L_+^1(\mathbb{R}^D)$ to denote the set of Lebesgue integrable (equivalence class of) functions $f : \mathbb{R}^D \to \mathbb{R}$ for which $\int f(x)\,\lambda(dx) = 1$ and $f \geq 0$ $\lambda$-a.e; i.e. Lebesgue-densities of probability measures. We use $\mathcal{P}_1^+(\mathbb{R}^D)$ to denote the space of all Borel probability measures on $\mathbb{R}^D$ which are absolutely continuous with respect to $\lambda$, metrized by the total variation distance $d_{TV}$. For any $D \in \mathbb{N}$, we denote the set of $D \times D$ positive-definite matrices by $\mathrm{PD}_D$. Additionally, for any $N \in \mathbb{N}$, we define the $N$-simplex by $\Delta_N \stackrel{\text{def.}}{=} \{u \in [0,1]^N : \sum_{n=1}^N u_n = 1\}$. We also denote floor operation for any $x \in \mathbb{R}$ as $\lfloor x \rfloor \stackrel{\text{def.}}{=} \max\{n \in \mathbb{Z} | n \leq x\}$.

**Lemma E.1** (The Space $(\mathcal{P}_1^+(\mathbb{R}^D), d_{TV})$ is quantizable by Gaussian Mixtures). *For every $N \in \mathbb{N}$, let $D_N \stackrel{\text{def.}}{=} \frac{N}{2}((D^2 + 3D + 2))$ and define the map*

$$GMM_N : \mathbb{R}^{D_N} = \mathbb{R}^N \times \mathbb{R}^{ND} \times \mathbb{R}^{\frac{N}{2}D(D+1)} \to \mathcal{P}_1^+(\mathbb{R}^D)$$

$$\left(w, \{b_n\}_{n=1}^N, \{B_n\}_{n=1}^N\right) \mapsto \sum_{n=1}^N Proj_{\Delta_N}(w)_n\,\nu\big(b_n, \varphi(B_n)\big),$$

*where $Proj_{\Delta_N} : \mathbb{R}^N \mapsto \Delta_N$ is the $\ell^2$ orthogonal projection of $\mathbb{R}^N$ onto the $N$-simplex $\Delta_N$ and $\nu(b_n, \varphi(B_n))$ is the Gaussian measure on $\mathbb{R}^D$ with mean $b_n$, and non-singular covariance matrix given by $\varphi(B_n)$ where $\varphi : \mathbb{R}^{D(D+1)/2} \to \mathrm{PD}_D$ is given for each $B \in \mathbb{R}^{D(D+1)/2}$ by*

$$\varphi(B) \stackrel{\text{def.}}{=} \exp\left(\begin{pmatrix} B_1 & B_2 & \dots & B_D \\ B_2 & B_3 & \dots & B_{2D-1} \\ \vdots & \ddots & & \vdots \\ B_D & B_{2D-1} & \dots & B_{D(D+1)/2} \end{pmatrix}\right), \tag{44}$$

*where* exp *is the matrix exponential on the space of* $D \times D$ *matrices. Then, the family* $(GMM_n)_{n=1}^{\infty}$ *is a quantization of* $(P_1^+(\mathbb{R}^D), d_{TV})$ *in the sense of (Acciaio et al., 2024, Definition 3.2).*

*Proof.* As implied by (Arabpour et al., 2024, Equation (3.10) in Proposition 7) every Gaussian measure $\mathcal{N}(m, \Sigma) := \mu$ on $\mathbb{R}^D$ with mean $m \in \mathbb{R}^D$, symmetric positive-definite covariance matrix $\Sigma$ can be represented as

$$\mu = \mathcal{N}(m, \varphi(X)) \tag{45}$$

for some (unique) vector $X \in \mathbb{R}^{D(D+1)/2}$. Therefore, by definition of a quantization, see (Acciaio et al., 2024, Definition 3.2), it suffices to show that the family of Gaussian mixtures is dense in $(\mathcal{P}_1^+(\mathbb{R}^D), d_{TV})$.

Now, let $\nu \in \mathcal{P}_1^+(\mathbb{R}^D)$ be arbitrary. By definition of $\mathcal{P}_1^+(\mathbb{R}^D)$ the measure $\nu$ admits a Radon-Nikodym derivative $f \stackrel{\text{def.}}{=} \frac{D\mu}{D\lambda}$, with respect to the $D$-dimensional Lebesgue measure $\lambda$. Moreover, by the Radon-Nikodym theorem, $f \in L_\mu^1(\mathbb{R}^D)$; and by since $\mu$ is a probability measure then $\nu \in L_+^1(\mathbb{R}^D)$.

Since compactly-supported smooth functions are dense in $L_+^1(\mathbb{R}^D)$ then, for every $\varepsilon > 0$, there exists some $\tilde{f} \in C_c^\infty(\mathbb{R}^D)$ with $\tilde{f} \geq 0$ such that

$$\|f - \tilde{f}\|_{L^1(\mathbb{R}^D)} < \frac{\varepsilon}{3}. \tag{46}$$

Since $C_c^\infty(\mathbb{R}^D)$ is dense in $L^1(\mathbb{R}^D)$ then we may without loss of generality re-normalize $\tilde{f}$ to ensure that it integrates to 1.

Since $\tilde{f}$ is compactly supported and approximates $f$, then (if $f$ is non-zero, which it cannot be as it integrates to 1) then it cannot be analytic, and thus it is non-polynomial. For every $\delta > 0$, let $\varphi_\delta$ denote the density of the $D$-dimensional Gaussian probability measure with mean 0 and isotropic covariance $\delta I_D$ (where $I_D$ is the $D \times D$ identity matrix). Therefore, the proof of (Pinkus, 1999, Proposition 3.7) (or any standard mollification argument) shows that we can pick $\delta \stackrel{\text{def.}}{=} \delta(\varepsilon) > 0$ small enough so that the convolution $\tilde{f} \star \varphi_\delta$ satisfies

$$\left\|\tilde{f} - \tilde{f} \star \varphi_\delta\right\|_{L^1(\mathbb{R}^D)} < \frac{\varepsilon}{3}. \tag{47}$$

Note that $\tilde{f} \star \varphi_\delta$ is the density of probability measure on $\mathbb{R}^D$; namely, the law of a random variable which is the sum of a Gaussian random variance with law $\mathcal{N}(0, \delta I_N)$ and a random variable with law $\mu$. That is, $\tilde{f} \star \varphi_\delta \lambda \in L_+^1(\mathbb{R}^D)$. Together (46) and (47) imply that

$$\left\|f - \tilde{f} \star \varphi_\delta\right\|_{L^1(\mathbb{R}^D)} < \frac{2\varepsilon}{3}. \tag{48}$$

Recall the definition of the convolution: for each $x \in \mathbb{R}^D$ we have

$$\tilde{f}(x) \star \varphi_\delta \stackrel{\text{def.}}{=} \int_{u \in \mathbb{R}^D} \tilde{f}(u) \varphi_\delta(x - u) \lambda(du). \tag{49}$$

Since $\tilde{f}, \varphi_\delta \in C_c^\infty(\mathbb{R}^D)$ then Lebesgue integral of their product coincides with the Riemann integral of their product; whence, there is an $N \stackrel{\text{def.}}{=} N(\varepsilon) \in \mathbb{N}$ "large enough" so that

$$\left\|\int_{u \in \mathbb{R}^D} \tilde{f}(u) \varphi_\delta(x - u) \lambda(du) - \sum_{n=1}^{N} \tilde{f}(u_n) \varphi_\delta(x - u_n) \lambda(du)\right\|_{L^1(\mathbb{R}^D)} < \frac{\varepsilon}{3} \tag{50}$$

for some $u_1, \ldots, u_N \in \mathbb{N}$. Note that, $\sum_{n=1}^{N} \tilde{f}(u_n) \varphi_\delta(x - u_n)$ is the law of a Gaussian mixture. Therefore, combining (48) and (50) implies that

$$\left\|f - \sum_{n=1}^{N} \tilde{f}(u_n) \varphi_\delta(x - u_n) \lambda(du)\right\|_{L^1(\mathbb{R}^D)} < \varepsilon. \tag{51}$$

Finally, recalling that the total variation distance between two measures with integrable Lebesgue density equals the $L^1(\mathbb{R}^D)$ norm of the difference of their densities; yields the conclusion; i.e.

$$d_{TV}(\nu, \hat{\nu}) = \left\|f - \sum_{n=1}^{N} \tilde{f}(u_n) \varphi_\delta(x - u_n) \lambda(du)\right\|_{L^1(\mathbb{R}^D)} < \varepsilon$$

where $\frac{D\hat{\nu}}{D\lambda} \stackrel{\text{def.}}{=} \sum_{n=1}^{N} \tilde{f}(u_n) \varphi_\delta(x - u_n) \lambda(du)$. $\qquad \square$

**Lemma E.2** (The space $(\mathcal{P}_1^+(\mathbb{R}^D), d_{TV})$ is Approximate Simplicial). *Let $\hat{\mathcal{Y}} \overset{\text{def.}}{=} \bigcup_{N \in \mathbb{N}} \Delta_N \times [\mathcal{P}_1^+(\mathbb{R}^D)]^N$ and define the map $\eta : \hat{\mathcal{Y}} \mapsto \mathcal{P}_1^+(\mathbb{R}^D)$ by*

$$\eta(w, (r_n)_{n=1}^N) \overset{\text{def.}}{=} \sum_{n=1}^N w_n\, r_n.$$

*Then, $\eta$ is a mixing function, in the sense of (Acciaio et al., 2024, Definition 3.1). Consequentially, $(\mathcal{P}_1^+(\mathbb{R}^D), \eta)$ is approximately simplicial.*

*Proof.* Let $\mathcal{M}^+(\mathbb{R}^D)$ denote the Banach space of all finite signed measures on $\mathbb{R}^D$ with finite total variation norm $\|\cdot\|_{TV}$. Since $\|\cdot - \cdot\|_{TV} = d_{TV}$ when restricted to $\mathcal{P}_1^+(\mathbb{R}^D) \times \mathcal{P}_1^+(\mathbb{R}^D)$ and since $\|\cdot\|_{TV}$ is a norm, then the conclusion follows from (Acciaio et al., 2024, Example 5.1) and since $\mathcal{P}_1^+(\mathbb{R}^D)$ is a convex subset of $\mathcal{M}^+(\mathbb{R}^D)$. □

Together, Lemmata E.1 and E.2 imply that $(\mathcal{P}_1^+(\mathbb{R}^D), d_{TV}, \eta, Q)$ is a QAS space in the sense of (Acciaio et al., 2024, Definition 3.4), where $Q \overset{\text{def.}}{=} (GMM_M)_{M \in \mathbb{N}}$. Consequently, the following is a geometric attention mechanism in the sense of (Acciaio et al., 2024, Definition 3.5)

$$\hat{\eta} : \cup_{N \in \mathbb{N}} \Delta_N \times \mathbb{R}^{N \times D_M} \to \mathcal{P}_1^+(\mathbb{R}^D)$$

$$\left(w, \left(v_m, (b_{mn})_{n=1}^N, (B_{mn})_{n=1}^N\right)_{m=1}^M\right) \mapsto \sum_{n=1}^N w_n \sum_{m=1}^M Proj_{\Delta_M}(v_m)_n\, \nu\big(b_{mn}, \varphi(B_{mn})\big).$$

Before presenting our main theorem, we first introduce several definitions of activation functions that will be used in the theorem. These definitions, which are essential for completeness, are taken from (Acciaio et al., 2024, Definitions 2.2-2.4).

**Definition E.3** (Trainable Activation Function: Singular-ReLU Type). A trainable activation function $\sigma$ is of *ReLU+Step type* if

$$\sigma_\alpha : \mathbb{R} \ni x \mapsto \alpha_1 \max\{x, \alpha_2 x\} + (1 - \alpha_1)\lfloor x \rfloor \in \mathbb{R}$$

**Definition E.4** (Trainable Activation Function: Smooth-ReLU Type). A trainable activation function $\sigma$ is of *smooth non-polynomial type* if there is a non-polynomial $\sigma^\star \in C_c^\infty(\mathbb{R})$, for which

$$\sigma_\alpha : \mathbb{R} \ni x \mapsto \alpha_1 \max\{x, \alpha_2 x\} + (1 - \alpha_1)\sigma^\star(x) \in \mathbb{R}$$

**Definition E.5** (Classical Activation Function). Let $\sigma^\star \in C_c^\infty(\mathbb{R})$ be non-affine and such that there is some $x \in \mathbb{R}$ at which $\sigma$ is differentiable and has non-zero derivative. Then $\sigma$ is a classical regular activation function if, for every $\alpha \in \mathbb{R}^2$, $\sigma_\alpha = \sigma^\star$.

Further in the text, we assume that activation functions are applied element-wise to each vector $x \in \mathbb{R}^D$. We are now ready to prove the first part of our approximation theorem.

**Proposition E.6** (Deep Gaussian Mixtures are Universal Conditional Distributions in the TV Distance). *Let $\pi : (\mathbb{R}^D, \|\cdot\|_2) \to (\mathcal{P}_1^+(\mathbb{R}^D), d_{TV})$ be Hölder. Then, for every compact subset $K \subseteq \mathbb{R}^D$, every approximation error $\varepsilon > 0$ there exists $M, N \in \mathbb{N}$ and a MLP $\hat{f} : \mathbb{R}^D \mapsto \mathbb{R}^{N \times ND_M}$ with activations as in Definitions E.3, E.4, E.5 such that the (non-degenerate) Gaussian-mixture valued map*

$$\hat{\pi}(\cdot|x) \overset{\text{def.}}{=} \hat{\eta} \circ \hat{f}(x)$$

*satisfies the uniform estimate*

$$\max_{x \in K} d_{TV}\big(\hat{\pi}(\cdot|x) \| \pi(\cdot|x)\big) < \varepsilon.$$

*Proof.* Since Lemmata E.2 and E.1 imply that $(\mathcal{P}_1^+(\mathbb{R}^D), d_{TV}, \eta, Q)$, is a QAS space in the sense of (Acciaio et al., 2024, Definition 3.4), then the conclusion follows directly from (Acciaio et al., 2024, Theorem 3.8). □

Since many of our results are formulated in the Kullback-Leibler divergence, then our desired guarantee is obtained only under some additional mild regularity requirements of the target conditional distribution $\hat{\pi}$ being approximated.

**Assumption E.7** (Regularity of Conditional Distribution). Let $\pi : (\mathbb{R}^D, \|\cdot\|_2) \to (\mathcal{P}_1^+(\mathbb{R}^D), d_{TV})$ be Hölder and, for each $x \in \mathbb{R}^D$, $\pi(\cdot|x)$ is absolutely continuous with respect to the Lebesgue measure $\lambda$ on $\mathbb{R}^D$. Suppose that there exist some $0 < \delta \leq \Delta$ such that its conditional Lebesgue density satisfies

$$\delta \leq \frac{d\pi(\cdot|x)}{d\lambda} \leq \Delta \qquad \text{for all } x \in \mathbb{R}^D. \tag{52}$$

**Theorem E.8** (Deep Gaussian Mixtures are Universal Conditional Distributions). *Suppose that $\pi$ satisfies Assumption E.7. Then, for every compact subset $K \subseteq \mathbb{R}^{D_x}$, every approximation error $\varepsilon > 0$ there exists $M, N \in \mathbb{N}$ such that: for each $m = 1, \ldots, M$ and $n = 1, \ldots, N$ there exist MLPs: $a_m : \mathbb{R}^{D_x} \mapsto \mathbb{R}^{D_y}, v_m : \mathbb{R}^{D_x} \mapsto \mathbb{R}^M$ with ReLU activation functions and $w_n, B_n$ learnable parameters such that the (non-degenerate) Gaussian-mixture valued map*

$$\hat{\pi}(\cdot|x) \stackrel{\text{def.}}{=} \sum_{n=1}^{N} \sum_{m=1}^{M} z_{mn}(x) \, \nu\big(d_{mn}(x), \varphi(D_{mn}(x))\big)$$

*satisfies the uniform estimate*

$$\max_{x \in K} d_{TV}\big(\pi(\cdot|x), \hat{\pi}(\cdot|x)\big) < \varepsilon. \tag{53}$$

*If, moreover, $\hat{\pi}$ also satisfies (52) (with $\hat{\pi}$ in place of $\pi$) then additionally*

$$\max_{x \in K} \text{KL}\big(\pi(\cdot|x), \hat{\pi}(\cdot|x)\big) \in \mathcal{O}(\varepsilon), \tag{54}$$

*where $\mathcal{O}$ hides a constant independent of $\varepsilon$ and of the dimension $D$.*

The proof of Theorem E.8 makes use of the *symmetrized Kullback-Leibler divergence* $\text{KL}_{sym}$ which is defined for any two $\alpha, \beta \in \mathcal{P}(\mathbb{R}^D)$ by $\text{KL}_{sym}(\mu, \nu) \stackrel{\text{def.}}{=} \text{KL}(\alpha\|\beta) + \text{KL}(\beta\|\alpha)$; note, if $\text{KL}_{sym}(\alpha, \beta) = 0$ then $\text{KL}_{sym}(\alpha\|\beta) = 0$. We now prove our main approximation guarantee.

*Proof of Theorem E.8.* To simplify the explanation of our first claim, we provide the expression for $\hat{\pi}(y|x)$ from (19):

$$\hat{\pi}(y|x) = \sum_{n=1}^{N} w_n \sum_{m=1}^{M} v_m(x) \exp\left(\frac{a_m^\top(x) B_n a_m(x) + 2b_n^\top a_m(x)}{2\varepsilon}\right) \mathcal{N}(y \,|\, d_{mn}(x), \varepsilon B_n)$$

Thanks to the wide variety of activation functions available from Definitions E.3, E.4, E.5, we can construct the map $\hat{f}$ and directly apply Proposition E.6. This completes the proof of the first claim.

Under Assumption E.7, $\pi(\cdot|x)$ and $\hat{\pi}(\cdot|x)$ are equivalent to the $D$-dimensional Lebesgue measure $\lambda$. Consequently, for all $x \in \mathbb{R}^{D_x}$:

$$\pi(\cdot|x) \ll \hat{\pi}(\cdot|x)$$

Therefore, the Radon-Nikodym derivative $\frac{\hat{\pi}(\cdot|x)}{\pi(\cdot|x)}$ is a well-defined element of $L^1(\mathbb{R}^{D_x})$, for each $x \in \mathbb{R}^{D_x}$; furthermore, we have

$$\frac{\pi(\cdot|x)}{\hat{\pi}(\cdot|x)} = \frac{\pi(\cdot|x)}{d\lambda} \frac{d\lambda}{\hat{\pi}(\cdot|x)}. \tag{55}$$

Again, leaning on Assumption (52) and the Hölder inequality, we deduce that

$$\begin{aligned}
\sup_{a \in \mathbb{R}^D} \left|\frac{\pi(\cdot|x)}{\hat{\pi}(\cdot|x)}(a)\right| &= \sup_{a \in \mathbb{R}^D} \left|\frac{\pi(\cdot|x)}{d\lambda}(a) \frac{d\lambda}{\hat{\pi}(\cdot|x)}(a)\right| \\
&\leq \sup_{a \in \mathbb{R}^D} \left|\frac{\pi(\cdot|x)}{d\lambda}(a)\right| \sup_{a \in \mathbb{R}^D} \left|\frac{d\lambda}{\hat{\pi}(\cdot|x)}(a)\right| \\
&\leq \sup_{a \in \mathbb{R}^D} \left|\frac{\pi(\cdot|x)}{d\lambda}(a)\right| \frac{1}{\delta} \\
&\leq \frac{\Delta}{\delta}
\end{aligned} \tag{56}$$

where the final inequality under the assumption that $\hat{\pi}$ also satisfies Assumption 52. Importantly, we emphasize that the right-hand side of (56) holds *independently of* $x \in \mathbb{R}^{D_x}$ ("which we are conditioning on"). A nearly identical estimate holds for the corresponding lower-bound. Therefore, we may apply (Sason, 2015, Theorem 1) to deduce that: there exists a constant $C > 0$ (independent of $x \in \mathbb{R}^{D_x}$ and depending only on the quantities $\frac{\Delta}{\delta}$ and $\frac{\delta}{\Delta}$; thus only on $\delta, \Delta$) such that: for each $x \in \mathbb{R}^{D_x}$

$$\mathrm{KL}\left(\pi(\cdot|x), \hat{\pi}(\cdot|x)\right) \leq C \, d_{TV}\left(\pi(\cdot|x), \hat{\pi}(\cdot|x)\right). \tag{57}$$

The conclusion now follows, since the right-hand side of (57) was controllable by the first statement; i.e. since (53) holds we have

$$\mathrm{KL}\left(\pi(\cdot|x), \hat{\pi}(\cdot|x)\right) \leq C \, d_{TV}\left(\pi(\cdot|x), \hat{\pi}(\cdot|x)\right) \leq C\varepsilon. \tag{58}$$

A nearly identical derivation shows that

$$\mathrm{KL}\left(\hat{\pi}(\cdot|x), \pi(\cdot|x)\right) \leq C\varepsilon. \tag{59}$$

Combining (58) and (59) yields the following bound

$$\max_{x \in K} \mathrm{KL}_{sym}\left(\pi(\cdot|x), \hat{\pi}(\cdot|x)\right) \in \mathcal{O}(\varepsilon). \tag{60}$$

Since $\mathrm{KL}(\alpha \| \beta) \leq \mathrm{KL}_{sym}(\alpha, \beta)$ for every pair of Borel probability measures $\alpha$ and $\beta$ on $\mathbb{R}^{D_x}$ then (60) implies (54).

$\square$

| Source | Target | FSBM | Ours |
|--------|--------|------|------|

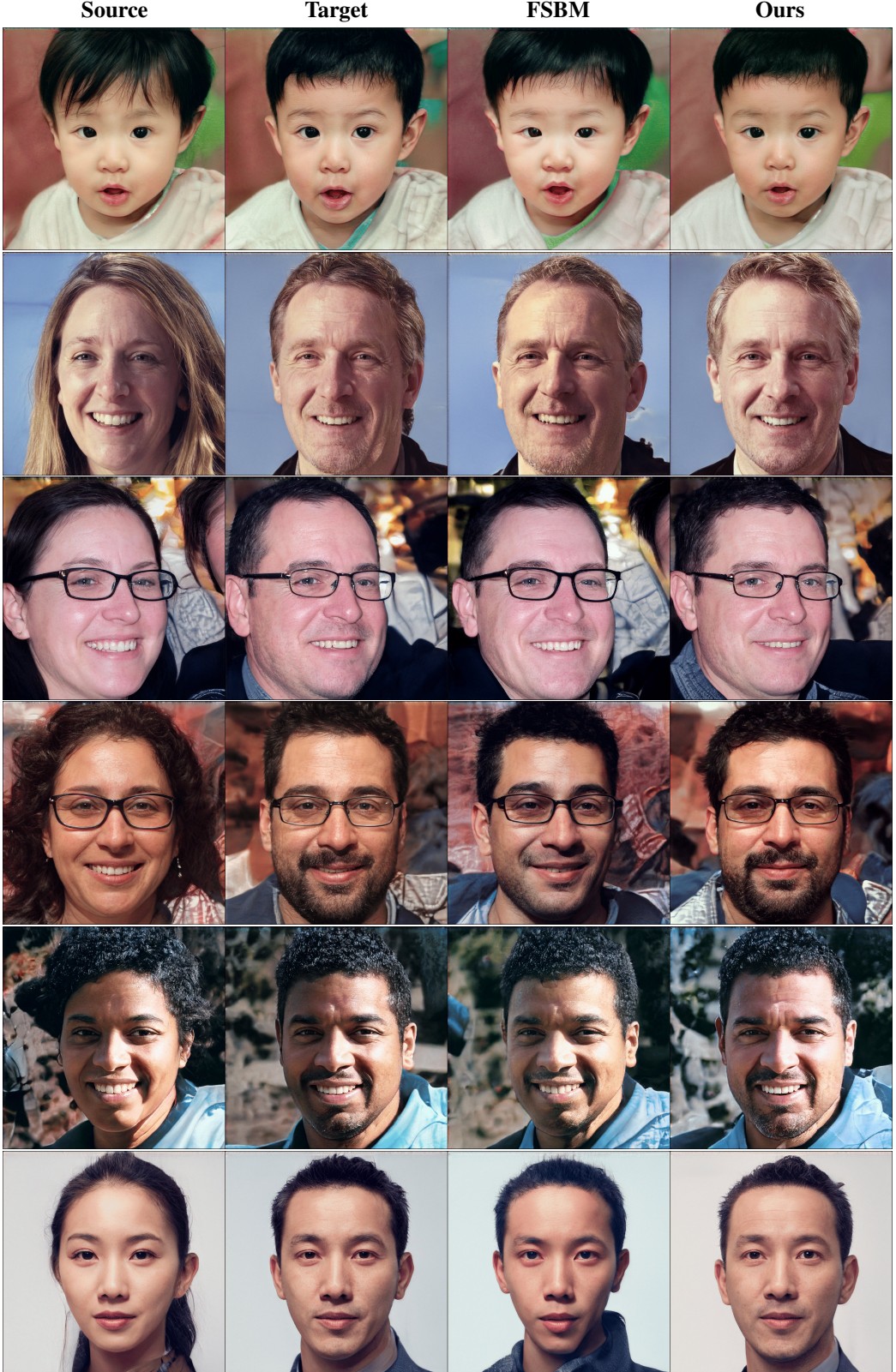

*Figure 10.* Extended visual comparisons between the FSBM (Theodoropoulos et al., 2024) method (3rd column) and our method (4th column) for Woman-to-Man translation are shown here. The task is described in §5.3. The first column shows the source image and the second column the target image.

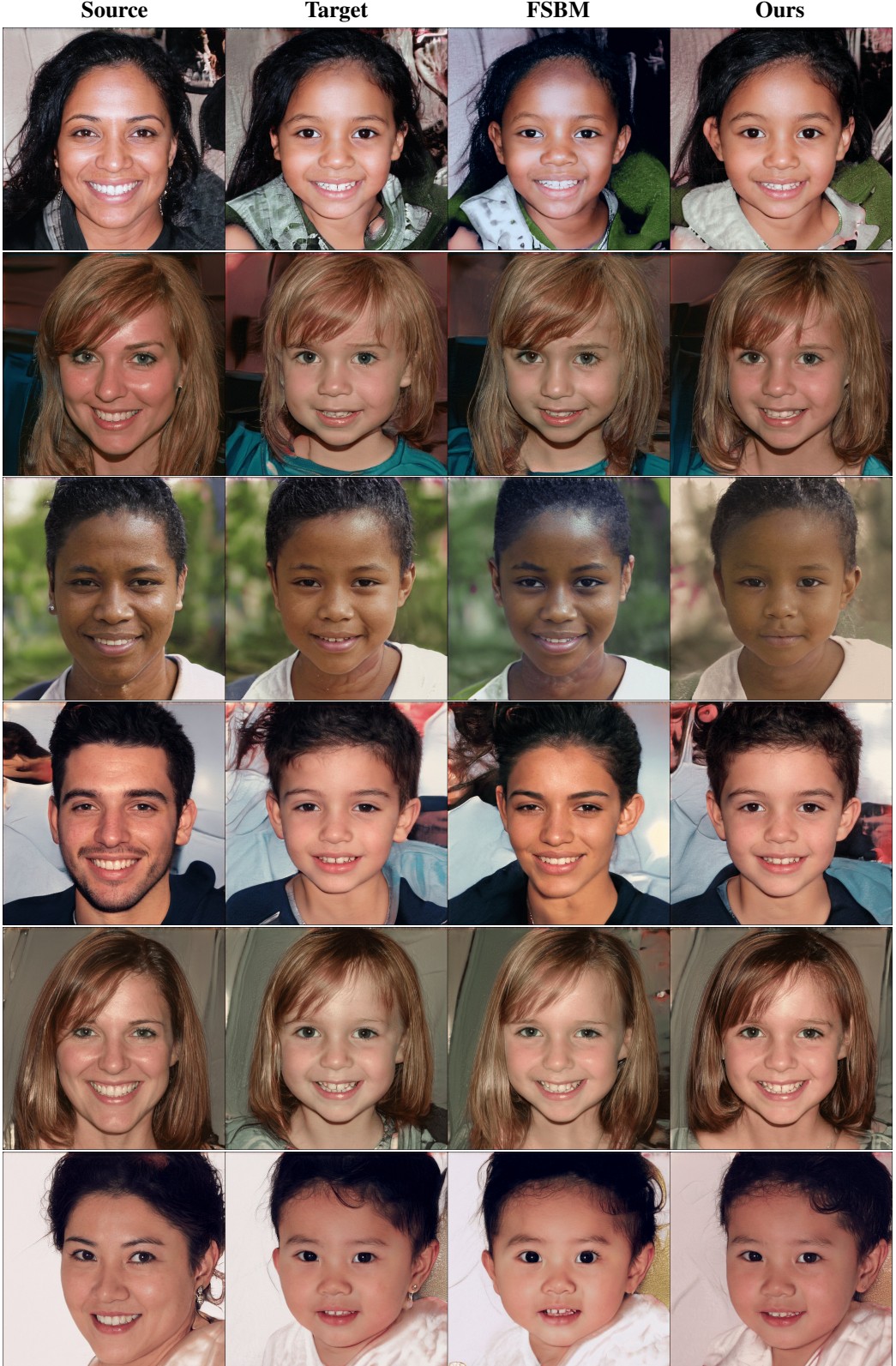

*Figure 11.* Visual comparisons for the Old-to-Young translation task between the FSBM (Theodoropoulos et al., 2024) method (3rd column) and our method (4th column). The task is described in §5.3. The first column displays the source image, and the second column shows the target image.

