# OpenReview forum: "Inverse Entropic Optimal Transport Solves Semi-supervised Learning via Data Likelihood Maximization"
_ICML.cc/2026/Conference — ICML 2026 regular_

### Official Review · Reviewer_cjEn · 2026-03-05

**Soundness:** 3
**Presentation:** 3
**Significance:** 3
**Originality:** 3
**Overall Recommendation:** 5
**Confidence:** 3

**Summary:**

This paper addresses the problem of learning conditional distributions pi*(.|x) in a semi-supervised setting where both paired samples (x,y) ~ pi* and unpaired samples x ~ pi*_x, y ~ pi*_y are available. The authors derive a loss function from first principles via KL-divergence minimization (equivalently, data likelihood maximization), using a Gibbs-Boltzmann energy parameterization that decomposes into a cost function c_theta(x,y) and a dual potential f_theta(y). This decomposition naturally separates terms requiring paired data from those using only unpaired marginals, yielding a principled objective (Eq. 15) that seamlessly integrates both data types. The authors then show this loss is equivalent to the inverse entropic optimal transport problem, connecting semi-supervised learning to OT theory. For tractability, they propose a Gaussian mixture parameterization that yields closed-form normalization constants and conditional distributions, and prove a universal approximation result. Experiments on a 2D synthetic task, a weather prediction dataset, and image translation in ALAE latent space demonstrate the method's ability to leverage unpaired data alongside limited paired supervision.

**Compliance With Llm Reviewing Policy:**

Affirmed.

**Key Questions For Authors:**

1. **Can you provide quantitative metrics for the Swiss Roll experiment?** Reporting log-likelihood, Wasserstein-2 distance, or similar metrics across all baselines would allow objective comparison beyond visual inspection. If your method's advantage is statistically significant under proper metrics, this would substantially strengthen the empirical contribution.

2. **How does the method perform on a benchmark where the ground truth is NOT constructed via OT?** For example, a conditional distribution defined by a known stochastic function y = g(x) + noise, or a standard conditional generation benchmark. This would address the concern about the synthetic benchmark design (W2) and increase confidence in generality.

3. **What is the computational scaling behavior of the Gaussian mixture parameterization as M, N, D_x, and D_y grow?** Wall-clock training/inference times and memory usage as a function of these parameters (even on synthetic data) would help readers assess practical applicability. If the method remains tractable for moderate dimensions (e.g., D_y ~ 1000), this would significantly mitigate W1.

**Limitations:**

Yes — the authors discuss limitations adequately. They identify the Gaussian mixture parameterization as a scalability bottleneck, acknowledge the instability of the neural EBM variant (Appendix A), and note divergence in extremely low-data regimes (Appendix C.3). The restriction to continuous targets is stated explicitly, with discrete extensions noted as future work. One additional limitation worth noting: the partially paired data correction (Appendix B.3) is presented theoretically but not validated experimentally.

**Strengths And Weaknesses:**

### Strengths:

S1. **Principled and elegant derivation (Soundness/Originality).** The loss function is derived cleanly from KL-divergence minimization via a well-motivated Gibbs-Boltzmann energy parameterization. The key insight — decomposing the energy into c_theta(x,y) - f_theta(y) to separate joint and marginal terms — is simple yet effective. Unlike prior heuristic composite losses for semi-supervised domain translation, this approach has a clear probabilistic interpretation and does not require manual balancing of loss terms.

S2. **Connection to inverse entropic OT (Originality/Significance).** The equivalence between the proposed loss and inverse EOT is a genuinely interesting theoretical observation. It bridges two previously disconnected lines of work — semi-supervised learning and inverse optimal transport — and could facilitate cross-pollination. The observation that inverse EOT admits a maximum likelihood interpretation may inspire new solvers leveraging probabilistic modeling tools.

S3. **Tractable parameterization with theoretical guarantees (Soundness).** The Gaussian mixture parameterization delivers closed-form expressions for the normalization constant Z_theta(x) (Proposition 3.1) and conditional distributions (Proposition 3.2), enabling straightforward end-to-end training and fast inference. The universal approximation result (Theorem 3.3) provides formal assurance that this parameterization class can recover arbitrary conditional distributions under mild regularity assumptions (proofs checked best-effort).

S4. **Comprehensive synthetic baseline comparison (Presentation/Soundness).** The Swiss Roll experiment compares against 12 baselines spanning regression, GANs, normalizing flows, and OT-based methods. The paper provides detailed loss formulations for each baseline (Appendix D.2), explains data usage per method (Table 5), and tests under both low-data and high-data regimes. The ablation on source vs. target unpaired data (Appendix D.4) is informative.

### Weaknesses:

W1. **Limited experimental scale and scope (Significance/Soundness).** All experiments operate in low-to-moderate dimensions: 2D synthetic, 94D weather features (with M=1 mixture component), and 512D ALAE latent space. There are no pixel-space image translation experiments, which is the most natural application domain. The Gaussian mixture parameterization, requiring M x N components, is acknowledged as a scalability bottleneck but is not addressed experimentally. The practical applicability to realistic-scale problems remains undemonstrated.

W2. **Synthetic benchmark design may favor OT-based methods (Soundness).** The ground truth plan pi* in the Swiss Roll experiment is constructed via the Sinkhorn algorithm with a custom bimodal cost — by definition an entropic OT plan, precisely the structure the proposed method is designed to recover. This creates an inherent advantage over non-OT baselines (regression, CGAN, CondNF). A more neutral benchmark where the ground truth does not arise from an OT construction would provide a stronger test of generality.

W3. **Missing quantitative metrics on the synthetic task (Soundness).** The Swiss Roll experiment relies entirely on visual comparison (Figure 3). No quantitative metrics (log-likelihood, Wasserstein distance, total variation, MMD) are reported for this key experiment. [Note: Figures not visually assessed.] Given that several baselines may produce qualitatively similar outputs, quantitative evaluation is essential for meaningful comparison. Only the weather prediction experiment provides numerical metrics.

W4. **The neural network parameterization needed for scalability is underdeveloped (Soundness).** The fully neural EBM/MCMC variant (Appendix A) is presented as a proof-of-concept but shows clear instability (acknowledged in the text). Since the Gaussian mixture parameterization limits scalability, the neural variant is essential for the method's future applicability, yet it receives only cursory treatment with no quantitative evaluation, no hyperparameter sensitivity analysis, and no comparison with the GMM variant under controlled conditions.

---

> ### Author Rebuttal · Authors · 2026-03-31
>
> Dear Reviewer cjEn,
>
> We thank you for highlighting the strengths of our work, including the principled KL-based derivation, the connection to inverse EOT, the tractable GMM parameterization, the universal approximation result (Theorem 3.3), and the comprehensive synthetic comparisons.
>
> Below, we address all your questions and concerns in detail.
>
> ---
> **1. "Limited experimental scale and scope. [...] There are no pixel-space image translation experiments, which is the most natural application domain. [...] The neural network parameterization needed for scalability is underdeveloped [...]"**
>
> As we note in lines 428-435, the Gaussian mixture parameterization is not intended as a scalable solution for high-dimensional problems, and we explicitly discuss this limitation in the paper. To address it, Appendix A introduces a fully neural parameterization, and Figure 5 provides a proof-of-concept experiment on $3\times32\times32$ RGB images using neural energy-based models. This experiment shows that the proposed framework is not restricted to GMMs and can be extended to image data.
>
> Scaling the neural EBM parameterization to higher-resolution image domains is a promising extension, and prior work [1,2] indicates that this is primarily an implementation and modeling question rather than a limitation of our framework. Our goal in this paper is to establish the method itself: we propose a principled likelihood-based objective and show in controlled experiments that standard semi-supervised baselines can fail to recover the correct conditional transport plan even in relatively simple settings.
>
> ---
> **2. "[...] The ground truth plan pi\* in the Swiss Roll experiment is constructed via the Sinkhorn algorithm [...] How does the method perform on a benchmark where the ground truth is NOT constructed via OT? [...]"**
>
> We humbly note that the choose of the paired data should not affects the method applicability, since method agnostic to the chose of the pairs on that it trained. However, following your suggestion we implemented the experiment for $y = g(x)+\varepsilon$, where
> $$
> g(x)=(t\cos(t), t\sin(t)), t=\tan(\Vert x\Vert).
> $$
> $g(x)$ is the analytical transformation for Swiss-Roll, and we applied the transformation $\tan{(\Vert x\Vert)}$. The final Swiss-Roll looks the same, however, ground-truth conditional plan $\pi^\star(\cdot\vert x)$ is different. The metric results for this setup descibed below.
>
> ---
> **3. "Missing quantitative metrics on the synthetic task [...] Can you provide quantitative metrics for the Swiss Roll experiment? [...]"**
>
> We agree that adding quantitative evaluation strengthens the Swiss Roll experiment. Accordingly, we complemented the qualitative comparison in Figure 3 with **MMD** [3] and **Sinkhorn divergence** ($\mathcal{W}_2$). The results are available in the metrics folder at the **anonymous** `Google Drive` link:
>
> https://drive.google.com/drive/folders/1s4oNlAvVj4RsV_eYBasNaDP51vtiHSHH?usp=sharing
>
> For the conditional evaluation, we use the same three starting points as in the paper and generate 1024 samples per point, providing a direct quantitative counterpart to Figure 3. Increasing the number of starting points and varying seeds would reduce metric variance, but time and computational constraints prevented this; we plan to include it in the revised manuscript.
>
> Due to retraining costs, we did not rerun every baseline configuration. Nevertheless, for the main 2D setup, we report results for both $P=128$, $Q=R=1024$ and $P=Q=R=16k$, as well as for the reviewer-requested setting $P=128$, $Q=R=1024$. These quantitative results will be included in the revision.
>
> ---
> **4. What is the computational scaling behavior of the Gaussian mixture parameterization [...].**
>
> In Appendix C.2, we report that on our hardware, 25K iterations take approximately 20 minutes. We believe our implementation is not fully optimized and could be improved. Since runtime and memory usage depend heavily on implementation details and hardware, such measurements are inherently system- and implementation-specific.
>
> ---
> **Conclusion**
>
> We hope these clarifications sufficiently address your concerns. If you have any additional questions, we would be glad to elaborate further. Otherwise, we kindly ask you to reconsider your evaluation in light of the explanations provided.
>
> ---
> **References**
>
> [1] Geng, Cong, et al. "Improving adversarial energy-based model via diffusion process." ICML, 2024.
>
> [2] Zhu, Yaxuan, et al. "Learning Energy-Based Models by Cooperative Diffusion Recovery Likelihood." ICLR, 2024.
>
> [3] Gretton, Arthur, et al. "A kernel two-sample test." JMLR, 2012.
>
> [4] Feydy, Jean, et al. "Interpolating between optimal transport and mmd using sinkhorn divergences." AISTATS, 2019.

---

> > ### Author Rebuttal · Reviewer_cjEn · 2026-03-31
> >
> > I thank the authors for the thorough and responsive rebuttal. Two of my four weaknesses are directly addressed with new experiments:
> >
> > - **W2 (synthetic benchmark favoring OT):** The non-OT Swiss Roll experiment with an analytical transformation directly addresses my concern. The method's applicability is not restricted to OT-constructed ground truths. This resolves W2.
> > - **W3 (missing quantitative metrics):** MMD and Sinkhorn divergence metrics are now provided for the Swiss Roll experiment, with results for both the original and reviewer-requested settings. This resolves W3.
> >
> > The remaining concerns (W1: limited scale, W4: neural variant underdeveloped) are acknowledged by the authors and were already honestly discussed in the paper. For a primarily theoretical contribution establishing a new principled framework and connection to inverse EOT, the current experimental scope is acceptable.
> >
> > The rebuttal demonstrates that the authors engaged constructively with every concern, ran new experiments on short timelines, and were transparent about implementation limitations. The theoretical framework is clean, the connection to inverse EOT is novel, and the experimental gaps have been meaningfully narrowed.
> >
> > I raise my score to **5 (Accept)**. The principled derivation, the inverse EOT connection, the tractable parameterization with universal approximation guarantees, and the now-strengthened experimental evaluation collectively make a solid contribution to the field.

---

> > > ### Author Response · Authors · 2026-04-07
> > >
> > > Dear Reviewer cjEn,
> > >
> > > Thank you for your thoughtful feedback and for raising your score. We appreciate your constructive comments, which helped us strengthen the paper.
> > >
> > > In response, we have made cosmetic revisions to improve clarity: we reorganized the Appendix to better structure the additional experiments and discussions, and we will include the new experimental setups and quantitative metrics in the revised manuscript.

---

### Official Review · Reviewer_47Ta · 2026-03-11

**Soundness:** 4
**Presentation:** 2
**Significance:** 3
**Originality:** 4
**Overall Recommendation:** 4
**Confidence:** 4

**Summary:**

The paper addresses the fundamental problem of learning conditional distributions $\pi^{*}(y|x)$ in a semi-supervised setting, where the training data consists of a limited number of paired samples and a larger set of unpaired samples from the marginal distributions. The authors propose a novel learning objective grounded in the principle of likelihood maximization. They reveal a theoretical equivalence between this likelihood-based loss and the inverse entropic optimal transport (EOT) problem. Leveraging this connection, they develop an end-to-end learning algorithm using a Gaussian mixture parameterization for the cost function, which allows for closed-form expressions of the loss terms. The authors also provide a theoretical proof of the universal approximation property for their method and demonstrate its effectiveness through empirical tests on 2D synthetic tasks and high-dimensional latent space domain translation.

**Compliance With Llm Reviewing Policy:**

Affirmed.

**Key Questions For Authors:**

## Questions for Authors:
1. Mixture Sensitivity: How sensitive is the model performance to the choice of the number of mixtures $M$ and $N$ in parameterization? Is there a heuristic for selecting these based on the complexity of the target distribution?
2. Paired Data Efficiency: Your results show the method works with a "modest amount" of paired data. Can you quantify the minimum ratio of paired-to-unpaired data required for the model to remain stable in higher-dimensional tasks?
3. In the discussion of related work on inverse entropic optimal transport, the paper states that the inverse problem is invariant with respect to the entropic regularization parameter. However, this claim is somewhat unclear. Intuitively, if the entropic regularization coefficient becomes very large, the entropy term would dominate the objective, pushing the transport plan toward a more uniform distribution. Please clarify it.

**Limitations:**

yes

**Strengths And Weaknesses:**

### Strengths:
1. Theoretical Grounding: Unlike many semi-supervised domain translation methods that rely on heuristic combinations of supervised and unsupervised losses, this work derives its objective from first principles (likelihood maximization). The link to inverse EOT opens potential future avenues.

2. Stability: Introduction of an objective that naturally integrates both paired and unpaired data for semi-supervised learning. The proposed objective is non-minimax, avoiding the common training instabilities associated with GAN-based approaches.

3. Versatility: The method performs well in scenarios where other models fail, such as capturing bi-modal conditional distributions in 2D tasks where models like CondNF overfit or others like FSBM produce biased results.


### Weaknesses:
1. Scalability Concerns: The reliance on a Gaussian Mixture parameterization for the cost function may limit direct scalability to high-dimensional raw data (e.g., raw pixel space). While the authors show success in latent spaces, a fully neural version is only explored as a proof-of-concept in the appendix.

2. Computational Complexity: Although the Gaussian mixture provides closed-form expressions, the number of mixtures ($M$ and $N$) could still impact computational overhead as the complexity of the data grows.

3. Presentation: Although the paper clearly states its goal at the beginning, the overall flow of the presentation can sometimes be confusing. The exposition starts from the formulation of Inverse Entropic Optimal Transport (EOT), then shifts to a KL-divergence formulation through maximum likelihood. Before the final section where these two perspectives are connected into the final objective, it is difficult for the reader to understand the relationship between them and what the intermediate derivations are aiming to achieve.

---

> ### Author Rebuttal · Authors · 2026-03-31
>
> Dear Reviewer 47Ta,
>
> We thank you for recognizing the theoretical foundation of our work, including the likelihood-based formulation, the connection to inverse EOT, and the stability of our non-minimax objective, as well as its strong performance in recovering ground-truth conditional mappings.
>
> ---
> **1. "Scalability Concerns: [...]"**
>
> We agree that the GMM parameterization may be limiting for very high-dimensional data. As you noted, in Appendix A we provide a proof-of-concept experiment on raw pixel-space $3 \times 32 \times 32$ RGB images using neural EBMs (Figure 5), demonstrating the feasibility of our approach beyond GMMs.
>
> While scaling neural EBMs to higher-resolution data is possible, prior works [1,2] show that this typically requires substantial engineering effort, which is orthogonal to our main **theoretical contributions**. Our focus in this paper is on methodological and theoretical insights - for instance, Section 5.1 illustrates that many modern semi-supervised approaches fail to recover the ground-truth conditional plan even in low-dimensional settings, suggesting that large-scale versions may also produce arbitrary mappings unrelated to paired data. Therefore, we believe that large-scale optimizations are beyond the scope of the current work.
>
> ---
> **2. "Computational Complexity: [...]"**
>
> We agree that the number of Gaussian components ($M$ and $N$) can impact computational cost as data complexity increases. To address this, Appendix A outlines how our framework can be implemented using fully neural parameterizations, which are better suited for larger-scale tasks where GMMs may become inefficient.
>
> ---
> **3. "Presentation: [...]"**
>
> In lines 120–125 at the beginning of Section 2.2, we note that this background can be skipped without affecting the constructive derivation of our loss.
>
> We included Section 3.2 to better motivate the choice of parameterization. We would also appreciate any further suggestions on how to make this part of the manuscript clearer.
>
> ---
> **4. "Mixture Sensitivity: [...]"**
>
> Following your request, we conducted the ablation study regarding the sensitivity to number of $M$ and $N$. Due to computational constraints and limited time, we run on grid for $M=N= (8, 16, 32, 64)$  and consideres case when $N > M$. We follow the setup proposed by **Reviewer cjEN** for $y = g(x) + \varepsilon$ to construct the ground truth conditional mapping, see details in his response. We computed MMD and Sinkhorn Divergence [3] denoted as $\mathcal{W}_2$. The results can be found at `senitivity_analysis` folder via **anonymous** `google drive` link:
>
> https://drive.google.com/drive/folders/1s4oNlAvVj4RsV_eYBasNaDP51vtiHSHH?usp=sharing.
>
> It can be seen that the bigger number of Gaussians increases the conditional metrics and decreases the unconditional metrics, which is logical since the greater the number of components, the greater the overfitting for the conditional distribution. Anyway, since GMM parametrization is lightweight, it's rather convenient to run the parameter search, e.g., with Optuna [4], based on black-box Bayesian optimization techniques.
>
> ---
> **5. "2. Paired Data Efficiency: [...]"**
>
> Table 1 presents an ablation study over different amounts of paired and unpaired data. The results show that even a modest number of paired samples, when combined with additional unpaired data, significantly improves the quality of the learned conditional mapping. We also refer to our response to **Reviewer T8mU**, where a *semi-supervised classification* experiment with the proposed loss further confirms that incorporating unpaired data leads to improved overall performance.
>
> The optimal number of paired samples is task-dependent and should be tuned in practice, as is common in deep learning applications.
>
> ---
> **6. "3. [...], this claim is somewhat unclear. [...]"**
>
> We address this point in lines 209–213, immediately after defining the inverse EOT problem. Specifically, for any $\varepsilon' > 0$, one can rescale the cost as
> $$
> c(x,y) = \frac{\varepsilon}{\varepsilon'} \, c'(x,y),
> $$
> which scales the objective by a constant but does not change the optimal solution (up to this scaling). We welcome any suggestions on how to make this explanation clearer in the manuscript.
>
> ---
> **Conclusion**
>
> We hope these clarifications sufficiently address your concerns. If you have any additional questions, we would be glad to elaborate further. Otherwise, we kindly ask you to reconsider your evaluation in light of the explanations provided.
>
> ---
> **References**
>
> [1] Geng, Cong, et al. "Improving adversarial energy-based model via diffusion process." ICML, 2024.
>
> [2] Zhu, Yaxuan, et al. "Learning Energy-Based Models by Cooperative Diffusion Recovery Likelihood." ICLR, 2024.
>
> [3] Feydy, Jean, et al. "Interpolating between optimal transport and mmd using sinkhorn divergences." AISTATS, 2019.
>
> [4] Akiba, Takuya, et al. "Optuna: A next-generation hyperparameter optimization framework." KDD, 2019.

---

> > ### Author Rebuttal · Reviewer_47Ta · 2026-04-03
> >
> > My concerns have been sufficiently addressed. Based on a re-reading of the manuscript, I wonder if the authors can provide quantitative bounds on N in Theorem 3.3 in terms of the problem parameters. At such, it seems that this result is qualitative and the desired N can be very large.
> > Regarding presentation, I believe the manuscript would benefit from a more streamlined discussion of the relationship between the inverse EOT and Semi-Sup Learning. While it is all there, it is still hard to appreciate. In this context, can the authors further explain what more comes out of this connection, e.g. can complexity and provable guarantees for results in EOT inform complexity and provable guarantees in SSL? Are there any surprises that emerge?

---

> > > ### Author Response · Authors · 2026-04-07
> > >
> > > Dear Reviewer 47Ta,
> > >
> > > We greatly appreciate your suggestion to provide a quantitative bound on $N$ in Theorem 3.3, since the main content of that result is precisely that universality is possible in the infinite-dimensional, non-linear, and non-Euclidean space of EOT conditional densities. In general, there are very few results available on universal approximation in target spaces that are non-Banach, and even fewer in the infinite-dimensional non-Euclidean setting; the authors are only aware of [1]–[3], none of which achieve dimension-free scaling or similarly favourable rates. In short, the approximation theory of infinite-dimensional non-Euclidean spaces is simply not there yet. Additionally, even in the much simpler linear infinite-dimensional setting (whether Hilbert or Banach), favourable approximation rates are currently known only under extreme smoothness assumptions, namely analyticity [5] or holomorphy [4]. Indeed, there are lower bounds showing that optimal minimax rates, even under finite smoothness, must scale exponentially in the reciprocal accuracy [6,7], and only highly specialized settings can overcome such scaling, albeit still not in a dimension-free manner in the infinite-dimensional setting [8].
> > >
> > > Given this state of the art in non-Euclidean universal approximation in infinite dimensions, and since the already novel objective of Theorem 3.3 is to support the correctness of the proposed architecture by showing that it is asymptotically capable of solving the computational EOT problem, we believe that even if favourable scaling were possible, establishing it would rightfully constitute a contribution in and of itself and would therefore be more appropriately published separately. We would be very happy to do so as it would be a solid advance in approximation theory for deep learning in of itself.
> > >
> > > In short, the main novelty of Theorem 3.3 is that it establishes approximability in an infinite-dimensional non-linear setting, where, to the best of our knowledge, no truly comparable results are currently available.
> > >
> > > [1] Acciaio, Beatrice, Anastasis Kratsios, and Gudmund Pammer. "Designing universal causal deep learning models: The geometric (hyper) transformer." Mathematical Finance 34.2 (2024): 671-735.
> > >
> > > [2] Horvath, Blanka, et al. "Transformers Can Solve Non-Linear and Non-Markovian Filtering Problems in Continuous Time For Conditionally Gaussian Signals." (2023).
> > >
> > > [3] Galimberti, Luca. "Neural networks in non-metric spaces." Analysis and Applications (2025): 1-49.
> > >
> > > [4] Adcock, Ben, et al. "Deep Neural Networks Are Effective At Learning High-Dimensional Hilbert-Valued Functions From Limited Data." Mathematical and Scientific Machine Learning. PMLR, 2022.
> > >
> > > [5] Marcati, Carlo, and Christoph Schwab. "Exponential convergence of deep operator networks for elliptic partial differential equations." SIAM Journal on Numerical Analysis 61.3 (2023): 1513-1545.
> > >
> > > [6] Lanthaler, Samuel. "Operator learning with PCA-Net: upper and lower complexity bounds." Journal of Machine Learning Research 24.318 (2023): 1-67.
> > >
> > > [7] Lanthaler, Samuel, and Andrew M. Stuart. "The parametric complexity of operator learning." IMA Journal of Numerical Analysis (2025).
> > >
> > > [8] Furuya, Takashi, et al. "One model to solve them all: 2BSDE families via neural operators." arXiv preprint arXiv:2511.01125 (2025).

---

### Official Review · Reviewer_LAwA · 2026-03-13

**Soundness:** 3
**Presentation:** 3
**Significance:** 3
**Originality:** 3
**Overall Recommendation:** 5
**Confidence:** 4

**Summary:**

This paper studies semi-supervised learning of conditional distributions. The authors propose a likelihood-based objective that integrates supervised and unsupervised data and show that optimizing this objective is equivalent to solving an inverse entropy-regularized optimal transport (OT) problem.

The method includes the conditional transport plan via an energy-based model with a Gaussian mixture structure, which enables tractable training. In addition, the paper also provides a universal approximation result, showing that the proposed parameterization can approximate the true conditional coupling under suitable assumptions.

Empirically, the approach is evaluated on synthetic OT benchmarks and a real weather dataset, demonstrating improved performance compared to heuristic semi-supervised baselines. The work aims to provide a principled OT-based interpretation of semi-supervised conditional learning.

**Compliance With Llm Reviewing Policy:**

Affirmed.

**Final Justification:**

Thank you for the detailed rebuttal. My questions and concerns has been addressed. I keep my assessment of the paper.

**Key Questions For Authors:**

The method appears generally sound. I have one question:

How sensitive is the proposed approach to the choice of the entropy regularization parameter and to the proportion of paired versus unpaired data used during training?

**Limitations:**

yes.

**Strengths And Weaknesses:**

Strengths.

1. The connection between likelihood maximization and inverse entropic OT provides a clear and principled perspective on semi-supervised conditional learning.

2. The work connects optimal transport, energy-based modeling, and semi-supervised learning, which may interest multiple research communities.

Weaknesses.

1. Experiments are relatively small-scale and lack comparisons with modern strong baselines (e.g., diffusion-based conditional models).

2. Inconsistent capitalization is present in the manuscript.
For example, in “Gaussian To Swiss Roll Mapping”, the word “To” should be lowercase (“to”).
Similarly, the title “Neural optimal transport with pair-guided cost functional” (page 27) should follow consistent title capitalization.

---

> ### Author Rebuttal · Authors · 2026-03-31
>
> Dear Reviewer LAwA,
>
> We sincerely appreciate your recognition of our theoretically grounded likelihood-based objective, as well as the principled connection we establish between semi-supervised learning, inverse entropic optimal transport, and energy-based modeling. We are also grateful that you see the potential relevance of our approach across multiple research communities.
>
> Below, we address your questions and concerns in detail.
>
> ---
> **1. "Experiments are relatively small-scale and lack comparisons with modern strong baselines (e.g., diffusion-based conditional models)."**
>
> The concern about missing diffusion-based baselines is understandable; however, we already include the relevant ones.
>
> To the best of our knowledge, OTCS [1] and FSBM [2] are the only diffusion-based methods adapted to this semi-supervised OT setting, and we compare against both. As discussed in Lines 373–377, these methods fail even on simple 2D tasks, where they cannot recover bimodal conditional mappings. This appears to be due to a biased objective induced by the use of artificial cost functions tied to key-point alignment introduced in [3].
>
> Our goal is primarily **methodological**: we introduce a principled likelihood-based objective and show, through controlled 2D experiments, that existing semi-supervised approaches can fail to recover the correct conditional plan even in low-dimensional settings.
>
> ---
> **2. "Inconsistent capitalization is present in the manuscript. For example, in “Gaussian To Swiss Roll Mapping”, the word “To” should be lowercase (“to”). Similarly, the title “Neural optimal transport with pair-guided cost functional” (page 27) should follow consistent title capitalization."**
>
> We thank you for pointing our this missprint. We will fix it in revised version.
>
> ---
> **3. "How sensitive is the proposed approach to the choice of the entropy regularization parameter [...]?"**
>
> As discussed in Lines 209–214, the solution of our objective is invariant to the choice of the entropy regularization parameter $\varepsilon$ up to a scaling of the cost. Specifically, for any $\varepsilon' > 0$, one can rescale the cost as
> $$
> c(x,y) = \tfrac{\varepsilon}{\varepsilon'} \, c'(x,y),
> $$
> which results in the objective being scaled by a constant, without changing the optimal solution (up to this scaling). Therefore, the method is not sensitive to the specific value of $\varepsilon$. In practice, we fix $\varepsilon = 1$ to improve numerical stability and reduce rounding errors during training.
>
> ---
> **4. How sensitive is the proposed approach to [...] the proportion of paired versus unpaired data used during training?**
>
> Table 1 provides an ablation study over different proportions of paired and unpaired data. The results show that even a relatively small amount of unpaired data can significantly improve performance. More generally, the optimal balance between paired and unpaired samples is task-dependent and should be selected based on the specific application and data availability.
>
> ---
> **Conclusion**
>
> We hope these clarifications sufficiently address your concerns. If you have any additional questions, we would be glad to elaborate further. Otherwise, we kindly ask you to reconsider your evaluation in light of the explanations provided.
>
> ---
> **References**
>
> [1] Gu, Xiang, et al. "Optimal transport-guided conditional score-based diffusion model." NeurIPS, 2023.
>
> [2] Theodoropoulos, Panagiotis, et al. "Feedback Schrödinger Bridge Matching." ICLR (oral), 2025.
>
> [3] Gu, Xiang, et al. "Keypoint-guided optimal transport with applications in heterogeneous domain adaptation." NeurIPS, 2022.

---

> > ### Author Rebuttal · Reviewer_LAwA · 2026-03-31
> >
> > Thank you for addressing my questions and concerns. I keep my assessment of the paper.

---

> > > ### Author Response · Authors · 2026-04-07
> > >
> > > Dear Reviewer LAwA,
> > >
> > > Thank you for your positive assessment and thoughtful remarks. We will incorporate your remarks in the manuscript to further improve clarity.

---

### Official Review · Reviewer_T8mU · 2026-03-13

**Soundness:** 2
**Presentation:** 2
**Significance:** 2
**Originality:** 2
**Overall Recommendation:** 4
**Confidence:** 1

**Summary:**

This paper addresses the domain translation problem by proposing a semi-supervised framework to learn conditional distributions when paired data is limited but unpaired marginal data is available. The authors derive a novel loss function grounded in likelihood maximization that seamlessly incorporates both types of data samples within a single objective. They establish a significant theoretical link between this objective and the inverse entropic optimal transport problem, enabling the application of computational OT techniques to semi-supervised learning. To facilitate optimization, the researchers introduce a tractable algorithm using Gaussian mixture parameterization and provide a proof of its universal approximation capabilities. Empirical results across synthetic benchmarks, meteorological forecasting, and latent image translation demonstrate that the method accurately recovers complex distributions more effectively than existing heuristic-based baselines.

**Compliance With Llm Reviewing Policy:**

Affirmed.

**Final Justification:**

The authors have addressed my concerns. Since I have already given a positive rating, I would like to keep my score and vote for acceptance.

**Key Questions For Authors:**

See weakness

**Limitations:**

yes

**Strengths And Weaknesses:**

Strength:
1. The paper provides a mathematically principled bridge between semi-supervised learning and inverse entropic optimal transport.
2. The method demonstrates robust performance across a diverse range of tasks.


Weakness:
1. My main concern is the use of Gaussian Mixture Models (GMM) for parameterization. While I understand it's done to keep the normalization constant tractable, I wonder if this limits the model’s expressiveness. For very high-dimensional data with complex, non-Gaussian structures, a GMM might become a representation bottleneck.
2. Every experiment in the paper focuses on the domain translation problem. Because of this, the current title "Semi-supervised Learning" feels a bit too broad. I’d suggest changing it to "Inverse Entropic Optimal Transport Solves Semi-supervised Domain Translation via Data Likelihood Maximization" to better reflect what the paper actually does.
3. For the image translation part, the experiments are all done in a pre-trained ALAE latent space. Can the method actually handle the messiness of direct pixel-to-pixel translation? It’s hard to tell how much of the success is due to the method itself versus the power of the pre-trained feature extractor.

---

> ### Author Rebuttal · Authors · 2026-03-31
>
> Dear Reviewer T8mU,
>
> We sincerely thank you for your positive assessment of our work. We particularly appreciate your recognition of our likelihood-based formulation and the principled connection we establish between semi-supervised learning and inverse EOT. We are also grateful for your acknowledgment of the robustness of our method across a diverse set of tasks.
>
> Below, we address your questions and concerns in detail.
>
> ---
> **1. "My main concern is the use of Gaussian Mixture Models (GMM) for parameterization.[...]"**
>
> We agree that a GMM)parameterization may become limiting in very high-dimensional settings with complex, non-Gaussian structure. As you correctly noted, our choice of GMMs is primarily motivated by the need to keep the normalization constant tractable, which allows for a clean and efficient instantiation of our objective.
>
> Importantly, this choice is **not inherent to the proposed framework**. As discussed in lines 428–433 and further elaborated in Appendix A, our loss formulation readily extends to more expressive, fully neural parameterizations (e.g., neural energy-based models). To support this claim, we include a proof-of-concept experiment on $3 \times 32 \times 32$ RGB images using neural EBMs (Figure 5), demonstrating that the method can indeed model more complex, non-Gaussian data distributions beyond the GMM setting.
>
> ---
> **2. "Every experiment in the paper focuses on the domain translation problem. Because of this, the current title "Semi-supervised Learning" feels a bit too broad. [...]"**
>
> We agree that the current title may sound broader than the present set of experiments. At the same time, our framework is more general and can be applied beyond domain translation. To illustrate this, we applied the proposed loss in Eq. 15 to a semi-supervised classification task on MNIST, following the sketch in Appendix B.2. The goal is to learn $\pi(y \mid x)$ from a small number of labeled examples together with additional unlabeled samples. We use a discrete energy-based model of Gibbs-Boltzmann form (Eq. 12):
>
> $$
> \pi^{\theta}(y \mid x) \propto \exp\left(\frac{f^\theta_y - c^{\theta}(x,y)}{\varepsilon}\right), \qquad \log Z^{\theta}(x) = \operatorname{LSE}_{y}\left(\frac{f^\theta_y - c^{\theta}(x,y)}{\varepsilon}\right),
> $$
>
> where $f^\theta \in \mathbb{R}^K$ is a learnable vector of class potentials and $c^\theta(x,y)$ is parameterized by a CNN-based cost function. We optimize the same likelihood-based objective combining labeled and unlabeled data:
> $$
> \mathcal{L} = \frac{1}{\varepsilon}\mathbb{E}[c^{\theta}(x_p, y_p)] - \frac{1}{\varepsilon}\mathbb{E}_{y}[f^\theta_y] + \mathbb{E}_x[\log Z^{\theta}(x)].
> $$
>
> We then performed an ablation study by varying the number of paired samples $P$ and unlabeled samples $Q$. The results are available in the `semi_supervised_classification` folder at the following **anonymous** `Google Drive` link:
>
> https://drive.google.com/drive/folders/1s4oNlAvVj4RsV_eYBasNaDP51vtiHSHH?usp=sharing
>
> The heatmaps on both validation and test sets show that adding unlabeled data generally improves accuracy, especially in the low-label regime ($P < 100$). This trend is consistent with the findings already reported in Table 1. At the same time, very large $Q$ can hurt final performance, suggesting overfitting or reduced stability in that regime. We also observe that $P = 200$ paired samples is already sufficient to learn a strong classifier, while the validation heatmaps indicate that some reduction in paired data can be compensated by increasing the amount of unlabeled data.
>
> We will include an extended discussion of this in the revised version. While we acknowledge that these classification experiments are preliminary, we believe it is important to include them in the paper to highlight generality of our approach.
>
> ---
> **3. "[...] Can the method actually handle the messiness of direct pixel-to-pixel translation? [...]"**
>
> Regarding scalability to higher-dimensional or higher-resolution data, prior works [1, 2] have shown that neural EBMs can be successfully applied in such regimes. However, these approaches typically require substantial engineering and optimization techniques (e.g., advanced sampling or architectural design), which are orthogonal to the main focus of our paper. Our goal here is to introduce and validate the **methodological and theoretical contribution**, for which, we believe, the presented experiments already provide a sufficient proof of concept.
>
> ---
> **Conclusion**
>
> We hope these clarifications sufficiently address your concerns. If you have any additional questions, we would be glad to elaborate further. Otherwise, we kindly ask you to reconsider your evaluation in light of the explanations provided.
>
> ---
> **References**
>
> [1] Geng, Cong, et al. "Improving adversarial energy-based model via diffusion process." ICML, 2024.
>
> [2] Zhu, Yaxuan, et al. "Learning Energy-Based Models by Cooperative Diffusion Recovery Likelihood." ICLR, 2024.

---

> > ### Author Rebuttal · Reviewer_T8mU · 2026-04-03
> >
> > The authors have addressed my concerns. Since I have already given a positive rating, I would like to keep my score.

---

> > > ### Author Response · Authors · 2026-04-07
> > >
> > > Dear Reviewer T8mU,
> > >
> > > We thank you for your positive assessment of our work and thoughtful feedback. Following your suggestions, we have made cosmetic revisions to improve clarity: we reorganized the Appendix to better group related experiments, and we will include the additional classification experiment in the revised version.

---

### Decision · Program_Chairs · 2026-04-30

**Decision:**

Accept (regular)

**Comment:**

This paper makes a new connection between inverse entropic optimal
transport and semi-supervised learning. It shows that the solution of the inverse entropic optimal transport problem is equivalent to the solution of a maximum likelihood
problem for a specific model. All reviewers have found the paper interesting and
had mostly minor comments and questions, such as the limit of GMMs and their
usage on image data (using Neural EBM), sensitivity to entropy, limited
experiments and scalability. The authors provided a very good response to the
reviews with new experiments and clarified most of the points that were raised by the reviewers. All
reviewers acknowledged the response and found that their concerns were mostly
resolved.

I agree with the reviewers that this paper makes a very interesting connection and provides a new
perspective on semi-supervised learning.
I would therefore recommend an
acceptance but I would expect the authors to include in the final version all
the clarifications and new experiments that they did in the response.